# Brain-wide presynaptic networks of functionally distinct cortical neurons

Ana R. Inácio[1 ✉], Ka Chun Lam[2,4], Yuan Zhao[2,4], Francisco Pereira[2], Charles R. Gerfen[3] & Soohyun Lee[1 ✉]

Revealing the connectivity of functionally identified individual neurons is necessary to understand how activity patterns emerge and support behaviour. Yet the brain-wide presynaptic wiring rules that lay the foundation for the functional selectivity of individual neurons remain largely unexplored. Cortical neurons, even in primary sensory cortex, are heterogeneous in their selectivity, not only to sensory stimuli but also to multiple aspects of behaviour. Here, to investigate presynaptic connectivity rules underlying the selectivity of pyramidal neurons to behavioural state[1–10] in primary somatosensory cortex (S1), we used two-photon calcium imaging, neuropharmacology, single-cell-based monosynaptic input tracing and optogenetics. We show that behavioural state-dependent activity patterns are stable over time. These are minimally affected by direct neuromodulatory inputs and are driven primarily by glutamatergic inputs. Analysis of brain-wide presynaptic networks of individual neurons with distinct behavioural state-dependent activity profiles revealed that although behavioural state-related and behavioural state-unrelated neurons shared a similar pattern of local inputs within S1, their long-range glutamatergic inputs differed. Individual cortical neurons, irrespective of their functional properties, received converging inputs from the main S1-projecting areas. Yet neurons that tracked behavioural state received a smaller proportion of motor cortical inputs and a larger proportion of thalamic inputs. Optogenetic suppression of thalamic inputs reduced behavioural state-dependent activity in S1, but this activity was not externally driven. Our results reveal distinct long-range glutamatergic inputs as a substrate for preconfigured network dynamics associated with behavioural state.

Anatomical connectivity within and between brain areas governs the distinct activity patterns of individual neurons. In sensory cortical areas, the properties of local presynaptic connections, including their number, strength and spatial arrangement, shape the selectivity of individual neurons to sensory stimuli[11–18]. Cortical neurons are, however, functionally highly heterogeneous in relation to various aspects of behavioural contexts and states, such as unexpected events, rewards, attentional demands or spontaneous movements[1–10,19,20]. Even in sensory cortical areas, neurons show highly dynamic activity in the absence of external sensory stimuli[1,6,9]. These behavioural contextual and state signals are thought to be conveyed by long-range projections, including neuromodulatory and glutamatergic afferents from multiple brain areas[21–28].

One way to establish functional heterogeneity is that a specific set of inputs projects to a subset of neurons, whether the inputs are neuromodulatory or long-range and local glutamatergic and GABAergic (γ-aminobutyric acid-expressing), or all of these (Fig. 1a). Alternatively, presynaptic inputs may be random and highly plastic (Fig. 1a). Although the mesoscale level of connectivity within and between brain areas has been mapped[29], linking connectivity rules with the functional identity of individual neurons remains challenging. Here, we investigated the architecture of brain-wide presynaptic connectivity in the context of spontaneous movement-sensitive neurons in S1. Spontaneous movements, including whisking and locomotion, along with cortical synchronization and pupil dilation, are components of innate behaviour that reflect a behavioural state. These spontaneous movements are robustly represented in a subset of neurons in a wide range of brain areas[8,9]. However, how the functional specificity of these neurons arises is not clear.

## Cortical activity during spontaneous movements

We first characterized the neural representations of spontaneous movements in the whisker primary somatosensory cortex (wS1) of mice. We used two-photon Ca²⁺ imaging to monitor the activity of pyramidal neurons (PNs) of right hemisphere wS1 layer (L) 2/3 (Fig. 1b). We induced the expression of the Ca²⁺ sensor GCaMP6 in PNs by either injecting a virus carrying GCaMP6f in the wS1 of Emx1-IRES-Cre mice or crossing the CaMK2a-tTA and tetO-GCaMP6s transgenic mouse lines.

[1]Unit on Functional Neural Circuits, Systems Neurodevelopment Laboratory, National Institute of Mental Health, National Institutes of Health, Bethesda, MD, USA. [2]Machine Learning Core, National Institute of Mental Health, National Institutes of Health, Bethesda, MD, USA. [3]Section on Neuroanatomy, National Institute of Mental Health, National Institutes of Health, Bethesda, MD, USA. [4]These authors contributed equally: Ka Chun Lam, Yuan Zhao. ✉e-mail: ana.inacio@nih.gov; soohyun.lee@nih.gov

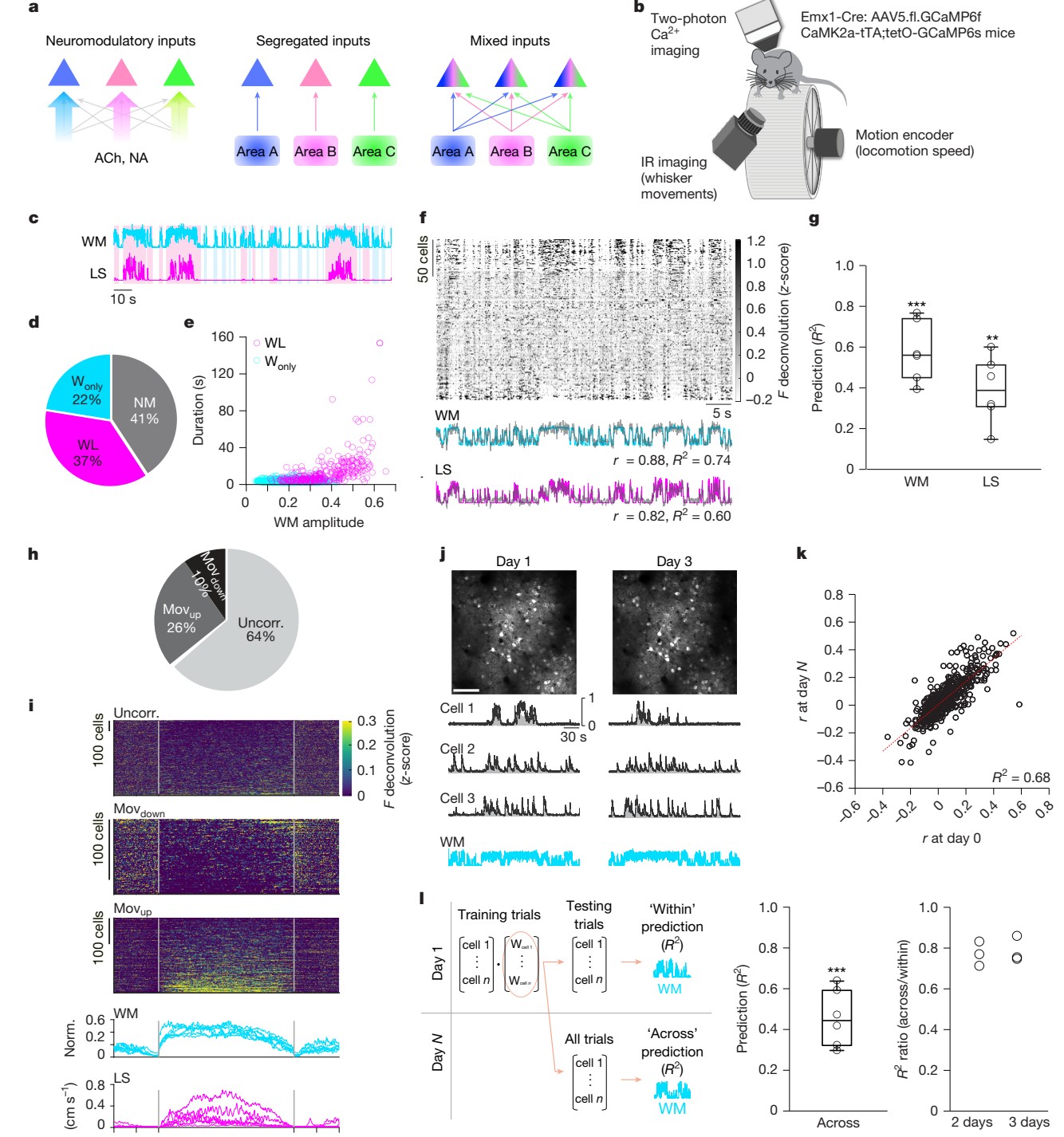

**Fig. 1 | Heterogeneity and stability of neuronal activity patterns in relation to spontaneous movements. a**, Models of input connectivity for functionally distinct PNs (triangles of different colours). NA, noradrenaline. **b**, Experimental paradigm. IR, infrared. **c**, Example time course of whisker movements (WM) and locomotion speed (LS). Cyan-shaded, $W_{only}$; magenta-shaded, WL. **d**, Fraction of time spent per behavioural event type ($n = 5$ sessions, 5 mice). NM, no movement. **e**, Whisker movement duration versus amplitude for $W_{only}$ ($n = 1,268$) and WL ($n = 321$). **f,g**, Prediction of whisker movements and locomotion speed from population activity. $R^2$, variance explained. **f**, Example session. Top, raster plot of neuronal activity; neurons ($n = 201$) are sorted from top to bottom by decreasing weight on the first principal component. Bottom, whisker movements (cyan) and locomotion speed (magenta) overlaid with predicted (grey) traces ($P < 0.0001$ for both; $r$, correlation). **g**, Independent sessions ($n = 6$ FOVs, 5 mice; WM, $P = 0.00011$; LS, $P = 0.0010$; one-sided paired-sample $t$-test).

**h**, Fraction of movement-uncorrelated ($n = 715$), $Mov_{down}$ ($n = 123$) and $Mov_{up}$ ($n = 286$) neurons. **i**, Peri-event time histogram (PETH) of neuronal activity ($n$ as in **h**). WL events are time-normalized from onset to offset (vertical bars). Norm., normalized. **j**, Example imaging FOV (top) and activity of example $Mov_{up}$ neurons during spontaneous movements at days 1 and 3 (bottom). Black, $F$; grey, deconvolved $F$; normalized to maximum. Scale bar, 100 μm. **k**, Correlation ($r$) between the activity of individual neurons and whisker movements at days 3–4 versus day 1 ($n = 682$, 6 FOVs, 5 mice; $P < 0.0001$, regression). **l**, Prediction of whisker movements across days. Left, decoder scheme. Middle, variance explained across days ($P < 0.0001$, one-sided paired-sample $t$-test). Right, out-of-sample $R^2$ ratio for day 3 versus day 1 ($n = 3$ FOVs, 3 mice) and day 4 versus day 1 ($n = 3$ FOVs, 2 mice). **g,l**, In box plots, the central line and box represent the median and 25th–75th percentiles, and whiskers extend to the most extreme data points excluding outliers (larger than 1.5× the interquartile range).

Mice were head-fixed on top of a wheel in darkness. A near-infrared camera was used to capture whisker movements (WM) (Fig. 1b). A wheel speed encoder was used to measure locomotion speed (LS) (Fig. 1b). Mice were not instructed to move, nor were they rewarded. We observed two types of spontaneous movements throughout the recording session. One type consisted of short duration and small amplitude whisker movements without locomotion ($W_{only}$). The other type consisted of longer and larger amplitude whisker movements accompanied by locomotion (WL) (Fig. 1c–e). To assess the relationship between neuronal activity and spontaneous movements, we trained two separate linear decoders to predict whisker movements or locomotion speed from population activity. Population activity exhibited fluctuations coupled to the presence or absence of spontaneous movements and reliably predicted both whisker movements and locomotion speed in out-of-sample data (Fig. 1f,g). Since the fraction of variance explained by neuronal activity was higher for whisker movements than for locomotion speed (Fig. 1g), we based our population analysis on the whisker movement prediction[7]. Embedded in the population, individual neurons exhibited heterogenous patterns of activity in relation to spontaneous movements (Fig. 1f and Extended Data Fig. 1). Two subsets of neurons exhibited activity patterns time-locked to spontaneous movements. The first showed increased activity ($Mov_{up}$, $26 \pm 6.7\%$ of all neurons) and the second, decreased activity during spontaneous movements ($Mov_{down}$, $10 \pm 6.3\%$ of all neurons) (Fig. 1h,i). Most $Mov_{up}$ neurons ($75 \pm 19\%$ of all neurons) increased their activity during WL bouts and are hereafter referred to as movement-correlated (corr.) (Fig. 1i and Extended Data Fig. 1). These results are consistent with the interpretation that wS1 is strongly engaged during locomotion[7]. All other neurons did not change their activity with respect to spontaneous movements, and are referred to as movement-uncorrelated (uncorr.).

To understand how the spontaneous movement-correlated and sensory stimulus-responsive neuronal populations are represented in wS1, we also recorded sensory stimulus-evoked neuronal activity. Sensory stimulation consisted of a periodic deflection of the left whiskers, using a vertically oriented pole. We found that $10 \pm 5.7\%$ of all neurons responded positively to passive whisker deflection[19] (Extended Data Fig. 2). A fraction of movement-uncorrelated (9.4%), $Mov_{up}$ (14%) and $Mov_{down}$ (7.9%) neurons responded to sensory stimulation (Extended Data Fig. 2), as observed previously in the visual cortex[9] and wS1[7,19]. Together, these results show that a subset of wS1 PNs reliably encodes spontaneous movements and suggest that the coding of spontaneous movements and sensory stimuli may be independent given that some neurons have either one of these or both properties.

We next tested whether the representation of spontaneous movements in wS1 PNs is a stable feature. Although both stability and plasticity of sensory coding schemes[30] and spontaneous ensemble activity[31] in primary sensory cortices have been reported, the stability of spontaneous movement-dependent patterns of activity has not been directly tested. When tracking the same population of neurons over multiple days, we observed a highly stable correlation between neuronal activity patterns and spontaneous movements (Fig. 1j), both at single-neuron and population levels. The correlation ($r$) between the activity of individual neurons and whisker movements was highly reliable across days (Fig. 1k). To evaluate the stability of population activity in relation to spontaneous movements, we built a linear decoder using data from the first imaging session (day 1, within; Fig. 1l) and applied this decoder to out-of-sample data collected 2–3 days later (day $N$, across days). The day 1 model consistently predicted spontaneous movements across days (Fig. 1l), suggesting a stable representation in the population. Our results demonstrate that wS1 PNs reliably represent spontaneous movements over multiple days.

## Suppression of direct neuromodulatory inputs

We next investigated the source of the stable encoding of spontaneous movements by wS1 PNs. Neuromodulators, including neuropeptides,

exert powerful effects over cortical function[26,32]. Acetylcholine (ACh) and noradrenaline terminal activity in sensory cortices, for example, is highly correlated to spontaneous movements[25,27,33–36]. These neuromodulators have been tightly linked to the locomotion-related gain modulation of sensory responses in PNs[24,37,38]. However, a direct effect of ACh and noradrenaline on the activity of large populations of L2/3 PNs during spontaneous movements remains to be established. To that end, we implanted a custom-made cranial window with an access port that allowed local application of ACh, noradrenaline or glutamate receptor blockers while imaging wS1 L2/3 PNs (Fig. 2a). We blocked a different type of receptor or no receptor at all (in sham controls) per imaging session, and each session was performed on a different day. In a final session, tetrodotoxin (TTX), a $Na^+$ channel blocker, was applied to evaluate the effectiveness of drug diffusion per field of view (FOV). TTX silenced nearly all neuronal activity over the entire FOV within 20 min (Extended Data Fig. 3). To further test whether ACh and noradrenaline receptor blockers reach all neurons in the FOV, we expressed genetically engineered ACh ($GRAB_{ACh}$) or noradrenaline ($GRAB_{NE}$) sensors in wS1 L2/3 neurons using viral vectors[34,36]. We observed increases in both ACh and noradrenaline receptor binding during spontaneous movements[25,33–36], which were abolished throughout the FOV upon application of the respective receptor blockers, demonstrating the effectiveness of the blocker application approach (Extended Data Fig. 4).

The temporal correlation between the activity of individual neurons and spontaneous movements was largely preserved during blockade of ACh or noradrenaline receptors (Fig. 2b,c,e,f), suggesting that individual neurons maintain their activity profile during spontaneous movements. However, application of an NMDA (*N*-methyl-D-aspartate) receptor blocker diminished the correlation between individual neuron activity and spontaneous movements, revealing a strong NMDA component of spontaneous movement-dependent activity (Fig. 2d,g). The combined action of blockers against the NMDA receptor and AMPA (α-amino-3-hydroxy-5-methyl-4-isoxazole propionic acid) receptor (both glutamate (Glu) receptors) further uncovered the contribution of glutamatergic signalling through AMPA receptors for ongoing activity (Fig. 2d,h and Extended Data Fig. 5). Whereas ACh and noradrenaline receptor blockers did not disrupt the correlation of PNs with spontaneous movements, both blockers modulated the magnitude of movement-dependent activity in a fraction of movement-correlated neurons. The effects of the blockers were greater on evoked activity in stimulus-responsive neurons (Extended Data Fig. 5).

We examined the effect of neuromodulatory and glutamatergic inputs on population activity. For each recording session, we built a linear decoder using data recorded prior to the application of the receptor blockers or Ringer's reapplication in sham controls (within), then tested this decoder using the out-of-sample data acquired after blocker application (across conditions). The model consistently predicted spontaneous movements from population activity across conditions when either ACh or noradrenaline receptor blockers were applied, indicating that the structure of movement-dependent population activity is largely maintained during neuromodulator receptor blockade (Ringer's sham controls, $R^2 = 0.54 \pm 0.085$; ACh receptor, $R^2 = 0.37 \pm 0.084$; noradrenaline receptor, $R^2 = 0.52 \pm 0.15$; $P < 0.0001$ for all conditions, paired-sample $t$-test, across $R^2 \leq 0$ versus $R^2 > 0$). By contrast, the decoder did not predict spontaneous movements from population activity when either NMDA or Glu receptor blockers were tested (NMDA receptor, $R^2 = 0.025 \pm 0.084$, $P = 0.26$; Glu receptor, $R^2 = -11.32 \pm 15$, $P = 0.95$). The out-of-sample $R^2$ ratio ($R^2_{across}/R^2_{within}$) for ACh or noradrenaline receptor blockade did not differ from that of the sham controls, but it was significantly lower for NMDA and Glu receptor blockade (Fig. 2i). Local application of the different blockers did not affect spontaneous movement behaviour (Extended Data Fig. 6). These results provide evidence for a limited role of direct

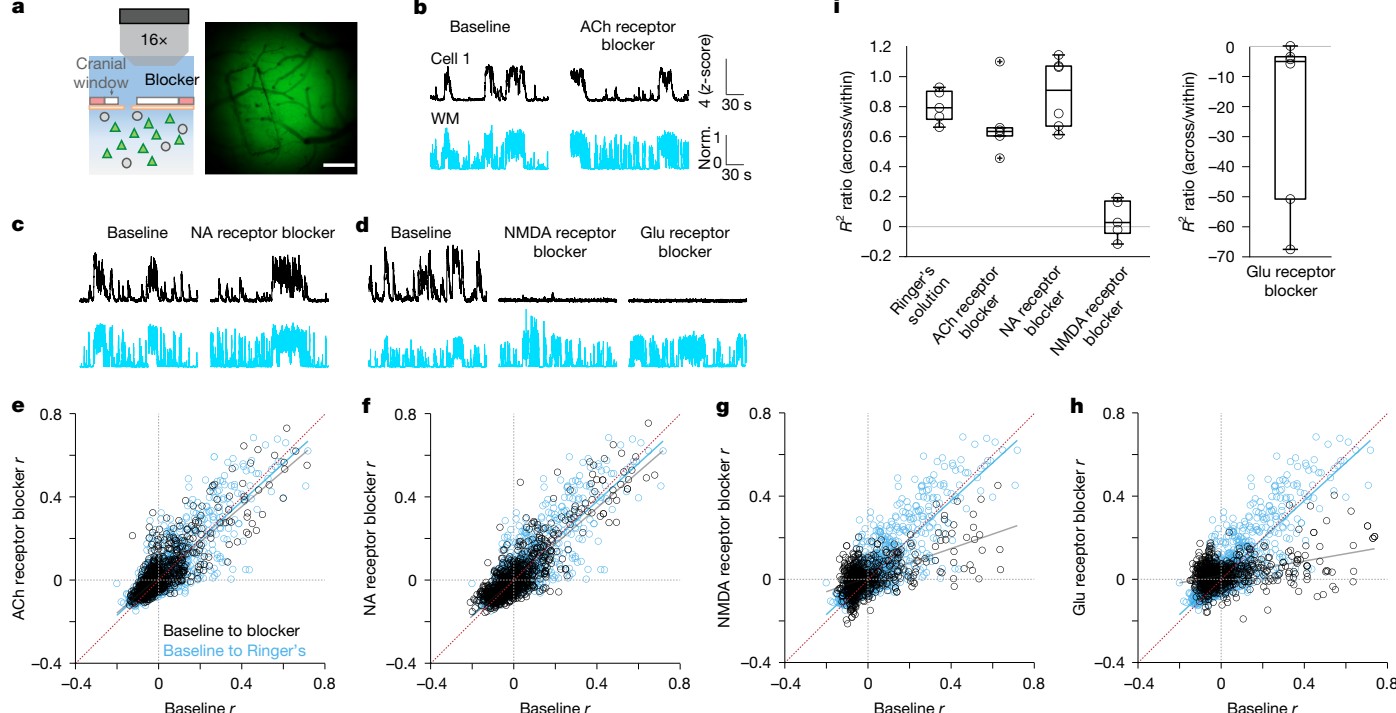

**Fig. 2 | Limited role of direct neuromodulatory inputs for spontaneous movement-dependent neuronal activity in wS1. a**, Experimental approach for local delivery of receptor blockers during two-photon Ca²⁺ imaging. Triangles, PNs (GCaMP6⁺); circles, GABAergic neurons. Scale bar, 0.3 mm. **b–d**, Activity of an example Mov_up neuron during spontaneous movements before and after blockade of ACh (atropine and mecamylamine; **b**), noradrenaline (prazosin and propranolol; **c**), NMDA (D-2-amino-5-phosphonovaleric acid (D-AP5); **d**), or NMDA and AMPA (Glu, D-AP5 and 6,7-dinitroquinoxaline-2,3-dione (DNQX); **d**) receptors. Black, fluorescence traces; cyan, whisker movements. **e–h**, Correlation (*r*) between the activity of individual neurons and whisker movements during ACh (*n* = 1,300, 6 FOVs, 6 mice; **e**), noradrenaline (*n* = 1,483, 6 FOVs, 6 mice; **f**), NMDA (*n* = 720, 5 FOVs, 5 mice; **g**) or Glu (*n* = 956, 6 FOVs; 6 mice, **h**) receptor blockade versus baseline, respectively; sham sessions

(Ringer's only versus baseline) are included in **e–h** (cyan, *n* = 1,597, 5 FOVs, 3 mice) (null hypothesis, equal slopes: ACh receptor versus sham, *P* = 0.033; noradrenaline receptor versus sham, *P* = 0.13; NMDA receptor versus sham, *P* < 0.0001; Glu receptor versus sham, *P* < 0.0001; multiple comparisons with Bonferroni correction). **i**, Prediction of whisker movements from population activity. Out-of-sample *R*² ratio (ACh receptor versus sham, *P* = 0.33; noradrenaline receptor versus sham, *P* > 0.05; NMDA receptor versus sham, *P* = 0.032; Glu receptor versus sham, *P* = 0.017; NMDA receptor versus Glu receptor, *P* < 0.0001; Kruskal–Wallis test (*P* = 0.00022) followed by two-sided Wilcoxon rank-sum tests with Bonferroni correction; *n* as in **e–h**). In box plots, the central line and box represent the median and 25th–75th percentiles, and whiskers extend to the most extreme data points excluding outliers (larger than 1.5× the interquartile range).

neuromodulatory (ACh and noradrenaline) inputs in generating highly stable patterns of spontaneous movement-dependent activity in wS1.

## Brain-wide monosynaptic input tracing

On the basis of the highly stable, glutamate receptor-sensitive activity during spontaneous movements, we hypothesized that a distinct anatomical organization of presynaptic networks may constrain the spontaneous movement-dependent activity profiles of PNs. To reveal the presynaptic network of functionally identified neurons at the single-cell level, we adapted a single-neuron-based monosynaptic retrograde tracing approach[14,17,39] (Fig. 3a). We first imaged wS1 L2/3 PNs and selected a target neuron on the basis of its activity profile during spontaneous movements (example movement-correlated neuron, Fig. 3a,b). We then performed two-photon guided electroporation of the target neuron with DNA encoding TVA (rabies virus receptor), G (rabies virus spike protein) and mCherry (for validation of transfection) (Fig. 3a and Extended Data Fig. 7). Following electroporation, we injected a rabies virus variant carrying a red fluorescent protein (RFP (mCherry), *n* = 2 brains; tdTomato (tdTom), *n* = 20 brains) close to the electroporated neuron. In line with previous studies[14,17], at day 4–5 we observed the emergence of RFP⁺ presynaptic neurons in wS1. We detected the RFP⁺ neurons exclusively in brains in which the electroporated neuron

survived for several days and expressed mCherry (postsynaptic neuron), confirming the specificity of our approach (Extended Data Fig. 8). To reveal presynaptic neurons, mice were perfused at day 11 (±1.5 days) and whole brains were analysed histologically. Presynaptic neurons were manually annotated according to anatomical area and identified as glutamatergic or GABAergic on the basis of immunostaining with GABA (γ-aminobutyric acid) (Methods, 'Whole-brain reconstruction, annotation and registration'). Brains were registered to the Allen Mouse Common Coordinate Framework and annotations were validated (Fig. 3c).

We performed single-cell-based retrograde tracing of two functionally distinct subsets of L2/3 PNs: movement-uncorrelated (*n* = 11, 11 mice) and movement-correlated (*n* = 11, 11 mice) neurons (Fig. 3b,d). To refine the functional specificity and eliminate the confounding factor of somatosensory input in the two groups, we targeted neurons that did not respond to sensory stimulation. The spontaneous movements and proportion of spontaneous movement-dependent neuronal subsets were similar between the groups. Additionally, the cortical depth of the postsynaptic neurons were also similar (Extended Data Fig. 9).

To analyse the brain-wide arrangement of presynaptic connectivity, we compared the fraction of presynaptic cells from each brain area between the two groups. Variability in the total number of presynaptic neurons across brains is expected owing to differences in the survival

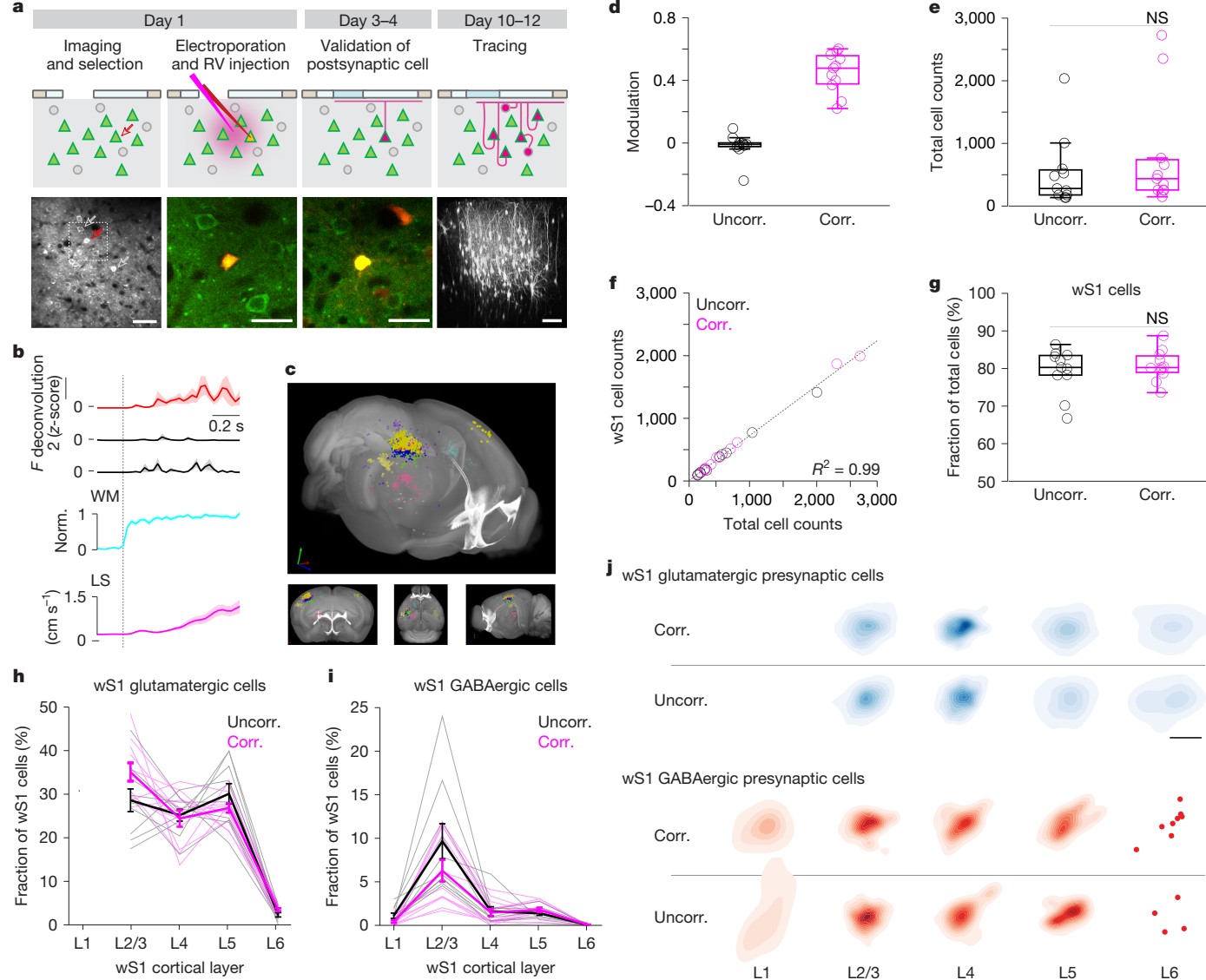

**Fig. 3 | Functionally distinct neurons have anatomically similar local presynaptic networks. a**, Brain-wide monosynaptic input tracing. Top, schematics. Bottom, example two-photon images (*n* = 22 mice). Left to right: FOV (red arrows, postsynaptic PNs; white arrows, surrounding PNs; scale bar, 50 μm); postsynaptic PN immediately after electroporation (Alexa 594⁺; scale bar, 25 μm); postsynaptic PN 4 days after electroporation (mCherry⁺; scale bar, 25 μm); wS1 presynaptic neurons (RFP⁺; 3D projection of an 814 × 814 × 785 μm z-stack; scale bar, 100 μm). RV, rabies virus. **b**, Activity of the example postsynaptic (red) and surrounding (black) PNs aligned to WL onset (mean ± s.e.m.). **c**, Corresponding presynaptic network (individual symbols represent individual presynaptic neurons; colours represent anatomical regions). **d**, Mean modulation of postsynaptic neurons during WL. **e**, Number of presynaptic neurons per brain (*P* = 0.39, two-sided randomization test). **f**, wS1 versus total presynaptic neurons (*P* < 0.0001, regression). **g**, Fraction of presynaptic neurons in wS1 (*P* = 0.52, two-sided randomization test). **h**, Laminar distribution of wS1 glutamatergic

presynaptic neurons (layers: L6 versus all other layers, *P* < 0.01, Kruskal–Wallis test (*P* < 0.0001) followed by Dunn's tests with Bonferroni correction; uncorr. versus corr.: $P_{L2/3}$ = 0.065, $P_{L4}$ = 0.74, $P_{L5}$ = 0.20, $P_{L6}$ = 0.12, two-sided randomization tests). **i**, Laminar distribution of wS1 GABAergic presynaptic neurons (layers: L2/3 versus all other layers, *P* < 0.01, Kruskal–Wallis test (*P* < 0.0001) followed by Dunn's tests with Bonferroni correction; uncorr. versus corr.: $P_{L2/3}$ = 0.16; $P_{L4}$ = 0.79; $P_{L5}$ = 0.28, two-sided randomization tests). **j**, Horizontal, laminar projections of the weighted distributions of wS1 glutamatergic or GABAergic presynaptic neurons (uncorr. versus corr.: glutamatergic, $P_{L2/3}$ = 0.73, $P_{L4}$ = 0.39, $P_{L5}$ = 0.066, $P_{L6}$ = 0.26; GABAergic, $P_{L2/3}$ = 0.86; $P_{L4}$ = 0.84; $P_{L5}$ = 0.31; two-sided randomization tests). Scale bar, 500 μm. **d**–**g**, Postsynaptic neurons ($n_{\text{uncorr.}}$ = 11; $n_{\text{corr.}}$ = 11). In box plots, the central line and box represent the median and 25th–75th percentiles, and whiskers extend to the most extreme data points excluding outliers (larger than 1.5× the interquartile range). **h**–**j**, Postsynaptic neurons ($n_{\text{uncorr.}}$ = 10; $n_{\text{corr.}}$ = 11).

time of each postsynaptic neuron[40]. Nevertheless, the distribution of the total number of presynaptic neurons per brain was similar for the movement-uncorrelated (range 137–2,038) and movement-correlated (range 148–2,727) groups (Fig. 3e). We observed a strong linear association between the number of local versus total presynaptic neurons per brain in both groups (Fig. 3f and Extended Data Fig. 10). On the basis of the linear association, we concluded that comparing fractional presynaptic inputs onto postsynaptic neurons of the two groups is appropriate.

## Presynaptic networks in wS1

We first investigated whether spontaneous movement-correlated neurons receive inputs from distinctive local, wS1 presynaptic networks. Presynaptic neurons located in wS1 constituted the largest fraction of brain-wide inputs to each postsynaptic neuron: 79 ± 6.0% for the movement-uncorrelated group and 81 ± 6.0% for the movement-correlated group (Fig. 3g and Extended Data Fig. 10). Of all wS1 presynaptic neurons, the majority were glutamatergic:

86 ± 7.4% for the movement-uncorrelated group and 90 ± 5.5% for the movement-correlated group (Extended Data Fig. 10). Glutamatergic presynaptic neurons were, on average, broadly distributed across L2/3 to L5 but significantly less predominant in L6 for both groups (Fig. 3h). Individual glutamatergic presynaptic networks were often characterized by a smaller fraction of L4 compared with L2/3 or L5 neurons[41,42]. The predominance of this laminar profile was similar between the groups ($P = 0.95$, two-sided randomization test with Chow test). Additionally, the fraction of glutamatergic inputs from each layer was similar between the groups. In contrast to the broad vertical distribution of glutamatergic inputs, GABAergic presynaptic neurons were mostly limited to L2/3[17] (Fig. 3i). Yet, the fraction of GABAergic inputs per layer did not differ between the two groups. These results demonstrate that the functionally distinct neurons have a similar proportion of local presynaptic neurons across layers.

Next, we explored the spatial distributions of wS1 presynaptic networks. The three-dimensional and two-dimensional (layer-by-layer horizontal flat projections) spatial distributions of wS1 glutamatergic and GABAergic presynaptic networks were not significantly different between the groups (Fig. 3j). To further characterize the horizontal spread of wS1 glutamatergic presynaptic networks across layers, we performed gaussian density estimation on the layer projections (Extended Data Fig. 11). Glutamatergic presynaptic neurons in L4 were restricted to a smaller cortical span (cell pairwise distance mean approximately 309 μm) than in L2/3 or L5 (mean approximately 417 μm and 505 μm, respectively), consistent with the notion that L2/3 neurons receive inputs from mainly one barrel[41]. We observed this feature in all 22 presynaptic networks of both groups. GABAergic presynaptic neurons were more spatially confined (L2/3 cell pairwise distance mean, 272 ± 71 μm) than glutamatergic presynaptic neurons[17] ($P = 2.5 \times 10^{-9}$; two-sided $t$-test) (Fig. 3j). In summary, the glutamatergic and GABAergic presynaptic networks of single L2/3 PNs in wS1 exhibit anatomical features found in population studies[41,42]. Yet, despite their markedly different activity patterns, functionally distinct sets of neurons (movement-uncorrelated and movement-correlated) have a similar pattern of local glutamatergic and GABAergic inputs, in terms of both number and spatial distribution across all cortical layers.

## Brain-wide presynaptic networks

We next investigated whether movement-correlated neurons receive a distinct set of brain-wide, long-range inputs (Fig. 1a). Recent studies have provided insights into the organization of local inputs onto individual PNs in primary visual cortex in the context of visual stimuli[16,19]. Nevertheless, how the heterogenous activity patterns of L2/3 neurons relate to long-range inputs has not been explored. Analysis of the single-cell-based, whole-brain presynaptic networks revealed the distribution of presynaptic neurons in multiple cortical and subcortical brain areas that are known to project to wS1[42,43]: secondary somatosensory cortex, primary and secondary motor cortices (M1/2), other sensory cortical areas (including auditory visual cortices), thalamus, contralateral wS1, perirhinal cortex and basal forebrain (Fig. 4a). Brain areas that contain, on average, less than 0.5% of total presynaptic cells were grouped as 'others'. All long-range presynaptic neurons were glutamatergic. Most input areas were ipsilateral, with a few exceptions, such as the perirhinal cortex. Irrespective of its activity pattern, each postsynaptic neuron from both the movement-uncorrelated and movement-correlated groups received inputs from, on average, 6.5 of the 7 major brain areas known to project to wS1 (Fig. 4a,b). These results suggest that a high degree of integration or multiplexing can occur at the single-cell level. Although all postsynaptic cells from both groups received direct inputs from a wide range wS1-projecting brain areas, we found that, surprisingly, movement-correlated neurons receive a lower fraction of inputs from M1/2 than movement-uncorrelated neurons (Fig. 4a,c). By contrast, movement-correlated neurons received

a significantly higher fraction of inputs from thalamus relative to movement-uncorrelated neurons (Fig. 4a,d). Thalamic presynaptic neurons were almost equally distributed in the first-order thalamic relay nucleus, ventral posteromedial nucleus (VPm) and the higher-order thalamic nucleus, posteromedial nucleus (POm) for both groups (Extended Data Fig. 12). Similarly, the fraction and spatial distribution of M1 and M2 cells in motor cortical presynaptic networks were similar between the groups (Extended Data Fig. 12). Furthermore, the modulation of the activity of postsynaptic neurons across spontaneous movements (that is, change in activity during movement relative to baseline, prior to movement onset; Methods, 'Modulation of neuronal activity') was negatively correlated with the fraction of presynaptic neurons found in M1/2 (Fig. 4e) and positively correlated with the fraction of presynaptic neurons found in thalamus (Fig. 4f). Within each long-range input area, the spatial distribution of presynaptic neurons was similar for the two groups (Extended Data Fig. 13), suggesting that the functionally distinct postsynaptic neurons receive inputs from spatially intermingled long-range neurons. Overall, these results demonstrate that, despite the high degree of convergence of inputs from multiple brain areas to single neurons in wS1, movement-correlated wS1 PNs show characteristic brain-wide presynaptic inputs, with a relatively larger fraction of thalamic inputs, but a smaller fraction of motor cortical inputs.

Given that movement-correlated neurons receive more abundant inputs from the thalamus, we next explored whether the spontaneous movement-dependent activity in these neurons is caused by sensory feedback from voluntary whisker movements (Fig. 4g). We imaged right hemisphere wS1 L2/3 PNs before and after paralysis of the left facial muscles by injecting botulinum toxin (BTX) into the left mystacial pad (Fig. 4h). The same population of PNs was re-imaged 2–3 days after BTX injection. This procedure was minimally invasive but effectively abolished left whisker movements (Fig. 4i) without interrupting right whisker movements and locomotion (Extended Data Fig. 14). Since left and right whisker movements were highly correlated under head-fixed conditions ($r$ (left versus right) = 0.959 ± 0.0165, $n$ = 4 sessions, 4 mice), we used the right whisker movements to detect changes in movement-dependent activity after paralysis (Fig. 4i). We observed that the correlation between the activity of individual neurons and spontaneous movements was highly preserved even after paralysis (Fig. 4j). Beyond correlation, we found that the modulation of individual neurons during spontaneous movements was similar before and after paralysis (Extended Data Fig. 14). To examine changes in the relationship between population activity and spontaneous movements, we built a linear decoder using data collected prior to paralysis (within) and evaluated it on out-of-sample data recorded after paralysis (across days). Model predictions were highly reliable across days ($R^2 = 0.44 ± 0.092$, $P < 0.0001$, paired-sample $t$-test test, across $R^2 \leq 0$ versus $R^2 > 0$). In addition, the out-of-sample $R^2$ ratio ($R^2_{across}/R^2_{within}$) was similar when recordings were performed across days without (Fig. 1, chronic) and with unilateral whisker paralysis (Fig. 4k). To further eliminate a potential contribution of sensory feedback from the ipsilateral mystacial pad, we monitored neuronal activity in wS1 following bilateral whisker trimming, as well as bilateral whisker trimming combined with bilateral mystacial pad paralysis. We found that locomotion-dependent activity is largely preserved even after bilateral whisker trimming together with mystacial pad paralysis (Extended Data Fig. 15). These results revealed that the individual neuron and structure of population activity in relation to spontaneous movements are stable after facial paralysis and whisker trimming, suggesting that sensory feedback evoked by whisker movements does not have an instrumental role in driving spontaneous movement-dependent activity in wS1.

## Suppression of thalamic and motor inputs

Finally, we investigated whether the anatomical signature of predominant thalamic inputs to movement-correlated neurons contributes to

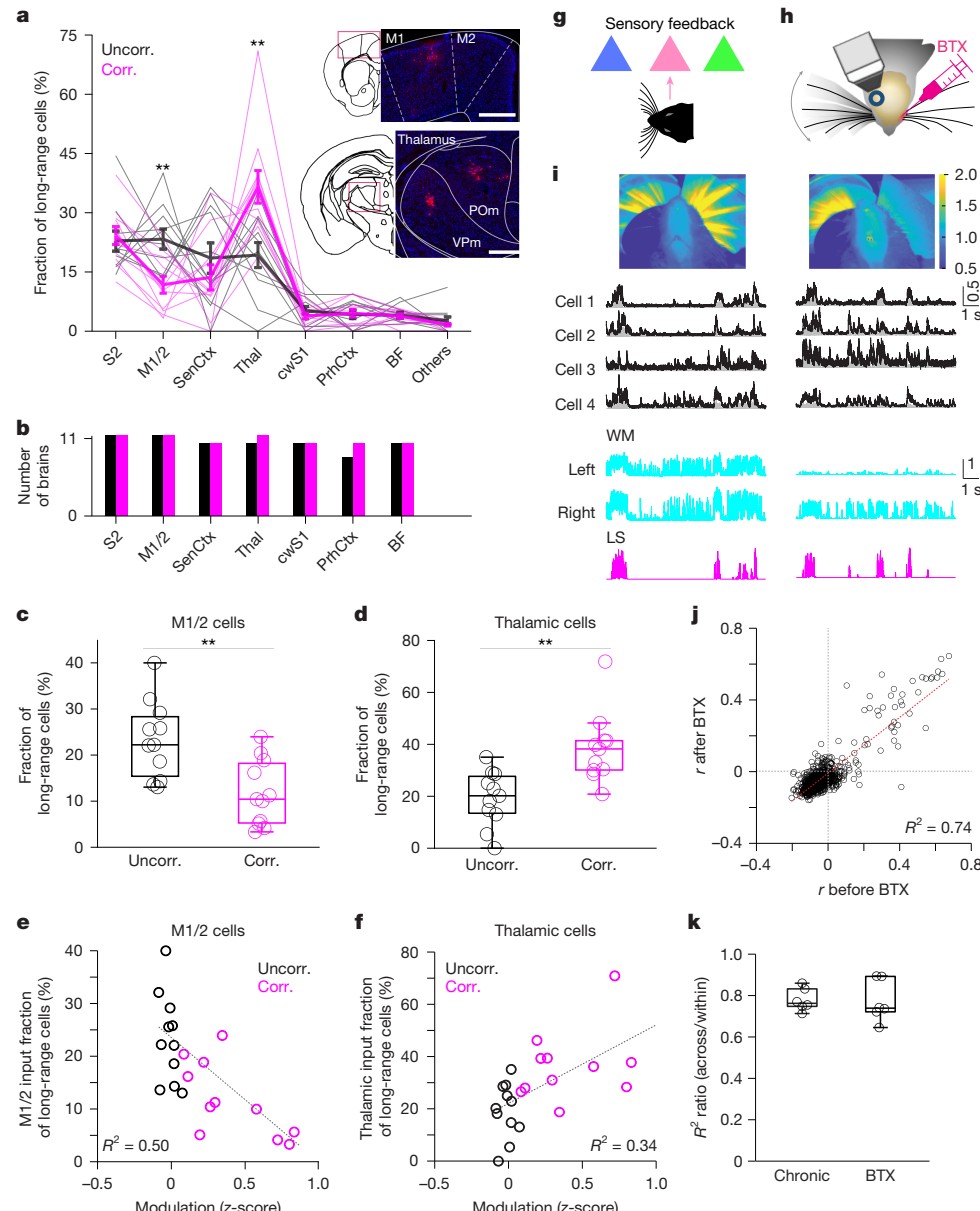

**Fig. 4 | Functionally distinct neurons receive characteristic long-range inputs. a**, Fraction of presynaptic neurons across brain areas (corr. versus uncorr.: M1/2, $P = 0.0030$; thalamus, $P = 0.0030$; all other areas, $P > 0.05$, two-sided randomization tests). Thin lines represent individual brains; thick lines are mean ± s.e.m. Inset, example epifluorescence images of coronal brain slices showing presynaptic neurons (RFP[+]) in M1/2 and thalamus. Scale bars, 500 µm. BF, basal forebrain; cwS1, contralateral wS1; PrhCtx, perirhinal cortex; S2, secondary somatosensory cortex; SenCtx, sensory cortical areas other than wS1 and S2, including auditory and visual cortices; thal, thalamus. **b**, Number of presynaptic networks exhibiting neurons in listed brain areas. Black, uncorr. group; magenta, corr. group. **c,d**, Fraction of presynaptic neurons in M1/2 (**c**) and thalamus (**d**). **e,f**, Motor cortical (**e**) and thalamic (**f**) input fraction as function of the average modulation of the postsynaptic neuron across spontaneous movements ($W_{only} + WL$) (M1/2, $P = 0.0024$; thalamus, $P = 0.0043$; regression). **g**, Sensory feedback from whisker movements as hypothetical source of movement-dependent activity. **h,i**, Two-photon Ca[2+] imaging before (left) and 1–2 days after (right) unilateral BTX injection in the mystacial pad. **i**, Top, example images of video-recorded whisker movements (mean of the absolute difference between consecutive frames over approximately 2 min) before and after BTX injection. Warmer colours reflect greater motion ($n = 5$ mice). Bottom, activity of example $Mov_{up}$ neurons during spontaneous movements. Black, fluorescence trace; grey, deconvolved fluorescence trace; normalized to maximum. **j**, Correlation ($r$) between the activity of individual neurons and whisker movements before versus after BTX injection ($n = 1,044$, 6 FOVs, 5 mice, $P < 0.0001$, regression). **k**, Prediction of whisker movements from population activity. Out-of-sample $R^2$ ratio for chronic (Fig. 1) versus BTX experiments ($n = 6$ FOVs, 5 mice; $P = 0.70$, two-sided Wilcoxon rank-sum test). In box plots, the central line and box represent the median and 25th–75th percentiles, and whiskers extend to the most extreme data points excluding outliers (larger than 1.5× the interquartile range).

their activity during spontaneous movements. We imaged PNs while simultaneously suppressing thalamic terminals in wS1 L2/3 (Fig. 5a). To that end, we expressed the red-shifted inhibitory opsin archaerhodopsin ArchT coupled to tdTomato in the thalamus (VPm and POm) or M1/2 of CaMK2a-tTA;tetO-GCaMP6s mice (Fig. 5a,b). Mice that did not express ArchT served as controls. Light pulses (1–1.5 s) were randomly

provided during the recording session. We observed that light pulses elicited a relatively brief (around 0.5 s) whisker movement in both control and ArchT-expressing mice (Extended Data Fig. 16). This whisker movement was independent of the ongoing movements of the animal and could potentially obfuscate neuronal recordings. To isolate changes in neuronal activity caused by thalamic terminal suppression

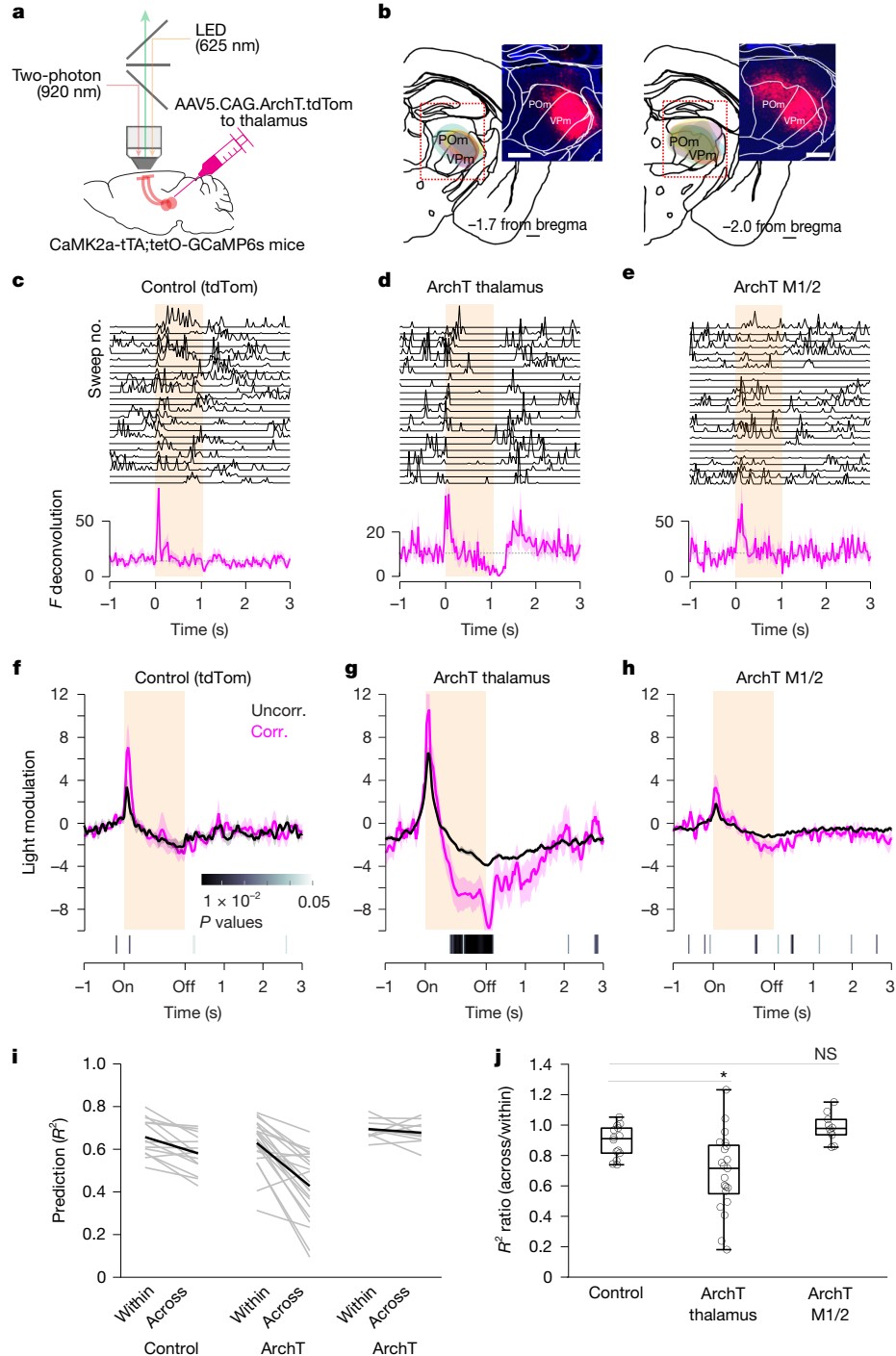

**Fig. 5 | Effect of optogenetic suppression of thalamic and motor cortical inputs on spontaneous movement-dependent activity. a**, Simultaneous two-photon Ca²⁺ imaging and optogenetic suppression of thalamic and motor cortical axon terminals (ArchT⁺tdTom⁺) in wS1. **b**, Areas encompassing ArchT⁺tdTom⁺ cell bodies in mice expressing ArchT in thalamus, overlaid on corresponding mouse atlas images. Different colours represent individual mice (*n* = 7). Insets, example composite epifluorescence images of coronal brain sections denoting tdTom⁺ cell bodies circumscribed to the VPm and POm in one mouse. Scale bars, 0.5 mm. **c**–**e**, Effect of light pulses on the activity of movement-correlated neurons. Example neurons from a control mouse (ArchT⁻tdTom⁺; **c**), a mouse expressing ArchT in thalamus (**d**) and a mouse expressing ArchT in M1/2 (**e**). Top, responses to individual light pulses (shaded region; time-normalized). Bottom, average PETH of baseline-subtracted activity. **f**–**h**, Baseline-subtracted mean activity (light modulation) of movement-uncorrelated and movement-correlated neurons significantly affected by

light pulses (mean ± s.e.m.; *P* values, uncorr. versus corr., two-sided Wilcoxon rank-sum test). **f**, Control (affected neurons, *n* = 30 of 184 uncorr. and *n* = 16 of 157 corr., 9 mice). **g**, ArchT expressed in thalamus (affected neurons, *n* = 146 of 645 uncorr. and *n* = 25 of 182 corr., 7 mice). **h**, ArchT expressed in M1/2 (affected neurons, *n* = 39 of 271 uncorr. and *n* = 13 of 170 corr., 3 mice). **i**, Prediction of whisker movements from population activity using a model trained on light-off data and evaluated on light-off (within) and light-on (across) data. *R²*, explained variance (control: *n* = 14 FOVs, 8 mice; thalamus: *n* = 21 FOVs, 7 mice; M1/2: *n* = 10 FOVs, 3 mice). **j**, Out-of-sample *R²* ratio (ArchT expressed in thalamus versus control, *P* = 0.012; ArchT expressed in M1/2 versus control, *P* = 0.10; Kruskal–Wallis (*P* = 0.00053) followed by two-sided Wilcoxon rank-sum tests with Bonferroni correction). In box plots, the central line and box represent the median and 25th–75th percentiles, and whiskers extend to the most extreme data points excluding outliers (larger than 1.5× the interquartile range).

from those caused by this light-evoked brief whisker movement, we restricted our single-neuron analysis to the later light pulse window (0.5–1.0 s or 0.5–1.5 s) during which the level of ongoing movements was similar to that of baseline (before the light pulse) (Extended Data Fig. 16). In ArchT-expressing mice, optogenetic suppression of thalamic terminals significantly altered the activity levels in a fraction of movement-correlated neurons (14%), producing a net decrease in their activity. This decrease was significantly larger than changes observed in movement-uncorrelated neurons (Fig. 5c–h). To further test the contribution from VPm and POm inputs, we expressed ArchT exclusively in either VPm or POm (Extended Data Fig. 17). Independent optogenetic suppression of VPm or POm terminals revealed that VPm predominantly contributes to the decreased activity of movement-correlated neurons. However, suppression of VPm terminals appears to produce a less robust effect than combined suppression of VPm and POm terminals. In M1/2 ArchT-expressing mice and in control mice, while we also detected changes in the activity of a fraction of movement-correlated neurons (7.6% and 10%, respectively), the net effect did not differ from that of movement-uncorrelated neurons. To address the optogenetic effect in a more comprehensive manner, we built a linear decoder using light-off periods and tested this model on data acquired during light presentation. We found a consistently larger decrease in $R^2$ values for thalamic ArchT-expressing mice than for M1/2 ArchT-expressing or control mice (Fig. 5i,j). In some experiments, the across $R^2$ value was close to zero, demonstrating that inhibition of thalamic inputs critically changed the relationship between population activity and spontaneous movements. These results provide evidence for the contribution of sensory feedback-independent, thalamic inputs, in particular from VPm, to spontaneous movement-dependent activity in cortical neurons.

## Discussion

We investigated anatomical wiring rules for the functional heterogeneity of cortical neurons. Our results provide insights into the functional and anatomical organization of cortical PNs in the context of behavioural state. We first demonstrated that the representation of spontaneous movements in wS1 PNs is stable over multiple days, both at the single-cell and population levels. We investigated the basis of the mechanisms underlying this stable, heterogenous representation. Based on the modest role of neuromodulatory inputs in wS1 and the decisive effect of glutamatergic inputs in driving this spontaneous movement-dependent activity, we investigated the anatomical architecture of brain-wide presynaptic networks of two subsets of wS1 L2/3 PNs (movement-uncorrelated and movement-correlated neurons). We found that individual PNs in superficial layers, regardless of their functional properties, receive highly converging inputs from most wS1-projecting brain areas, suggesting that individual cortical neurons have direct access to a broad range of information from diverse brain regions. Nonetheless, functionally distinct cortical neurons showed anatomical biases in the proportions of specific long-range presynaptic inputs (Extended Data Fig. 18), suggesting that information from a given presynaptic area may be selectively amplified or reduced through the number of inputs.

Movement-correlated neurons received a smaller fraction of motor cortical inputs and a larger fraction of thalamic inputs relative to movement-uncorrelated neurons. Moreover, we found a negative correlation between the modulation of neuronal activity by spontaneous movements and the fraction of motor cortical presynaptic cells and, conversely, a positive correlation between modulation of neuronal activity and fraction of thalamic presynaptic cells. Not only was the fraction of motor inputs onto movement-correlated neurons lower, but optogenetic suppression of motor cortical axon terminals in wS1 also had a minimal effect on spontaneous movement-dependent L2/3 PN activity. By contrast, in addition to an increased fraction of

thalamic inputs converging onto movement-correlated neurons, we found that thalamic inputs directly contribute to the activity of PNs during spontaneous movements. A role for thalamic nuclei in driving behavioural-state-dependent cortical activity is consistent with previous observations. First, robust manipulations of thalamic activity profoundly alter cortical state[44,45]. Second, thalamic nuclei activity correlates positively with spontaneous movements[9,23,45–48], whereas the activity of neurons in cortical areas can fluctuate positively and negatively with spontaneous movements[9]. Third, thalamic inputs can drive L2/3 PNs more efficiently than cortical inputs[49]. However, that individual PNs encoding spontaneous movements have an enhanced thalamic input fraction could not have been predicted from previous studies. Although movement-correlated PNs receive almost equally abundant inputs from VPm and POm, our optogenetic study demonstrates that the VPm nucleus is primarily responsible for driving the movement-dependent activity in wS1 L2/3. Eliminating whisker movements through muscle paralysis and sensory responses through whisker trimming did not disrupt spontaneous movement-dependent activity in wS1, strongly supporting the notion that this activity is unlikely to be a direct consequence of sensory feedback[44,48,50]. This raises the issue of what information VPm neurons relay to the primary somatosensory cortex beyond sensory transmission, and how the transmission of sensory and non-sensory information is achieved. Sensory thalamic neurons, as cortical neurons, are also active in the absence of sensory stimuli and feedback[44], suggesting that they can encode non-sensory information. Anatomical analysis supports the heterogeneous innervation patterns of individual VPm neurons across different cortical layers[51,52]. Future studies will be needed to address the potential functional heterogeneity in VPm neurons[47].

The release of ACh and noradrenaline in cortex is closely linked to spontaneous movements, as also confirmed in our GRAB sensor experiments[25,33–36]. Our results shows that these inputs may influence the activity levels, especially with respect to sensory responses, consistent with the gain modulation of sensory responses during movement by neuromodulation[24,37,38]. However, our neuromodulatory blockade results suggest that the direct neuromodulatory inputs to the cortex may not be the main drivers for the neuronal activity in relation to spontaneous movements. Instead, the subset of neurons that reliably tracked the behavioural state appeared to be driven by long-range glutamatergic inputs, particularly from the thalamus. Neuromodulators profoundly alter the thalamic activity mode, which can in turn affect the recipient cortex[53–55]. One possibility is that neuromodulators utilize strong thalamic connections to influence PNs that track behavioural states, rather than directly driving these neurons[53–55] (Extended Data Fig. 18).

The finding that movement-encoding neurons receive a smaller fraction of inputs from motor cortical areas is puzzling given the general association of motor cortical areas with movement execution. Motor cortical areas have been shown to be required for learning and production of skilled and accurate movements[56], rather than for the execution of innate movement sequences. Instead of being learned motor skills, whisking and locomotion are innate behaviours and part of the whole range of coordinated movements that represents a behavioural state. Although inactivation of M1/2 affects wS1 activity, it does not abolish the behavioural state-dependent changes in S1 activity[57,58]. M1/2 axons in wS1 do not appear to be exclusively dedicated to transmitting movement features, but instead carry multiple aspects of sensorimotor behaviour including touch[21]. The M1/2 pathway is known to alter cortical sensory responses through the engagement of a local disinhibitory circuit that changes stimulus-response gain to relevant inputs[22,38]. The M1/2 pathway may be more relevant for sensorimotor learning in wS1, instead of determining behavioural state-dependent representations. If the role of these behavioural state-sensitive neurons is to reliably update the behavioural state to the local network, one

might expect these neurons to be less susceptible to the plasticity of sensorimotor learning.

We utilized a single-cell-based monosynaptic retrograde tracing approach using a modified rabies virus, which is a powerful method for mapping brain-wide presynaptic inputs. However, this approach is limited due to only a fraction of inputs being labelled, lack of knowledge regarding the strength of labelled connections, and potential bias associated with viral tropism. To overcome these issues, we directly compared the brain-wide presynaptic networks of two functionally distinct groups using the same method. If a bias exists, it is reasonable to assume that it is similar in both groups. Our study is based on the proportion of presynaptic cells, in different brain areas, of individual cortical neurons. It remains undetermined whether the presynaptic cells are functionally similar, and what the output connectivity of these functionally distinct neurons is.

We found highly convergent yet characteristic presynaptic connectivity patterns depending on the functional property of each neuron. This raises questions about the functions of the movement-correlated neurons within the local network. These neurons may report moment-to-moment behavioural states within the sensory cortex and provide a self-referenced framework for integrating information across other brain areas. The stable, sensory-independent activity of wS1 L2/3 neurons may constitute a reflection of internal models of the body and environment that enable the integration of sensory information with external and internal contexts, prediction of sensory consequences and guidance of actions[59], and a reflection of a developmental wiring process[60].

Our results revealed anatomical biases in long-range glutamatergic inputs mapping onto functionally distinct neurons. These specific input patterns together with the stable representation uncovered here suggest the existence of preconfigured patterns of activity in S1 in the context of behavioural state[59].

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

# Methods

## Mice

All animal procedures were conducted in accordance with a protocol approved by the National Institutes of Health Institutional Animal Care and Use Committee (IACUC), Bethesda, MD, USA, and complied with Public Health Service Policy on Humane care and Use of Laboratory Animals and the Guide for the Care and Use of Laboratory Animals. We used the following transgenic mouse lines: Emx1-IRES-Cre (JAX 005628)[61], tetO-GCaMP6s (JAX 024742)[62], and CaMK2a-tTA (JAX 007004)[63]. We performed experiments on 10 Emx1-IRES-Cre, 71 CaMK2a-tTA;tetO-GCaMP6s, and 3 GCaMP6s$^+$.CaMK2a-tTA$^-$ mice. We used both male and female mice (females, 45%), 12–24 weeks old at the experimental endpoint. The percentage of $Mov_{up}$ and $Mov_{down}$ neurons and sensory-responsive neurons was similar between the male and female groups ($Mov_{up}$ and $Mov_{down}$ neurons, $30 \pm 11\%$ versus $33 \pm 11\%$, $n = 26$ versus 21 mice, respectively, $P = 0.27$; sensory-responsive neurons, $13 \pm 5.9\%$ versus $12 \pm 6.1\%$, $n = 19$ versus 14 mice, respectively; $P = 0.57$, two-sided Wilcoxon rank-sum test). Mice were housed in groups, in individually ventilated and enriched laboratory cages, in climate-controlled rooms (22 °C; 45% humidity), under a reverse 12 h light:12 h dark cycle (light on, 09:00), and with ad libitum access to water and food. After surgical procedures, mice were housed individually. All experiments were performed in the dark phase of the cycle. Mice in test and control groups were littermates and randomly selected.

## Surgeries

All surgical procedures were performed stereotaxically, including injection of recombinant adeno-associated viruses (rAAVs), head plate implantation, and cranial window implantation, and were carried out under aseptic conditions. Mice were anaesthetized with isoflurane ($1.0$–$2.0\%$ in $O_2$ at $0.8$ l min$^{-1}$). The eyes were protected with ophthalmological ointment, and body temperature was maintained at ~37 °C using a heating pad (Stoelting). Dexamethasone (0.2 mg kg$^{-1}$ of body weight; subcutaneous injection) was administered at least 1 h prior to cranial window implantation, to prevent brain oedema. Exposed dura mater was perfused with sterile Ringer's solution (in mM, 150 NaCl, 2.5 KCl, 10 HEPES, 2 CaCl$_2$, 1 MgCl$_2$; pH 7.3 adjusted with NaOH; 300 mOsm). After surgery, mice were treated with meloxicam (2 mg kg$^{-1}$; subcutaneous injection) every 24 h for 3 days, to minimize pain and inflammation, and with enrofloxacin (0.1 mg ml$^{-1}$ in drinking water) for 5 to 10 days, to prevent infection. Wellness and body weight were monitored daily for 10 days.

The first surgery consisted of rAAV injection combined with headpost and cranial window implantations, rAAV injection followed by headpost implantation, or headpost implantation only. For rAAV injection, at each target coordinate, the skull was thinned, and a craniotomy (~50 μm diameter) was made using fine forceps. A glass micropipette (5–10 μm outer diameter tip) attached to a nanoinjector (WPI) was used to deliver the viral vector (at 20–50 nl min$^{-1}$). After injection, the pipette was left in place for ~5 min before being slowly retracted. To express GCaMP6f in PNs, we injected rAAV5-Syn-Flex-GCaMP6f-WPRE-SV40 (Addgene 100833) in the right hemisphere wS1 of Emx1-IRES-Cre mice (in mm relative to Bregma: anterior–posterior (AP) −0.80 and medial–lateral (ML) 3.50; AP −1.20 and ML 3.40, the pipette was angled at 21°, and 30 nl were injected at the subdural depths of 350, 250 and 150 μm). For GRAB sensor experiments, we injected AAV9-hsyn-Ach4.3 (Ach3.0) or AAV9-hsyn-NE2m (NE3.1) (WZ Biosciences) in the right hemisphere wS1 of GCaMP6s$^+$.CaMK2a-tTA$^-$ mice (3 injection sites; same injection parameters as above). For simultaneous imaging of L2/3 PNs and optogenetic suppression of thalamic or motor cortical terminals in wS1, we injected rAAV5-CAG-ArchT-tdTomato (UNC Vector Core AV4595B), rAAV5-CAG-tdTomato (Addgene 59462), or rAAV5-Syn-tdTomato (Addgene 51506) in the VPm (AP −1.70, ML 1.85 and dorsal–ventral (DV) 3.15; 50–60 nl) and/or POm (AP 2.00, ML 1.32, and DV 3.00;

50–60 nl), or the M1/2 (AP −1.00, ML 1.00, and DV 0.8 to 0.2, 15 nl per each 100 μm) of tetO-GCaMP6s;CaMK2a-tTA mice. We implanted a custom-made Y-shaped titanium head plate using dental cement (Super-Bond C&B, Parkell). The exposed skull was covered with a thin layer of clear dental cement and, subsequently, opaque biocompatible silicone (Kwik-Cast, WPI), if applicable. A craniotomy was made over the right hemisphere wS1 (centred at AP 1.1 and ML 3.3), and a glass cranial window (diameter, 3 mm; thickness, 100–150 μm) was placed and secured over the craniotomy using cyanoacrylate adhesive (3M) and dental cement. For in vivo neuropharmacological experiments and single-neuron monosynaptic input tracing, the cranial window had a rectangular laser-cut opening (0.30 × 0.80 mm, Potomac Photonics) covered with transparent biocompatible silicone (Kwik-Sil, WPI)[64]. On the day of imaging, the silicone plug was removed and micro-durotomy was performed for direct access to the brain.

Recordings were initiated after a minimum period of 3–5 weeks post-rAAV injection, for stable expression of GCaMP6f, ArchT, GRAB$_{ACh}$ or GRAB$_{NE}$.

## Behaviour

All behavioural experiments were performed in darkness. Under head-fixation, mice with all intact whiskers were free to run on a wheel. The mouse face and whiskers were video-recorded at 250 fps using a high-speed camera (acA2000-340kmNIR, Basler) with an 8 mm lens (LM8JC, Kowa), under infrared LED illumination (850 nm, Mightex). Image acquisition was controlled by StreamPix (NorPix). The wheel (diameter, 15 cm; width, 5.5 cm) was set so that only forward movement was permitted. To extract locomotion speed, we used a 2500 CPR resolution motion encoder (Model 260 Accu-Coder, Encoder) affixed to the wheel shaft. Motion encoder pulses were converted to speed using a counter and LabView software (National Instruments), for online visualization of speed. To synchronize behavioural and neuronal data, voltage signals from each video-frame exposure and wheel speed were digitized and recorded at 10 kHz through a data acquisition card (PCI-6052E, National Instruments), using the Prairie View Interface (Bruker).

## Unilateral mystacial pad paralysis

In a subset of mice, paralysis of the left or both mystacial pad(s) was achieved through a local, subcutaneous injection of BTX (single injection per pad; 0.5 Units in 10 μl per injection, BOTOX), under isoflurane anaesthesia.

## Whisker stimulation

Deflection of the left whiskers was achieved using a solenoid valve-controlled pole (diameter, 3 mm; length, 5 cm). To maximize contact with all whiskers, the pole was positioned in alignment with the mystacial pad, at an angle of ~65° (distance between the mystacial pad and the pole during stimulation, ~5 mm). Stimuli (trains of 28–49 stimuli; speed, ~600 mm s$^{-1}$; duration, 50 or 250 ms; interval, 3–5 s) were produced and synchronized to behavioural and neuronal data using the Prairie View Interface (Bruker). To ensure that changes in neuronal activity related to whisker deflection could be isolated from changes in neuronal activity related to potential alterations in movements during deflection, recordings included trials of whisker deflections coupled with sound, as well as sound-only trials[61]. Neurons were classified as sensory stimulus-responsive if they responded exclusively during whisker deflections, but not during sound-only trials. The percentage of sensory-responsive neurons did not differ when using 50-ms-duration versus 250-ms-duration whisker stimuli ($14 \pm 5.9\%$ versus $11 \pm 6.0\%$, $n = 21$ versus 12 mice, respectively; $P = 0.15$, two-sided Wilcoxon rank-sum test).

## In vivo imaging and optogenetics

Imaging was performed using a two-photon microscope (Ultima Investigator, Bruker) and a fs-pulse Ti:Sapphire laser (Mai Tai DeepSee,

Spectra-Physics), tuned between 860 and 1040 nm, for imaging of difference fluorescent proteins. The microscope was equipped with an 8 kHz resonant galvanometer and a water-immersion 16× objective (0.8 NA, Nikon) coupled to a 400-µm-range, $z$-axis piezoelectric drive. GCaMP6 and RFP fluorescence signals were passed through a 525/70 m or 595/50 m filter, respectively. Fluorescence was detected and amplified using GaAsP PMTs (Hamamatsu) and a dual preamplifier, prior to digitization. For two-photon $Ca^{2+}$ imaging, we performed one session per day (recording time ~66 ± 21 min). Images (resolution, 512 × 512) were collected at ~30 Hz, in single plane mode. FOVs ranged from 271 × 271 µm (for functional identification of a neuron for subsequent electroporation) to 573 × 573 µm (for characterization of neuronal patterns of activity) with an average excitation laser power of ~33–76 mW at the objective. For simultaneous two-photon $Ca^{2+}$ imaging and optogenetic inhibition of thalamic axon terminals, we used a collimated 625 nm LED beam (Prizmatix UHP-T-625-SR)[65,66]. The LED was on during the turnaround of the resonant galvanometer. Trains of light pulses (25 pulses; duration per pulse, 1–1.5 s; interval, 5 s) were generated and synchronized to behavioural and neuronal data acquisition using the Prairie View Interface (Bruker). A filter (FF02-617/73-25, Semrock) was used to narrow the LED spectrum, and a dichroic mirror (FF556-SDi01, Semrock) was used direct the LED light onto the brain tissue and pass GCaMP6 florescence signals onto the PMT. The average power of the LED was 10–50 mW at the objective. To minimize the effect of light stimulation on the spontaneous movements of mice, we shielded the objective lens. However, this shield did not completely block the light stimulation. We observed a brief (<0.5 s) whisker movement in both control and ArchT-expressing mice at the onset of light stimulation (Extended Data Fig. 16).

## In vivo neuropharmacology

A durotomy (~50 µm) was made through the access port of the implanted cranial window, under isoflurane anaesthesia (Fig. 2a). Mice were allowed to recover for ~30 min prior to recordings. Following acquisition of baseline behavioural and neuronal activity data, Ringer's solution was replaced by Ringer's solution supplemented with receptor blockers, and recordings were reinstated. In sham sessions, Ringer's solution was replaced by Ringer's solution without blocker addition. Only data acquired 20 min or more after blocker application or Ringer's replacement (in sham sessions) were considered for analysis. Each session, 1–2 days apart, lasted a median of 2 h 15 min (effective spontaneous activity recording time, 1 h and 8 min): baseline period, 42 min (effective spontaneous activity recording time, 30 min); blocker application or Ringer's replacement, 3 min; waiting period following blocker application or Ringer's replacement, 21 min; receptor blockade period, 45 min (effective spontaneous activity recording time, 30 min). Baseline versus sham/receptor blockade recording times did not differ ($P > 0.05$, two-sided Wilcoxon signed-rank test). We used a combination of atropine (1 mM) and mecamylamine (1 mM) to block ACh receptor (AChR)[37,38,67,68], a combination of prazosin (1 mM) and propranolol (1 mM) to block noradrenaline receptor (NAR)[69], D-AP5 (1 mM) to block NMDA receptor (NMDAR), and a combination of D-AP5 and DNQX (2 mM) to block both NMDA and AMPA receptor (AMPAR). In one mouse, we used prazosin (1 mM), propranolol (1 mM) and yohimbine (1 mM) to block the noradrenaline receptor. Blocker application session sequences were either AChR–NAR–NMDAR/AMPAR ($n = 3$) or NAR–NMDAR/AMPAR–AChR ($n = 2$), randomly assigned per animal. In two mice, only ACh receptor ($n = 1$) or noradrenaline receptor ($n = 1$) blocker sessions were performed. The position of ACh or noradrenaline receptor in the sequence did not affect neuronal correlations ($P > 0.05$, linear mixed model controlling for days as confounding factor for position). To test the effectiveness of drug diffusion into the imaging FOV (481.4 × 481.4 to 572.9 × 572.9 µm², at 330 ± 30 µm of depth), we applied TTX (10–100 µM) in the final recording session. Experiments in which TTX did not silence neuronal activity over the entire FOV within 15–20 min were excluded ($n = 2$ mice).

To evaluate the effectiveness of ACh and noradrenaline receptor blockade throughout the entire FOV, we performed equivalent neuropharmacological experiments in mice expressing either $GRAB_{ACh}$ or $GRAB_{NE}$ in wS1 L2/3 neurons.

## In vivo single-neuron monosynaptic input tracing

After micro-durotomy, we performed two-photon $Ca^{2+}$ imaging and selected a target neuron based on its activity profile across behavioural states. Classification of the target neuron as movement-uncorrelated or movement-correlated was confirmed during post-hoc analysis. After imaging, the mouse was lightly anesthetized, and two-photon guided electroporation of the target neuron was performed as described previously, for monosynaptic input tracing[14,17,39,70–72]. A glass pipette (14 ± 1.5 MΩ) was filled with intracellular solution (in mM, 130 potassium gluconate, 6.3 KCl, 0.5 EGTA, 10 HEPES, 5 sodium phosphocreatine, 4 Mg-ATP, 0.3 Na-GTP; pH 7.4 adjusted with KOH; 280–300 mOsm) supplemented with Alexa 594 hydrazide (50 µM, A10442, Thermo Fisher Scientific) and two DNA plasmids (pAAV-EF1α-mTagBFP-HA-T 2A-mCherry-TVA-E2A-N2c, 0.15 µg µl⁻¹; pAAV-CAG-N2c, 0.05 µg µl⁻¹). The resistance of the pipette tip was monitored continuously (Axoporator 800A, Molecular Devices). Positive pressure was applied to the pipette (70 mbar), which was visually advanced through the durotomy, using a micromanipulator (PatchStar, Scientifica). Upon entering the cortex, the pressure was swiftly decreased (35 mbar). Then, within ~50–100 µm from the target neuron the pressure was further decreased (15 mbar). The pipette was slowly advanced towards the soma of the target neuron, until the tip resistance increased by at least 20%. The pressure was released, and a train (100 Hz, 1 s) of electric pulses (−10 V, 0.5 ms) was applied (Axoporator 800A), after which the pipette was retracted. The electroporated neuron was imaged 20 min later to evaluate its survival. Thereafter, we injected G-deleted, envelope-A coated CVS-N2c rabies virus carrying RFP (kindly provided by the Center for Neuroanatomy with Neurotropic Viruses) in the vicinity (within ~150 µm) the electroporated neuron (rate, 30 nl min⁻¹)[73]. Then, the access port of the cranial window was sealed using biocompatible silicone. Survival and successful transfection of the electroporated neuron was monitored within 2–3 days after electroporation and up to the experimental endpoint. Structural, two-photon $z$-stacks (1–5-µm steps; resolution, 512 × 512; FOV, 102 × 102 to 271 × 271 µm) including the imaging FOV and/or the target neuron were acquired before and after electroporation, to track individual cells volumetrically throughout the experiment. Local, wS1 presynaptic networks were followed structurally through two-photon imaging of GCaMP6 and RFP ($z$-stacks, 1–5-µm steps; resolution, 512 × 512; FOV, 271 × 271 to 814 × 814 µm). For $z$-stack acquisition the average laser power was depth-adjusted linearly and did not exceed 100–150 mW at the objective. Mice were euthanized at day 11 (±1.5 days) following electroporation, and brains were processed for ex vivo input tracing. Brains containing less than 100 presynaptic cells were excluded from analysis ($n = 1$).

## Histology

Upon completion of recordings, mice were deeply anesthetized and perfused transcardially with 4% formaldehyde in PBS. Post-perfusion, brains were immersion-fixed in 4% formaldehyde in PBS for 2–3 h and then transferred to 30% sucrose in PBS.

For input tracing experiments, whole-brain free-floating sequential coronal sections (50-µm-thick) were obtained using a microtome (SM2010R, Leica). Sections were rinsed 3 times in PBS, incubated in blocking solution (5% normal serum and 1% Triton X-100 in PBS) at room temperature for 1 h, and subsequently incubated in primary antibody solution (2% normal serum and 1% Triton X-100 in PBS) at 4 °C for 48 h. Primary antibodies were detected through incubation in secondary antibody solution (2% normal serum and 1% Triton X-100 in PBS) at room temperature for 2 h. We used the following primary and secondary antibodies and respective dilutions: anti-RFP 1:500

(600-901-379, Rockland); anti-GABA 1:500 (A2052, Sigma), IgY-Alexa Fluor 555 1:200 (A21437, Thermo Fisher Scientific); IgG-Alexa Fluor 647 1:200 (A21245, Thermo Fisher Scientific). Sections were rinsed in PBS and sequentially mounted on glass slides. Neuronal nuclei were revealed through a fluorescent Nissl stain (NeuroTrace 435/455, N21479, Thermo Fisher Scientific), after which sections were cover-slipped. Whole-brain serial sections were imaged using an epifluorescence illumination microscope (Zeiss Imager.M2). Multiple z-stacks (10 μm steps), covering each section in its entirety, were acquired using Neurolucida (MBF Bioscience). z-stacks were aligned and collapsed onto a single image using Deep Focus (Neurolucida).

For all other experiments, brain sections were similarly generated and mounted, and neuronal nuclei were visualized either using fluorescent Nissl stain (NeuroTrace 435/455 or NeuroTrace 530/615, N21482, Thermo Fisher Scientific) or DAPI (Fluoromount-G mounting medium, Thermo Fisher Scientific). Entire sections were imaged using an Axio1 Scanner (Zeiss). FOV location within wS1 was confirmed either by targeted two-photon laser microlesions (the laser beam was focused at a subdural depth of 200 μm; 800 nm; 5–30 s; -0.5 W)[74] or fluorescent dye (DiI (42364, Sigma) or Fast Blue (17740, Polysciences)) injection at the experimental endpoint and ex vivo histological analysis. Tissue imaging and histological analysis were done blinded to the experimental groups. For analysis of thalamic ArchT-expression areas, a composite image of a brain section at the injection centre for either VPm or POm was selected, and expression areas annotated; each brain section was manually aligned to the corresponding mouse brain atlas section, and the expression areas of the different mice were overlaid[75–77].

### Defining behavioural events
Video recordings of face and whiskers were processed using a custom MATLAB routine. To detect whisker movements, we first defined a region of interest (ROI) encompassing the left or right whiskers in an image that consisted of the s.d. of representative frames of each session. We then computed the absolute power of the spatial derivative of consecutive frames (WM trace). Both the whisker movement and locomotion speed traces were downsampled to 30 Hz, averaging the values acquired during a $Ca^{2+}$ imaging frame. To detect behavioural events, the whisker movement trace was baseline-subtracted (10th percentile of the full trace) and normalized. We then applied a threshold to the whisker movement trace (3× minimum s.d., calculated using a 30-s sliding window). Detection was visually inspected for all sessions, and the threshold was adjusted when applicable. Then, local maxima were calculated, and peaks less than -0.5 s apart were considered as part of a single event; the first peak was considered as onset and the last, as offset. Events with an integral value smaller than 20 (<3% of the session time) were excluded from analysis. Whisker movement events were considered as WL when the maximum locomotion speed was higher than 0.20 cm s$^{-1}$ and, conversely, as $W_{only}$ when the locomotion speed did not exceed 0.20 cm s$^{-1}$. For comparisons across mice, the raw whisker movement trace was baseline-subtracted and normalized to its maximum; for BTX experiments, across session data were normalized to maximum according to the first session.

### Processing of two-photon calcium images
Two-photon $Ca^{2+}$ images were processed using Suite2p[78], in Python, with default parameters, unless otherwise indicated. Following subtraction of neuropil (fixed scaling factor of 0.7) and baseline (calculated on filtered traces, using a gaussian kernel of width 20 and a sliding window of 60 s), fluorescence traces were deconvolved using non-negative spike deconvolution[79] with a fixed decay timescale of 0.7 s for GCaMP6f and 1.5 s for GCaMP6s. To ensure that only somatic traces were included in the analysis, ROIs were manually curated by an analyst blinded to the experimental group. Aligned image series were visually inspected to control for z-drifts; data showing z-drifts were excluded from analysis. Tracking of the same neurons across sessions

was done semiautomatically using a MATLAB script. All analysis was based on deconvolved traces; for presentation purposes only, we used neuropil subtracted fluorescence traces normalized to the maximum (F), overlaid with deconvolved fluorescence traces normalized to the maximum (F deconv.), unless otherwise indicated. To generate temporal raster plots (Fig. 1e), the activity of each neuron was averaged over ~0.5 bins, z-scored and smoothed using a 1-s moving average filter; individual neurons were sorted by the first principal component of neuronal activity.

### Processing of two-photon GRAB sensor images
Two-photon $Ca^{2+}$ images were motion-corrected using Suite2p[78]. Thereafter, we extracted the mean fluorescence intensity of pixels within 6 ROIs (75 × 75 pixels) manually spread over the entire FOV, avoiding large vessels. In addition, to extract a full FOV fluorescence intensity mean, we first isolated sensor$^+$ pixels and excluded vessel-related pixels by applying a threshold to pixel intensity (99.9th percentile > intensity > 50th percentile) over the motion-corrected, whole recording session average image[80]. Baseline traces were obtained essentially as described in 'Processing of two-photon $Ca^{2+}$ images' but using a gaussian kernel of width 30 and a sliding window of ~120 s. The mean fluorescent values of single ROIs or full FOVs were baseline-subtracted ($\Delta F$) and z-scored.

### Decoding analysis
We trained a linear decoder to decode behavioural variables from neuronal activity. We minimized the ridge regression[81] objective function (equation (1)).

$$\hat{w} = \underset{w}{\mathrm{argmin}} \sum_{i=1}^{N} \|y_i - w^T x_i\|_2^2 + \alpha \, \|w\|_2^2 \tag{1}$$

where $y_i$ is the behavioural variable at time (frame) $i$, $x_i$ is the neuronal activity matrix, $w$ is the weight vector, and $\alpha$ is the ridge parameter (regularization). We normalized the behaviour and neuronal activity by z-score for the cross-day recordings. We did not normalize the neural activity for the neuromodulatory experiments, as activity was recorded continuously on the same day. For optogenetic experiments, we ran analysis with (shown in figures) and without data normalization, as well as with and without (shown in the figures) rebound cells, and no substantial difference was found. We randomly split the trials into training (75%) and test sets (25%). The weight vector was estimated on the training set, and the ridge parameter was selected by leave-one-out cross-validation[82] on the training set.

We evaluated the decoding performance by the out-of-sample (test set) coefficient of determination ($R^2$) (equation (2))[81].

$$R^2 = 1 - \frac{\sum_{(i)} (y_{(i)} - \hat{y}_{(i)})^2}{\sum_i (y_{(i)} - \bar{y}_{(i)})^2} \tag{2}$$

where $(i)$ is the index of out-of-sample trials, $\hat{y}_{(i)} = \hat{w}^T x_{(i)}$ and $\hat{w}$ is the estimated weight vector from the training set by (1). Using the weight vector estimated from the training set, we decoded the behavioural variables on the test (held out) set within the same condition or session (referred as within) and the trials of other conditions or sessions (referred as across). Then, we calculated the out-of-sample $R^2$ using the predicted and true values for within and across data. The out-of-sample $R^2$ is not prone to overfitting and will not be inflated. Note that the out-of-sample (test set) $R^2$ can be negative if the fitted model does not predict the test set at all, which indicates that the neural correlations could be substantially different between the training and test sets.

### Modulation of neuronal activity
Data were analysed using MATLAB scripts. The activity (F deconvolved) of each neuron was aligned to the onset and offset of spontaneous

movements, $W_{only}$ and WL. Baseline and post-offset activities refer to a 0.5-s window preceding movement onset and a 0.5-s window after movement offset, respectively. A neuron was considered as $Mov_{up}$ if its average activity during $W_{only}$ and/or WL events was significantly higher than its average activity during baseline and/or significantly higher than its average activity post-event ($P < 0.01$, two-sided paired-sample $t$-test). Conversely, a neuron was considered $Mov_{down}$ if its average activity during $W_{only}$ and/or WL events was significantly lower than its baseline and/or significantly lower than its post-event average activity ($P < 0.01$, two-sided paired-sample $t$-test). Neurons exhibiting opposite changes in activity for $W_{only}$ and WL were rare ($0.7 \pm 0.2\%$ of all cells) and were not considered for further analysis. Otherwise, neurons were considered as movement-uncorrelated. Modulation refers to the mean activity during movement, irrespective of movement duration, minus the mean activity during baseline, averaged across spontaneous movements ($W_{only}$, WL, or $W_{only}$ + WL).

Time-normalized PETHs were created by normalizing the data to match the average duration of WL events across mice. To compute the correlation (Pearson's linear correlation coefficient, $r$) between the activity of individual neurons and each behavioural variable, deconvolved fluorescent traces, and corresponding whisker movements and locomotion speed raw traces were binned (bin size, ~0.5 s).

Sensory stimulus-responsive neurons were defined by a significantly higher average activity within a response window of 0.5 s after the onset of whisker stimulation (coupled with sound) versus baseline ($P < 0.01$, two-sided paired-sample $t$-test). We excluded cells that also responded to sound-only stimuli versus baseline ($P < 0.01$, two-sided paired-sample $t$-test). Baseline was calculated on a 0.5-s window preceding stimulus onset. Sensory stimulus-response magnitude refers to the mean activity during the response window minus the mean activity during baseline, averaged across all whisker stimulations.

For neuropharmacological experiments, the distribution of modulation values for each WL neuron in the presence of receptor blocker(s) was compared to that of baseline (prior to blocker application, two-sided $t$-test). Similarly, we compared the distribution of sensory stimulus-response magnitude values before and after blocker application for each sensory stimulus-responsive neuron (two-sided $t$-test).

For optogenetic experiments, light-off periods were used to identify neurons as movement-uncorrelated or movement-correlated. Neurons were classified as light modulated if their activity during the light pulse differed significantly from that of baseline (0.5-s window prior to light pulse onset). Most L2/3 PNs fire sparsely; only neurons that showed an average baseline activity ($F$ deconvolv.)—that is, prior to the light pulse, higher than the 75th percentile of the average baseline activity across all cells were included in the analysis. To generate time-normalized PETHs, light pulse data were normalized to match the maximum pulse duration across experiments (1.5 s). Statistical testing on PETHs of movement-uncorrelated and movement-correlated neuronal subpopulations was performed on the original data sampling rate (~30 Hz).

## Whole-brain reconstruction, annotation and registration

To analyse the brain-wide distribution of presynaptic neurons, we adapted a previous pipeline[83,84]. To reconstruct a whole-brain 3-dimensionally, individual section images were aligned using Brain-Maker (MBF Bioscience). Brain-wide presynaptic neurons (RFP+) were automatically segmented using NeuroInfo (MBF Bioscience) and manually annotated according to brain area. Local, wS1 presynaptic neurons were manually annotated based on cortical layer location. Distinct layers were identified based on the characteristic depth-varying density of NeuroTrace+ neurons. Presynaptic neurons within each layer were identified as glutamatergic (GABA−) or GABAergic (GABA+). To confirm the colocalization of RFP and GABA, a subset of brains was re-imaged using a confocal microscope ($z$-stacks, 3-μm steps, C2, Nikon). Identification of glutamatergic and GABAergic neurons was equivalent for the two different imaging methods. Each serially reconstructed

brain was registered to the Allen Mouse Common Coordinate Framework, and brain-wide presynaptic neurons (RFP+) were automatically re-identified according to distinct anatomical structures. Reconstructions and registrations were conducted blindly. Manual identification was independently performed by two analysts (one analysist was blinded to the activity profiles of the postsynaptic neurons). All manual and automatic identifications were coherent.

## Analysis of brain-wide presynaptic networks

**Fine-scale spatial registration of wS1 presynaptic networks.** Postsynaptic neurons did not survive until the experimental endpoint[14]. We used the centre of mass of glutamatergic presynaptic networks in L2/3 to estimate the position of the postsynaptic neurons for all 22 subjects[14], which were then averaged to construct a reference postsynaptic site on the Allen Mouse Common Coordinate Framework. From the brain atlas, we manually marked the boundaries of the cortical surface and performed surface triangulation. Using this triangulated cortical surface, we then estimated the surface's normal vector that goes through the reference postsynaptic site and serves as the reference normal vector. Individual wS1 presynaptic network of each subject was then rigidly aligned based on the position of the reference postsynaptic site and orientation of the reference normal vector.

**Layer-by-layer horizontal flat projections.** After the fine-scale registration, we concatenated all neurons from both movement-uncorrelated and movement-correlated groups. Then we performed PCA on each layer of the presynaptic networks to obtain the population best-fitted plane. Corresponding neurons from the layer were then projected onto the best-fitted plane. We then performed a rigid parameterization and mapped the neurons to a two-dimensional coordinate system. The resulting parameterization of each layer from every subject with gaussian kernel density estimation is visualized in Extended Data Fig. 11.

**Statistical analysis of group-wise spatial distribution differences.** To explore whether there was any difference in the spatial pattern of presynaptic networks between movement-uncorrelated and movement-correlated groups, we tested the null hypothesis of no spatial distribution difference between multiple local and long-range anatomically annotated presynaptic neurons of the two groups. The final statistical analysis incorporated Bonferroni correction. We chose the 2-Wasserstein distance function as the test statistics, and we performed a one-sided randomization test ($n = 10,000$) to approximate the permutation distribution[85,86]. Owing to the variability of total number of presynaptic neurons across brains, we introduced non-uniform sample weights when we computed the 2-Wasserstein distance to avoid any dominated effect from subjects having a large number of presynaptic neurons. Specifically, we first re-weighted every neuron by the inverse of the number of (for example, layer-wise/whole-wise) presynaptic neurons to ensure an equivalent contribution from subjects with non-empty neuron sets. Second, sample weights from subjects in the same group were concatenated and normalized into a probabilistic mass. This alleviates the imbalance of group-wise total mass if some of the subjects have no neurons detected in specific cortical layers (for instance, GABAergic presynaptic neurons in layer 6). To aid in the visualization of the spatial spread of presynaptic networks across cortical areas, we generated cortical flat maps. The 3D Allen Common Coordinate Framework coordinate points of registered neurons were projected along streamline paths orthogonal to the cortical surface to create a flat map 2D representation that preserves relative spatial position of cortical areas for analysis using tools provided by the Allen Institute[87].

**Statistical analysis of group-wise proportions of M1 and M2 presynaptic cells.** Motor cortical neurons that project to wS1 are distributed closely along the anatomical border between M1 and M2[88]. To evaluate

whether the movement-uncorrelated and movement-correlated groups exhibited an M1 or M2 bias in presynaptic cell proportions, in addition to the fraction of M1 and M2 cells, we computed the shortest 3D Euclidean distance of each M1/2 cell from the anatomical border between M1 and M2 (represented by an open surface mesh in the Allen Mouse Common Coordinate Framework). Distances were assigned with negative or positive values based on whether cells were in M1 or M2, respectively. Given the variability in cell counts across subjects, we implemented a reweighting procedure on the distance distribution, aiming at an equitable contribution of each subject to the group. This involved resampling cells based on a probability assigned to each cell, inversely proportional to the cell count within each subject, resulting in group-wise distribution of distances based on 10,000 resampled cells. Owing to a minimal number of M1/2 cells, four mice in the movement-uncorrelated and four mice in the movement-correlated groups were excluded. The observed mean difference was compared to a distribution of mean differences obtained by randomly permuting the group labels and recalculating the weighted mean difference for each permutation. This process was repeated 5,000 times to obtain the permutation distribution.

## Statistics

We did not use statistical methods to predetermine sample size. All data were acquired according to standard protocols, batch processed using the same codes, and independently processed and analysed by analysts blinded to the experimental groups. Analytical routines and were established using a subset of the data (training set), and these were applied to entire datasets, whenever applicable to avoid overfitting. Data were presented as mean ± s.d. throughout the text. In box plots, the central line represents the median, the box represents the 25th and 75th percentiles, and the whiskers extend to the most extreme data points excluding outliers (larger than 1.5× the interquartile range); when overlaid with individual data points, all data points, including outliers, were graphed. Statistical tests used were indicated in the figure legends. All comparisons using two-sample or paired-sample $t$-tests, Wilcoxon rank-sum or signed-rank tests were two-sided unless otherwise indicated. Linear fixed effects models (with interaction between drug type and neuronal correlations) were used to compare slopes (Fig. 2). Linear mixed effects models were used to test effect of receptor blocker application order. Bonferroni correction was applied to multiple comparisons unless otherwise indicated. Significance levels are indicated as: NS, not significant ($P \geq 0.05$); *$P < 0.05$; **$P < 0.01$; ***$P < 0.001$.

## Reporting summary

Further information on research design is available in the Nature Portfolio Reporting Summary linked to this article.

## Data availability

The data supporting the main findings of this study are available at https://doi.org/10.5281/zenodo.13926983 (ref. 89). Raw datasets are available from the corresponding authors upon reasonable request. Source data are provided with this paper.

## Code availability

The code for analysis is available at: https://github.com/NIMH-FNC/Brain-wide-presynaptic-networks.

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

**Acknowledgements** The authors thank all members of the S.L. laboratory, the C. McBain laboratory, the T. Petros laboratory and the W. Lu laboratory for helpful discussions; M. Han, J. Qi, W. Zhang for technical assistance with genotyping and histology; R. Paletzki for assistance with processing of whole-brain histological images; C. Scott Gerfen for generation of cortical flat maps; the NIMH Section on Instrumentation (G. Dold, D. Ide, J. Kim and T. Talbot) and the NIH IDEAS laboratory (L. Argueta, J. Krynitsky and T. Pohida) for providing custom hardware and software for data acquisition; the NIMH Systems Neuroscience Imaging Resource (J. Kuo, T. Usdin and S. Williams) for assistance with histological analyses; the NIMH Rodent Behavioral Core (Y. Chudasama) for providing surgical set-ups; the Center for Neuroanatomy with Neurotropic Viruses for kindly providing CVS-N2c rabies virus; and C. I. Baker, G. Fishell, R. Khazipov, D. Leopold, E. Merriam, B. Rudy and P. E. Rueda-Orozco for suggestions on the manuscript. This work was supported by the Intramural Research Program of the National Institute of Mental Health, National Institutes of Health (ZIAMH002959 to S.L., ZIC-MH002968 to F.P. and ZIAMH002497-35 to C.R.G.).

**Author contributions** A.R.I. and S.L. conceived the study and designed the experiments. A.R.I. performed the experiments. A.R.I., K.C.L., Y.Z., F.P., C.R.G. and S.L. analysed the data. A.R.I. and S.L. wrote the manuscript, with inputs from K.C.L., Y.Z., F.P. and C.R.G.

**Competing interests** The authors declare no competing interests.

**Additional information**
**Correspondence and requests for materials** should be addressed to Ana R. Inácio or Soohyun Lee.

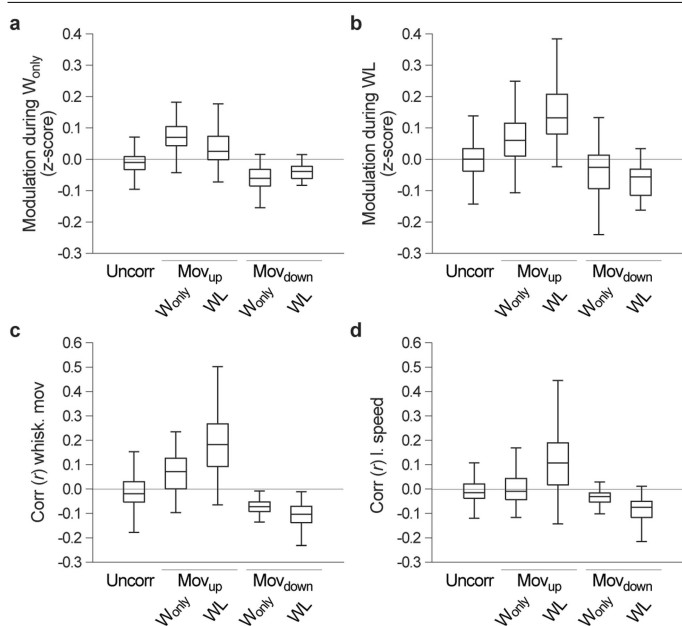

**Extended Data Fig. 1 | Classification of individual neurons based on their activity during spontaneous movements. a-d**, To increase neuronal classification accuracy, spontaneous movements were subdivided into two subtypes, $W_{only}$ (263 ± 98 events per session) and WL (50 ± 27 events per session). Neurons were considered as $Mov_{up}$ or $Mov_{down}$ if their activity changed significantly during either $W_{only}$ or WL. All other neurons were considered as movement-uncorrelated. Resulting subsets: movement-uncorrelated ($n$ = 715); $Mov_{up}$ ($W_{only}$, $n$ = 62; WL, $n$ = 224); $Mov_{down}$ ($W_{only}$, $n$ = 85; WL, $n$ = 38) (6 FOVs, 6 sessions, 5 mice). Neurons exhibiting a significant increase or decrease in activity across both spontaneous movement subtypes ($W_{only}$ + WL) were included in the $Mov_{up}$ (WL) or $Mov_{down}$ (WL) subsets, respectively. The proportions of $Mov_{up}$ neurons according to behavioral events were: 25 ± 19% for $W_{only}$, 41 ± 26% for WL, and 34 ± 19% for $W_{only}$ + WL. In boxplots, the central line and box represent the median and $25^{th}$-$75^{th}$ percentiles, and the whiskers extend to the most extreme data points excluding outliers (larger than 1.5 × the interquartile range). **a**, Modulation of all subsets during $W_{only}$. Note that $Mov_{up}$ ($W_{only}$) showed the highest modulation. **b**, Modulation of all subsets during WL. Note that $Mov_{up}$ (WL) showed the highest modulation (referred to as movement-correlated, corr., neurons). Note also that the $Mov_{up}$ ($W_{only}$) cell cluster still shows an increase in activity during WL compared to movement-uncorrelated cells, albeit typically more transient and less robust than that of $Mov_{up}$ (WL). **c-d**, Correlation ($r$) of neurons from the different subsets with whisker movements (**c**) and locomotion speed (**d**). Modulation and correlation values are largely coherent.

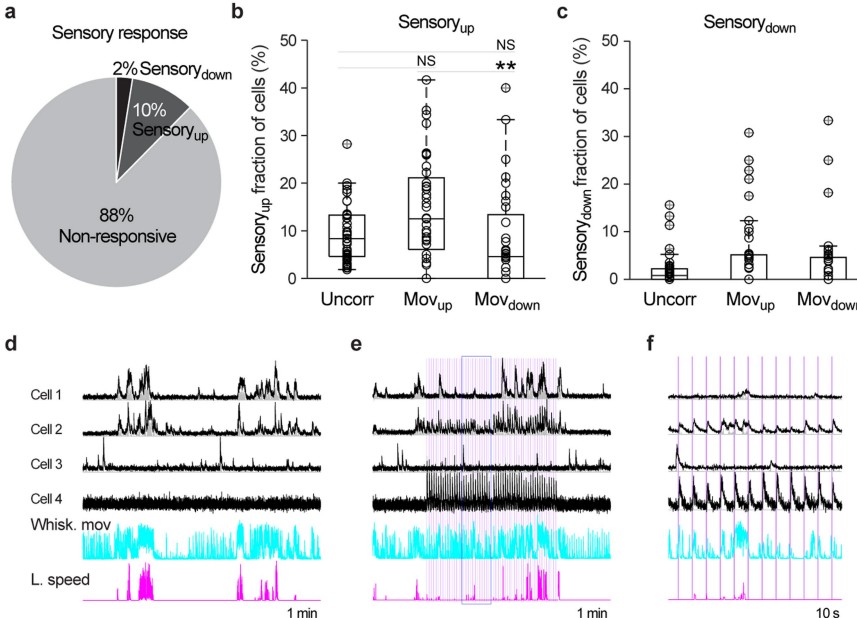

**Extended Data Fig. 2 | Responsiveness to sensory stimuli. a**, Fraction of sensory stimulus-responsive neurons from all recorded neurons irrespective of their activity pattern during spontaneous movements. Neurons responded by either increasing (Sensory_up, 801) or decreasing (Sensory_down, 236) their activity during stimulus presentation ($n$ = 8750 cells, 44 FOVs, 44 sessions, 33 mice). **b-c**, Fraction of sensory stimulus-responsive neurons within the movement-uncorrelated, Mov_up, and Mov_down subsets ($n$ = 8750 cells, 44 FOVs, 44 sessions, 33 mice). In boxplots, the central line and box represent the median and 25th-75th percentiles, and the whiskers extend to the most extreme data points excluding outliers (larger than 1.5 × the interquartile range).

**b**, Sensory_up neurons (uncorr. vs. Mov_up and uncorr. vs. Mov_down: $P > 0.05$; Mov_up vs. Mov_down: $P = 0.0031$; Kruskal-Wallis test ($P = 0.0046$) followed by multiple comparisons with Bonferroni correction). **c**, Sensory_down neurons ($P > 0.05$, Kruskal-Wallis test). **d-f**, Example neuronal activity during spontaneous movements and sensory stimulation. Example Mov_up (1 and 2) and movement-uncorrelated (3 and 4) neurons that either did not respond (1 and 3) or responded (2 and 4) to sensory stimulation. **d**, Absence of sensory stimulation. **e**, Presence of sensory stimulation. **f**, Expanded time window from **e** (blue box). Black, F; gray, F deconv.; both normalized to maximum. Purple vertical bars, individual sensory stimuli.

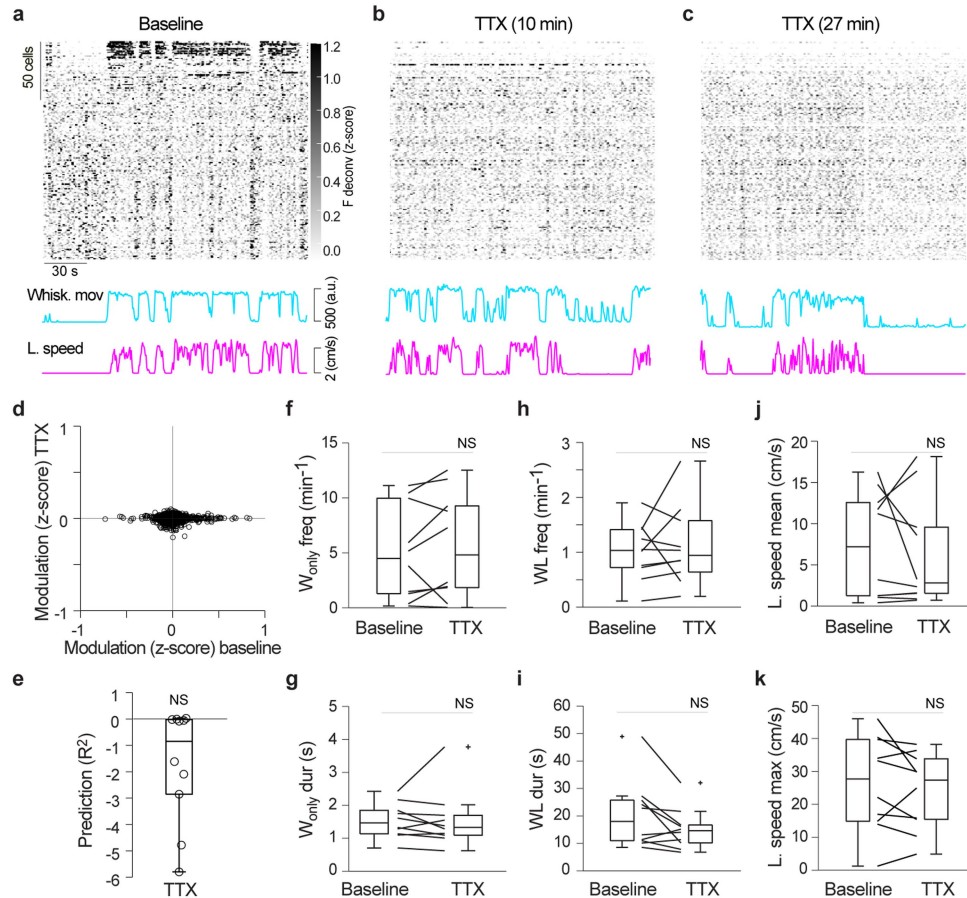

**Extended Data Fig. 3 | Suppression of cortical activity by TTX diffusion into the brain parenchyma using a custom cranial window. a-c**, Example session. Top, Raster plots of neuronal activity before and at different time points after application of TTX. Neurons (184) are sorted from top to bottom by decreasing weight on the first principal component. Bottom, Corresponding whisker movements and locomotion speed traces. **d**, Modulation of individual neurons (1853) during spontaneous movements ($W_{only}$ + WL) before vs. after application of TTX ($P > 0.05$, regression). **e**, Prediction of whisker movements from population activity. Linear decoder predictive $R^2$; the decoder was built using

baseline data and evaluated on TTX out-of-sample data ($n = 10$ FOVs, 10 sessions, 10 mice, $P > 0.05$, one-sided paired sample $t$-test). **f-k**, Spontaneous movement parameters before and after application of TTX: $W_{only}$ frequency (**f**) and duration (**g**), WL frequency (**h**) and duration (**i**), mean (**j**) and maximum (**k**) locomotion speed ($n = 10$ sessions, 10 mice, $P > 0.05$ for all panels, two-sided Wilcoxon signed-rank test). **f-k**, In boxplots, the central line and box represent the median and 25th-75th percentiles, and the whiskers extend to the most extreme data points excluding outliers (larger than $1.5 \times$ the interquartile range).

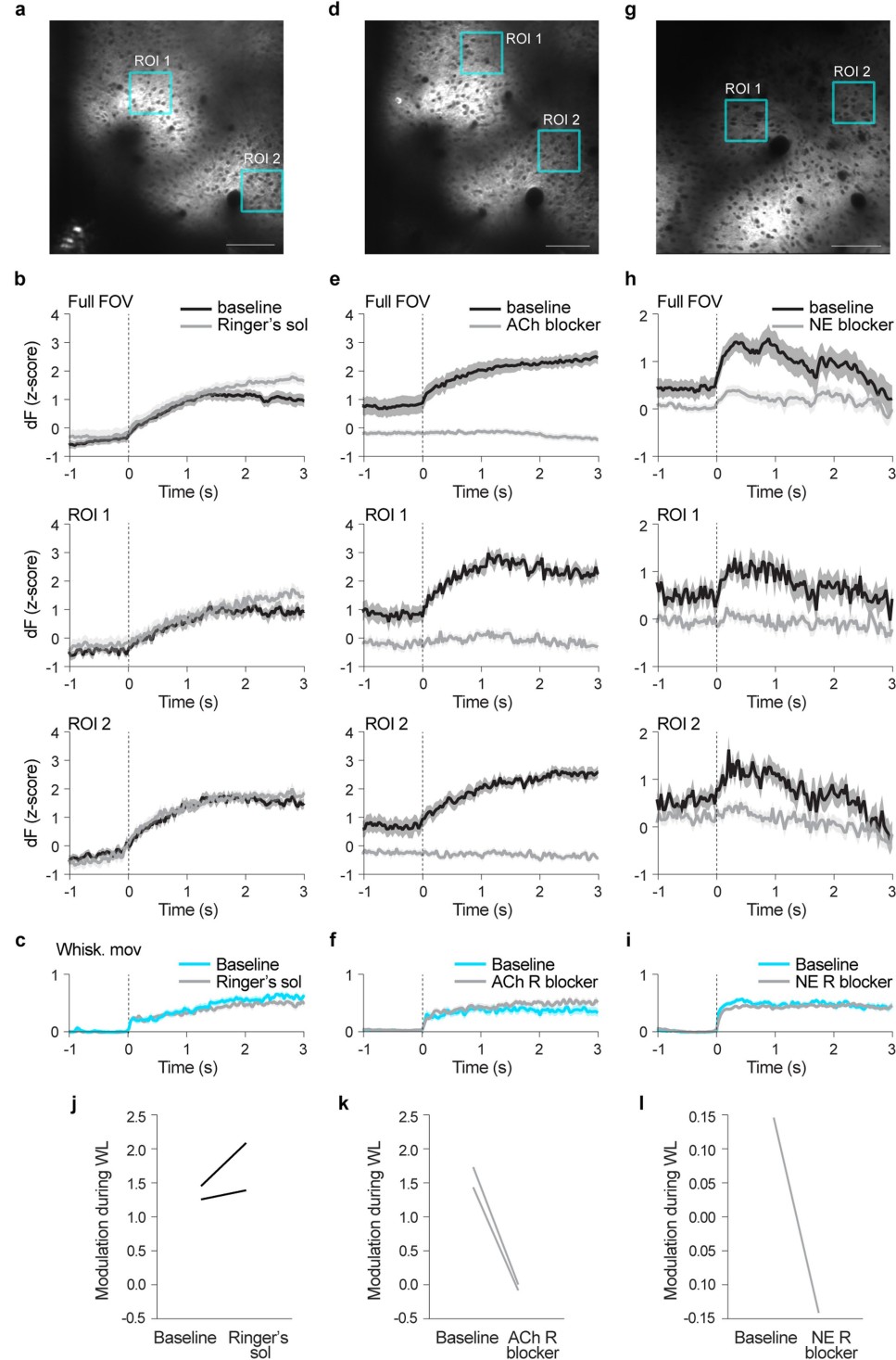

**Extended Data Fig. 4 | Effectiveness of local application of neuromodulatory receptor blockers through a cranial window port. a**, Example imaging FOV denoting the expression of the ACh sensor GRAB$_{ACh}$ in wS1 L2/3 neurons. **b**, ΔF averaged over the full FOV and two example ROIs, aligned to the onset of WL events (vertical bars) before and after reapplication of Ringer's solution in an example mouse (sham session). **c**, Corresponding whisker movements. **d-f**, As in **a-c**, but instead exemplifying the expression of GRAB$_{ACh}$ and data recorded before and after application of ACh R blockers. **g-i**, As in **a-c**, but instead exemplifying the expression of the norepinephrine (NE) sensor GRAB$_{NE}$ and data recorded before and after application of NE R blockers. **j-l**, Baseline-subtracted mean ΔF during WL events (modulation during WL) for sham ($n = 2$, **j**), ACh-R blockade ($n = 2$, **k**), and NE R blockade ($n = 1$, **l**) sessions. WL-related increases in ACh and NE were abolished when the respective R blockers were applied, but not in sham sessions. **a**, **d**, **g**, Scale bar, 100 μm. **b-c**, **e-f**, **h-f**, Results are presented as mean ± s.e.m.

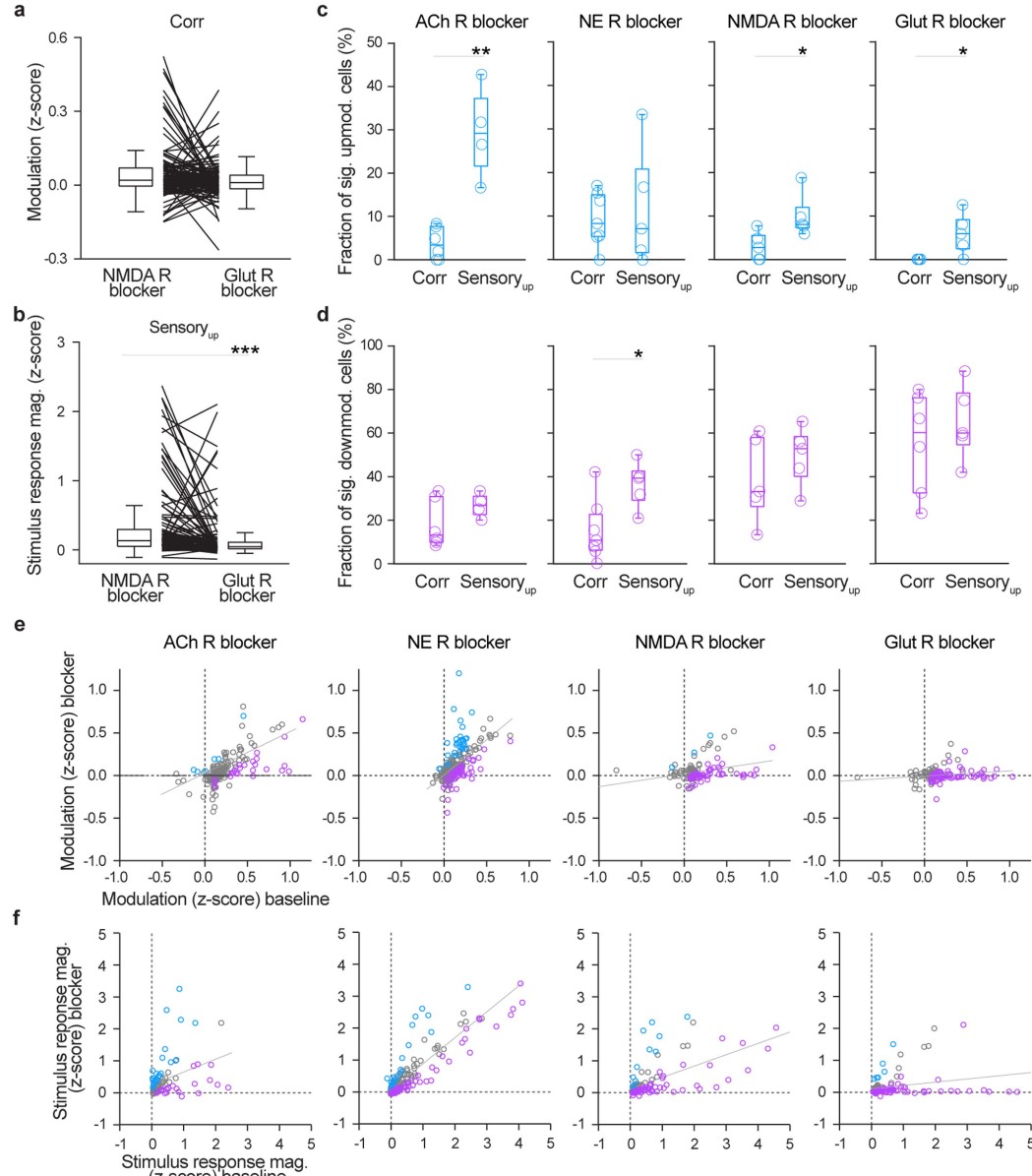

**Extended Data Fig. 5 | Effect of neuromodulatory inputs on the activity of wS1 L2/3 PNs. a-b,** Neurons were defined as movement-correlated or stimulus-responsive based on data acquired prior to antagonist application. **a,** Modulation of individual movement-correlated neurons (118) during spontaneous movements (WL) in the presence of an NMDA R blocker or Glut (glutamate) R blockers ($P > 0.05$, two-sided Wilcoxon signed-rank test). **b,** Sensory stimulus-response magnitude of individual stimulus-responsive neurons (115) in the presence of an NMDA R blocker or Glut R blockers ($P = 4.3 \times 10^{-7}$, two-sided Wilcoxon signed-rank test). Only neurons showing an increased activity following stimulus presentation were included (Sensory$_{up}$). **c,** Fraction of movement-correlated neurons that exhibited a significant increase in modulation during WL, and fraction of Sensory$_{up}$ neurons that exhibited a significant increase in stimulus response magnitude in the presence of either ACh (corr., $n = 6$ sessions, 6 mice; Sensory$_{up}$, $n = 4$ sessions, 4 mice; $P = 0.0095$), NE (corr., $n = 7$ sessions, 7 mice; Sensory$_{up}$, $n = 5$ sessions, 5 mice; $P > 0.05$), NMDA (corr. and Sensory$_{up}$, $n = 5$ sessions, 5 mice; $P = 0.024$) or Glut (corr., $n = 6$ sessions, 6 mice; Sensory$_{up}$, $n = 5$ sessions, 5 mice; $P = 0.030$) R blockers vs.

baseline (two-sided Wilcoxon rank-sum tests). **d,** Equivalent to **c**, but for movement-correlated and Sensory$_{up}$ neurons showing a decreased activity in the presence of R blockers ($n$ as in **c**; ACh R, $P > 0.05$; NE R, $P = 0.048$; NMDA R, $P > 0.05$; Glut R, $P > 0.05$; two-sided Wilcoxon rank-sum test). **e,** Modulation of individual movement-correlated neurons (118–273) during spontaneous movements (WL) in the presence of the different R blockers vs. baseline (ACh R, $P < 0.0001$; NE R, $P < 0.0001$; NMDA R, $P < 8.3 \times 10^{-4}$; Glut R, $P = 0.024$, regression). Purple, cells showing a significant increase. Blue, cells showing significant decrease. **f,** Stimulus response magnitude for individual Sensory$_{up}$ neurons (84–197) in the presence of the different R blockers vs. baseline (ACh R, $P < 9.5 \times 10^{-4}$; NE R, $P < 0.0001$; NMDA R, $P = 7.9 \times 10^{-12}$; Glut R, $P = 0.0090$, regression). The effect of neuromodulatory input blockade, in particular ACh, is more robust at the level of responses to sensory stimuli. **a-d,** In boxplots, the central line and box represent the median and 25th-75th percentiles, and the whiskers extend to the most extreme data points excluding outliers (larger than 1.5 × the interquartile range).

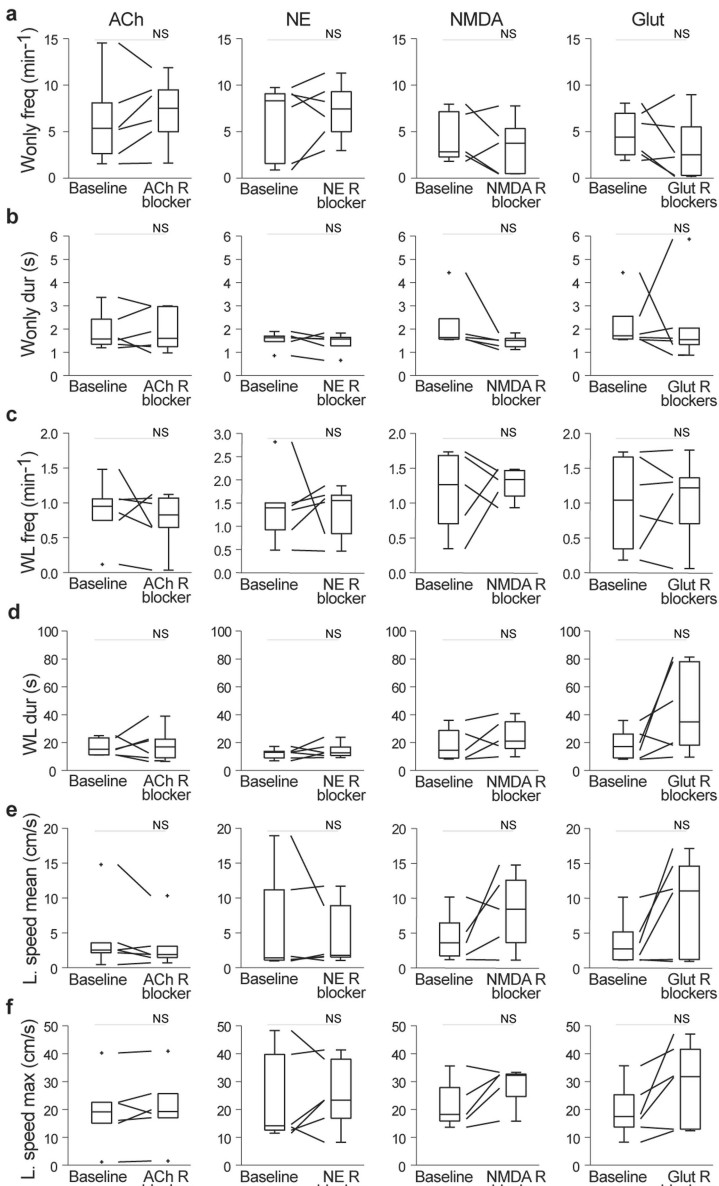

**Extended Data Fig. 6 | Effect of neuromodulatory and glutamatergic receptor blockade on spontaneous movements. a-f**, Spontaneous movement parameters: $W_{only}$ frequency (**a**) and duration (**b**), WL frequency (**c**) and duration (**d**), mean (**e**) and maximum (**f**) locomotion speed prior to (baseline) and during ACh, NE, NMDA or Glut R blockade (ACh, NE, and Glut R, $n = 6$ sessions, 6 mice; NMDA R, $n = 5$ sessions, 5 mice; $P > 0.05$ for all panels, two-sided Wilcoxon signed-rank test). In boxplots, the central line and box represent the median and 25th-75th percentiles, and the whiskers extend to the most extreme data points excluding outliers (larger than 1.5 × the interquartile range).

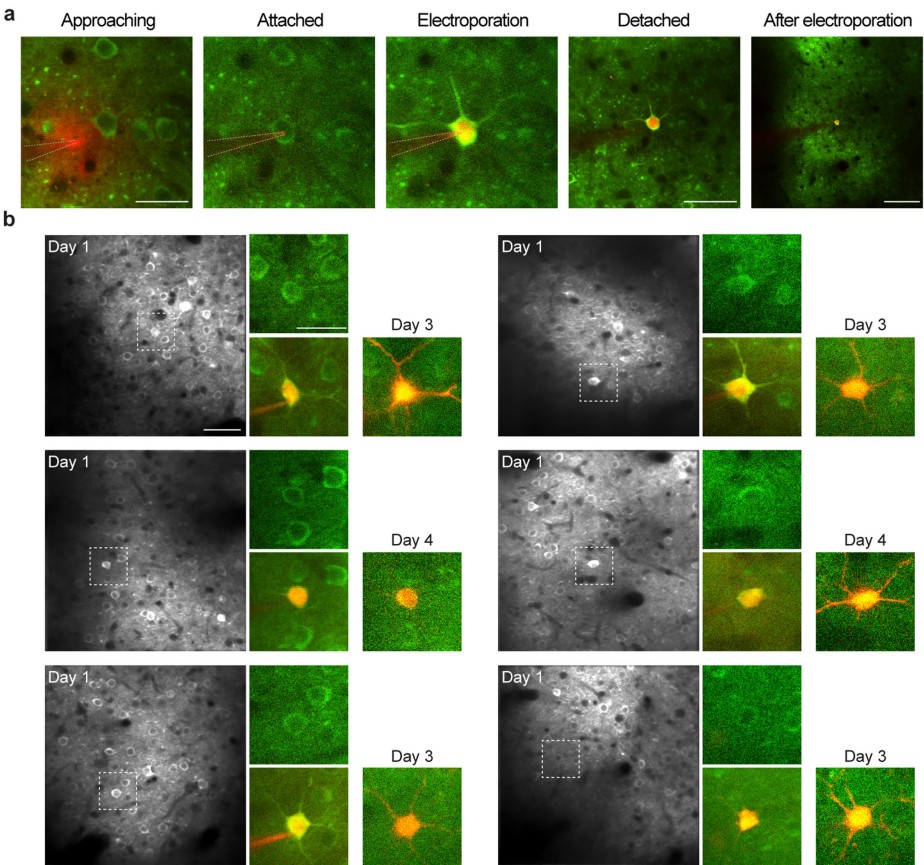

**Extended Data Fig. 7 | Electroporation of a functionally defined single neuron per brain. a**, Left to right, 2-PT guided approach of a PN in wS1 L2/3 with a pipette containing intracellular solution, Alexa 594 (red), and DNA (for TVA, G, and mCherry) (independent example demonstrative of the approach used to generate $n = 22$ postsynaptic neurons, 1 neuron per mouse). Principal neurons express GCaMP6 (green). Electroporation is indicated by the entry of Alexa 594 into the target cell. Images of the brain and target neuron at different magnifications. Scale bars, from left to right, 25, 50, and 100 μm. **b**, Example postsynaptic neurons ($n = 22$ neurons, 1 neuron per mouse). Left, Imaging FOV. Scale bar, 50 μm. Middle, High magnification 2-PT images of the postsynaptic neuron before (top) and immediately after (bottom, detached pipette) electroporation. Right, Image of the target neuron 3-4 days after electroporation (GCaMP6s⁺-mCherry⁺). Scale bar, 25 μm.

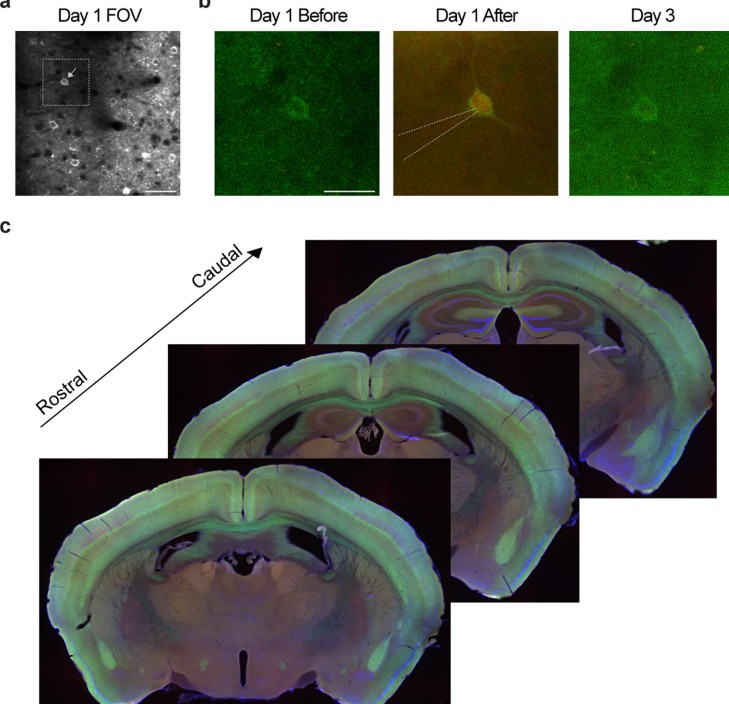

**a** Day 1 FOV

**b** Day 1 Before    Day 1 After    Day 3

**c** Caudal    Rostral

**Extended Data Fig. 8 | Absence of presynaptic labelling in sham single-cell-based monosynaptic input tracing experiments. a**, Imaging FOV for an example single-cell-based monosynaptic retrograde tracing experiment. The arrow indicates a cell selected for electroporation based on its modulation during spontaneous movements. Scale bar, 50 μm. **b**, Day 1, target cell (as indicated in **a**) before and after 2-PT guided electroporation with Alexa 594 and DNA (TVA, G, and mCherry). Note the entry of Alexa 594 into the target cell (red). Following electroporation, RV-RFP were injected close to the electroporated cell. Day 3, the target cell did not express the reporter protein mCherry, indicative of failed transfection. Note that we used a single plasmid encoding for TVA, G, and mCherry. Scale bar, 25 μm. **c**, Histological analysis of the example experiment (**a-b**). We did not observe the emergence of presynaptic cells ($n$ = 4 brains). Red, RFP. Green, GCaMP6s. Blue, NeuroTrace 435/455. Scale bar, 1 mm.

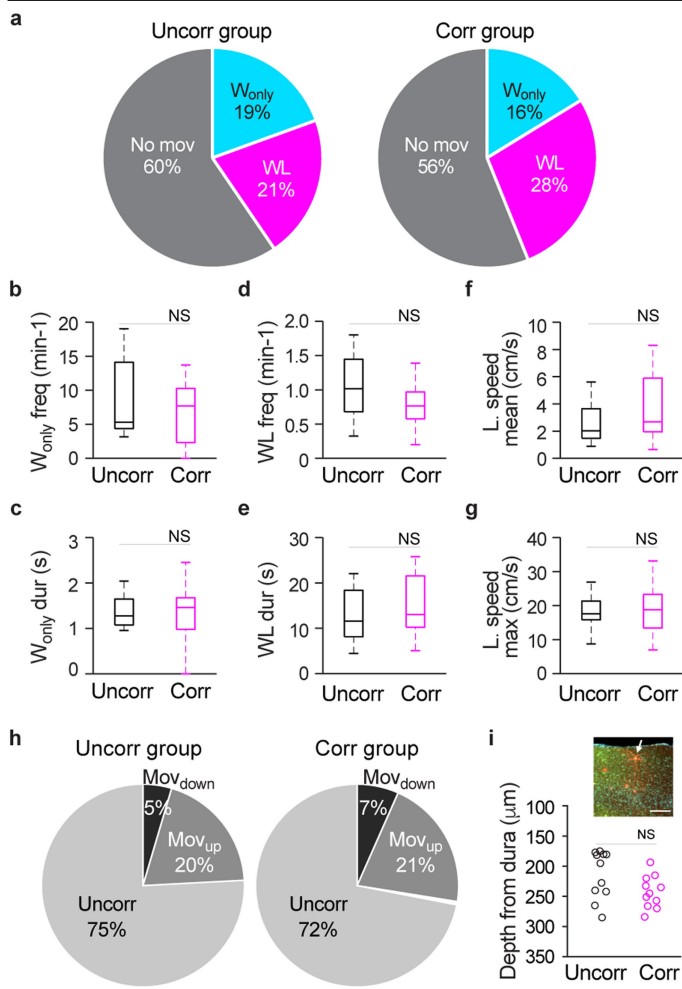

**Extended Data Fig. 9 | Spontaneous movements and neuronal properties of the movement-uncorrelated and movement-correlated groups for monosynaptic input tracing. a**, Pie chart of time spent per spontaneous movement type ($W_{only}$, WL) and not moving ($n_{uncorr.}$ and $n_{corr.}$ = 11 mice; $P > 0.05$ for uncorr. vs. corr., 2-way ANOVA). **b-g**, Spontaneous movement parameters: $W_{only}$ frequency (**b**) and duration (**c**), WL frequency (**d**) and duration (**e**), mean (**f**) and maximum (**g**) locomotion speed ($n_{uncorr.}$ and $n_{corr.}$ = 11 mice; $P > 0.05$ for **a-g**). In boxplots, the central line and box represent the median and $25^{th}$-$75^{th}$ percentiles, and the whiskers extend to the most extreme data points excluding outliers (larger than $1.5 \times$ the interquartile range). **h**, Fraction of movement-uncorrelated, $Mov_{down}$, and $Mov_{up}$ neurons per group ($n_{uncorr.}$ and $n_{corr.}$ = 11 mice; uncorr. vs. corr., $P > 0.05$, 2-way ANOVA). **i**, Cortical depth of each postsynaptic neuron. Depth was estimated starting from dura matter through in vivo 2-PT structural imaging ($n_{uncorr.}$ and $n_{corr.}$ = 11 mice; $P > 0.05$, two-sided Wilcoxon rank-sum test). Inset, epifluorescence image of a coronal brain slice encompassing wS1 and a postsynaptic neuron expressing both a red and green fluorescent protein (arrow, $n = 1$ neuron, 1 mouse; postsynaptic neurons typically survive do not until the experimental endpoint). Red, RFP. Green, GCaMP6s/GFP. Cyan, DAPI. Scale bar, 100 μm.

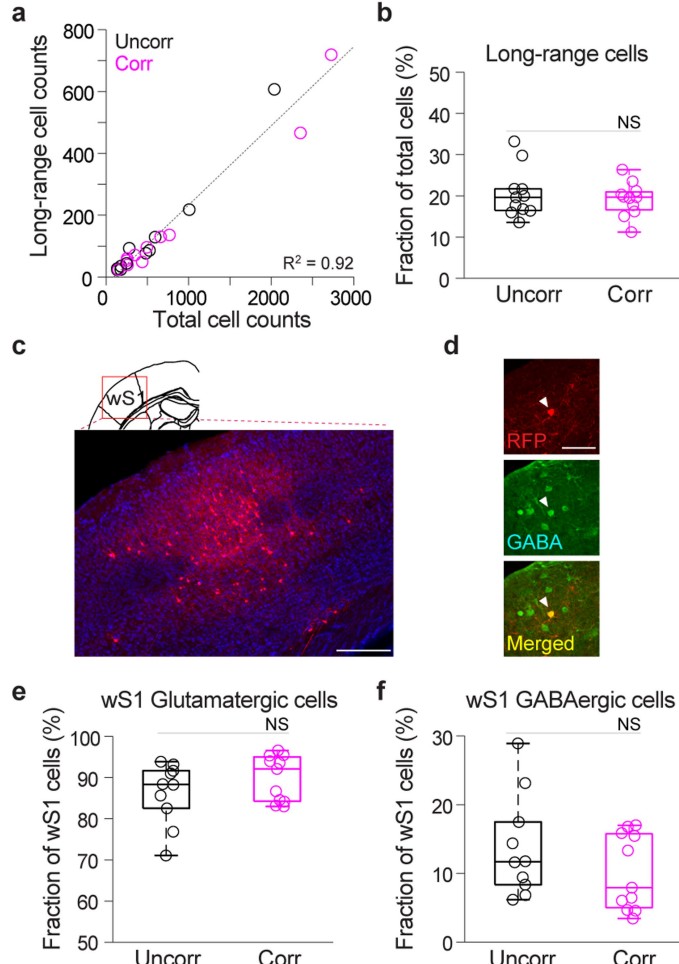

**Extended Data Fig. 10 | Long-range and local glutamatergic and GABAergic presynaptic neurons per brain. a**, Number of long-range vs. total presynaptic neurons per brain ($n_{uncorr.}$ and $n_{corr.}$ = 11 mice; $P < 0.0001$, regression). **b**, Long-range presynaptic neurons as fraction of total presynaptic neurons ($n_{uncorr.}$ and $n_{corr.}$ = 11 mice; $P > 0.05$, two-sided randomization test). **c**, Epifluorescence image of a coronal brain slice, wS1 (example presynaptic network included in Fig. 3a,d; $n = 22$ mice). Note the "hourglass" distribution of RFP⁺ presynaptic neurons across cortical L2/3, L4, and L5. Scale bar, 250 μm. **d**, Images denoting the co-localization of RFP (presynaptic neurons, Alexa 555, red) and GABA (GABAergic cells, Alexa 647, represented in green) in wS1 ($n = 22$ mice). Scale bar, 50 μm. **e-f**, Glutamatergic (**e**) and GABAergic (**f**) presynaptic neurons as fraction of wS1 presynaptic neurons ($n_{uncorr.}$ = 10 mice; $n_{corr.}$ = 11 mice; $P > 0.05$ for **e-f**, two-sided randomization tests). **b, e-f**, In boxplots, the central line and box represent the median and 25th-75th percentiles, and the whiskers extend to the most extreme data points excluding outliers (larger than 1.5× the interquartile range).

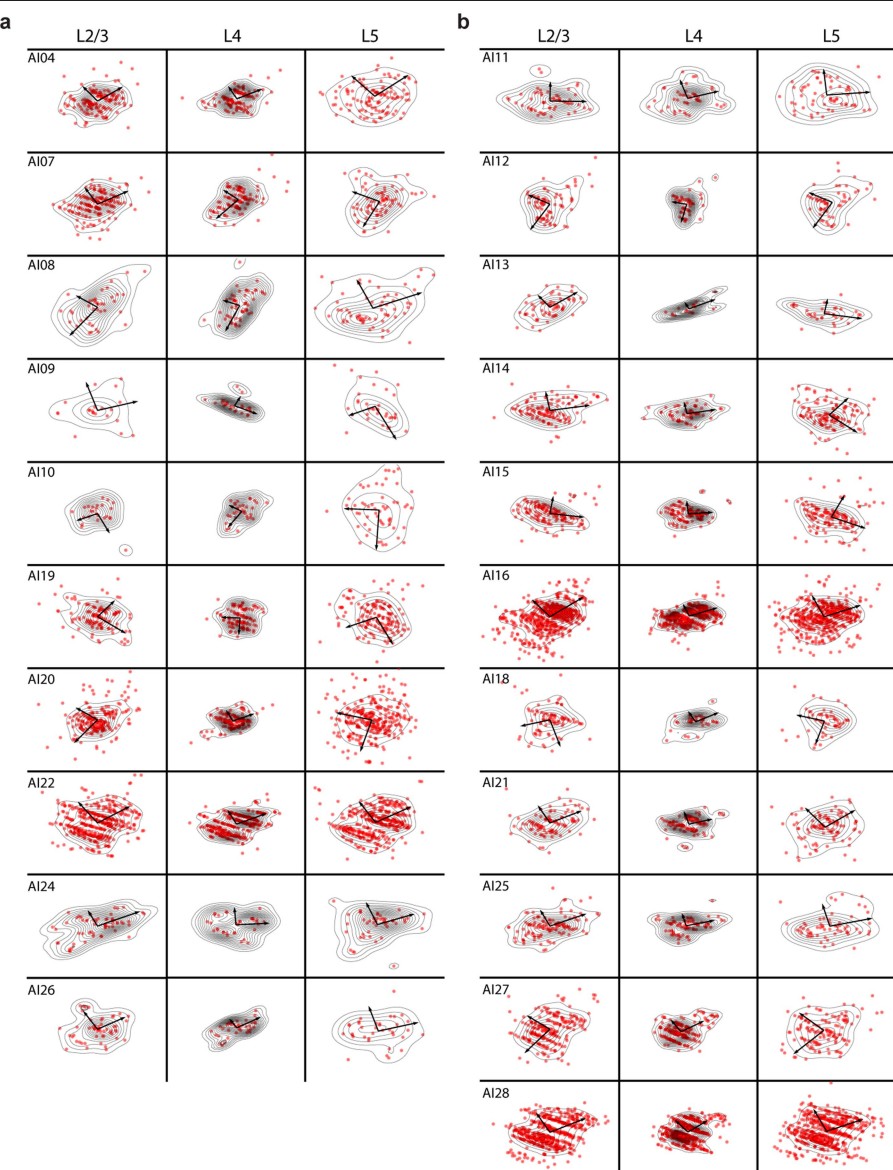

**Extended Data Fig. 11 | Spatial distribution of individual local glutamatergic presynaptic networks.** Layer-by-layer horizontal flat projections of individual wS1 glutamatergic presynaptic networks with gaussian kernel Density estimation. We use the determinant of the estimated covariance as a measure of presynaptic networks dispersion. Smaller spatial dispersion of glutamatergic presynaptic cells in L4 than in L2/3 ($P < 0.0001$) and L5 ($P < 0.0001$) for all brains, independently of group (uncorr. and corr., one-sided $t$-test). **a**, Movement-uncorrelated postsynaptic group. **b**, Movement-correlated postsynaptic group. Scale bar, 500 µm.

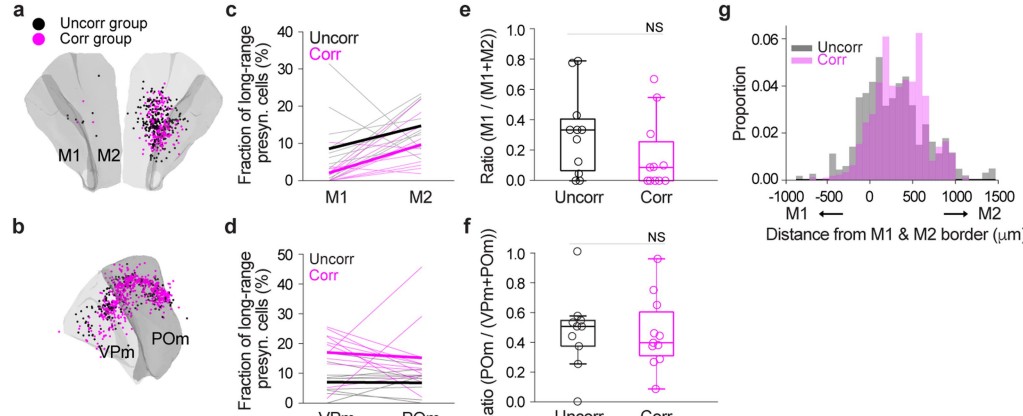

**Extended Data Fig. 12 | Motor cortical (M1 and M2) and thalamic (VPm and POm) presynaptic networks. a**, Three-dimensional distribution of all presynaptic neurons within motor cortex (M1 and M2) for the movement-uncorrelated and movement-correlated postsynaptic neuron groups (data aligned to the Allen Mouse Common Coordinate Framework; uncorr. vs. corr., $P > 0.05$, one-sided randomization test on 2-Wasserstein distance). **b**, As in **a**, but within the sensory thalamic nuclei, VPm and POm. The spatial distribution of presynaptic neurons of across the sensory thalamus are similar between the two groups (uncorr. vs. corr., $P > 0.05$, one-sided randomization test on 2-Wasserstein distance). **c**, M1 and M2 presynaptic neurons as fraction of long-range presynaptic neurons (uncorr. vs. corr.; M1, $P = 0.039$; M2, $P = 0.057$; two-sided Wilcoxon rank-sum test). Thin lines, individual brains; thick lines, ± s.e.m.

**d**, VPm and POm presynaptic neurons as fraction of long-range presynaptic neurons (uncorr. vs. corr.; VPm, $P = 0.0071$; POm, $P < 0.0151$; two-sided Wilcoxon rank-sum test). **e**, Relative proportion of M1 vs. M2 neurons ($P > 0.05$, two-sided Wilcoxon rank-sum test). **f**, Relative proportion of POm vs. VPm neurons ($P > 0.05$, two-sided Wilcoxon rank-sum test). **g**, Weighted distribution of motor cortical presynaptic neurons as function of distance from the M1 and M2 border ($n_{uncorr.} = 280$ cells; $n_{corr.} = 247$ cells; uncorr. vs. corr., $P > 0.05$, two-sided randomization test). **a-g**, Postsynaptic neuron groups ($n_{uncorr.}$ and $n_{corr.} = 11$ mice). **e-f**, In boxplots, the central line and box represent the median and 25th-75th percentiles, and the whiskers extend to the most extreme data points excluding outliers (larger than 1.5 × the interquartile range).

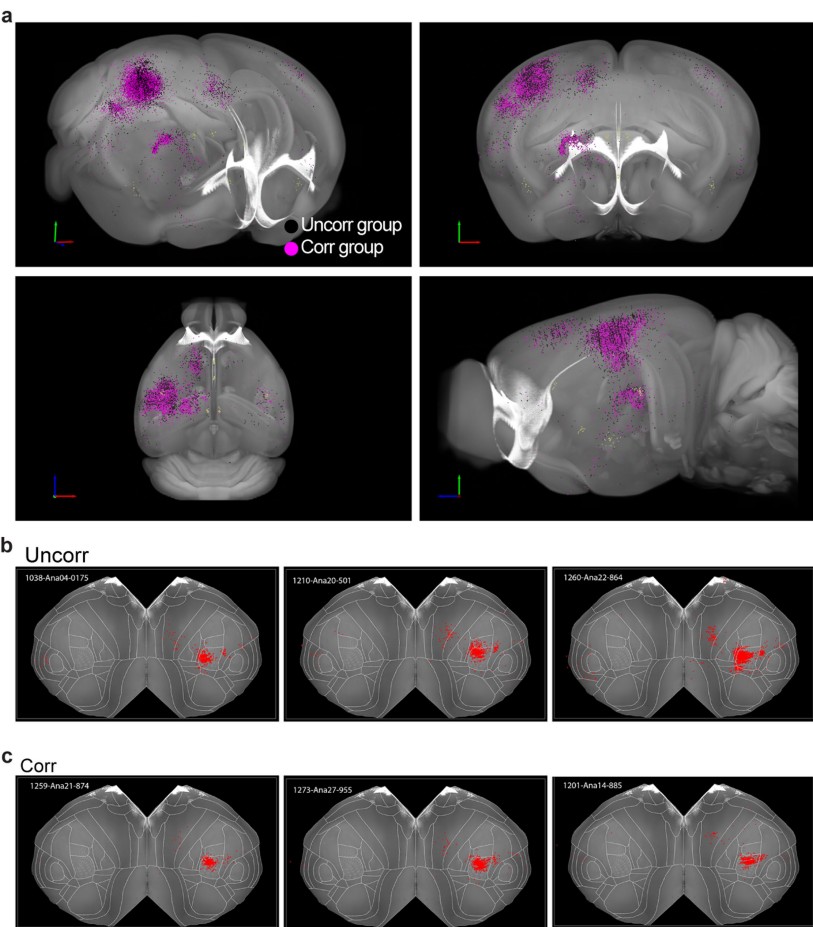

**Extended Data Fig. 13 | Brain-wide spatial distribution of presynaptic networks. a**, Presynaptic networks of the movement-uncorrelated ($n$ = 5801 cells, 11 networks, black) and movement-correlated ($n$ = 8699 cells, 11 networks, magenta) groups superimposed on the Allen Mouse Common Coordinate Framework, denoting the high degree of spatial overlap of presynaptic neurons from both groups ($P$ > 0.05 for each long-range area, one-sided randomization test on 2-Wasserstein distance). **b-c**, Cortical presynaptic networks from example movement-uncorrelated ($n$ = 3, **b**) and movement-correlated ($n$ = 3, **c**) brains (flat maps).

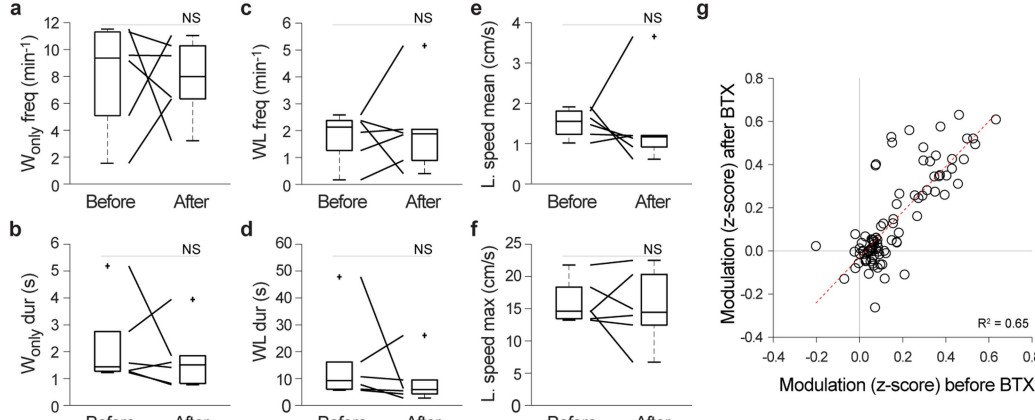

**Extended Data Fig. 14 | Neuronal activity and spontaneous movement before and after unilateral mystacial pad paralysis. a-f**, Spontaneous movement parameters, $W_{only}$ frequency (**a**) and duration (**b**), WL frequency (**c**) and duration (**d**), mean (**e**) and maximum (**f**) locomotion speed before and after BTX injection in the mystacial pad ($n$ = 6 FOVs, 2 sessions per FOV, 5 mice; $P$ > 0.05 for **a-f**, two-sided Wilcoxon signed-rank test). In boxplots, the central line and box represent the median and 25th-75th percentiles, and the whiskers extend to the most extreme data points excluding outliers (larger than 1.5 × the interquartile range). **g**, Modulation of movement-correlated neurons (93) during spontaneous movements (WL), before and after unilateral mystacial pad paralysis induced by BTX injection ($n$ = 6 FOVs, 2 sessions per FOV, 5 mice; $P$ < 0.0001, regression). **a-g**, $n$ = 6 FOVs, 2 sessions per FOV, 5 mice.

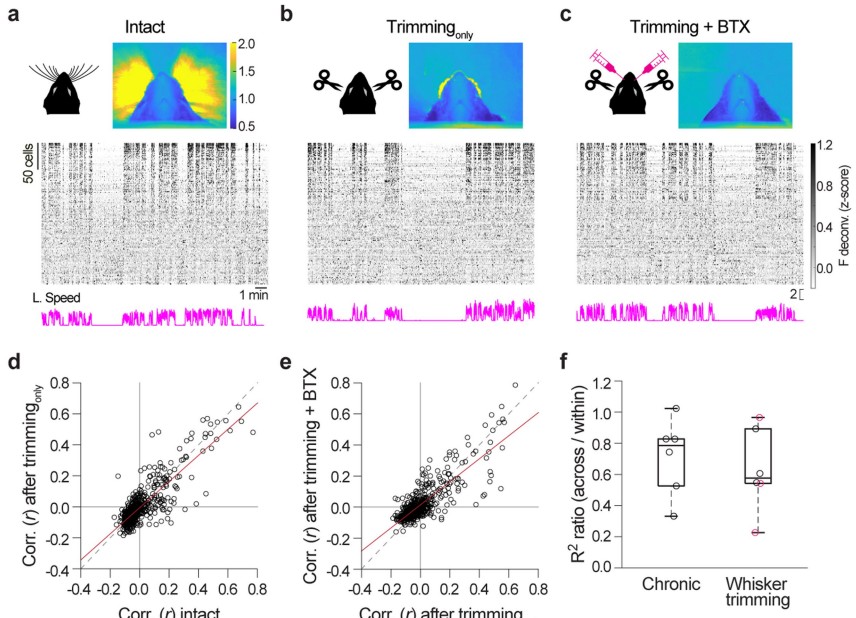

**Extended Data Fig. 15 | Effect of bilateral whisker trimming and whisker trimming combined with mystacial pad paralysis on wS1 L2/3 PNs activity in relation to spontaneous movements. a-c**, Top, Example images of video recorded whisker movements (mean of the absolute difference between consecutive frames over -14 s), before (intact, **a**) and after bilateral whisker trimming (Trimming$_{only}$, **b**), as well as combined bilateral whisker trimming and bilateral BTX injections (Trimming + BTX, **c**) in the same mouse. Bottom, Raster plots of neuronal activity during spontaneous movements and corresponding locomotion speed traces; neurons (153) tracked across the three imaging sections (intact, Trimming$_{only}$, and Trimming + BTX) are sorted from top to bottom by decreasing weight on the first principal component. **d**, Correlation (r) of the activity of individual neurons with locomotion speed

before (intact) vs. after Trimming$_{only}$ ($R^2 = 0.71$, $P < 0.0001$, regression). **e**, Correlation (r) of the activity of individual neurons with locomotion speed Trimming$_{only}$ vs. Trimming + BTX ($R^2 = 0.70$, $P < 0.0001$, regression). **f**, Prediction of locomotion speed from population activity. Out-of-sample $R^2$ ratio for chronic imaging (as in Fig. 1l), for intact vs. Trimming$_{only}$ (black circles), and for Trimming$_{only}$ vs. Trimming + BTX (red circles) (chronic vs. Trimming$_{only}$/ Trimming + BTX, $P > 0.05$, two-sided Wilcoxon rank-sum test). In boxplots, the central line and box represent the median and 25th-75th percentiles, and the whiskers extend to the most extreme data points excluding outliers (larger than $1.5 \times$ the interquartile range). **d-f**, $n = 690$ neurons, 3 FOVs, 2 sessions per FOV, 2 mice.

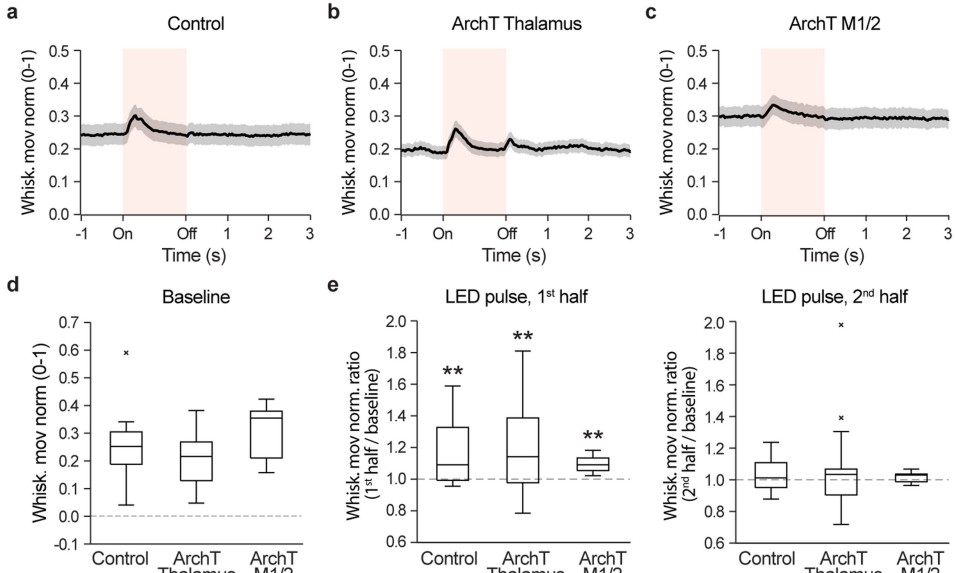

**Extended Data Fig. 16 | Behavioral effect of light presentation. a-c**, Effect of light presentation on the spontaneous movements of control ($n = 9$, **a**), ArchT thalamus ($n = 7$, **b**), and ArchT M1/2 ($n = 3$, **c**) mice. Whisker movements aligned to the onset of light stimulation (mean ± s.e.m.). **d**, Whisker movements during baseline ($n$ as in **a-c**; 0.5 s window prior to light onset; $P = 0.048$, Kruskal-Wallis test across mouse groups). **e**, Quantification of behavioral changes over the first part (left, 0–0.5 s) and second part (right, 0.5–1/1.5 s) of the light pulse relative to baseline. All groups showed a brief (first half) increase in whisker movements locked to light presentation onset ($n$ as in **a-c**; control: $P < 0.0072$; ArchT thalamus: $P < 0.0057$; ArchT M1/2: $P < 0.0020$; one-sided Wilcoxon

signed-rank test whisk. mov. norm ≠ 1 vs. whisk. mov. norm = 1), and this increase was comparable across groups ($n$ as in **a-c**; $P > 0.05$, Kruskal-Wallis test). Whisker movements in the second half of the light pulse did not differ from that of baseline ($P > 0.05$ for each mouse group, one-sided Wilcoxon signed-rank test whisk. mov. ratio ≠ 1 vs. whisk. mov. ratio = 1) and were similar across groups ($n$ as in **a-c**; $P > 0.05$, Kruskal-Wallis test). **d-e**, In boxplots, the central line and box represent the median and 25th-75th percentiles, and the whiskers extend to the most extreme data points excluding outliers (larger than 1.5 × the interquartile range).

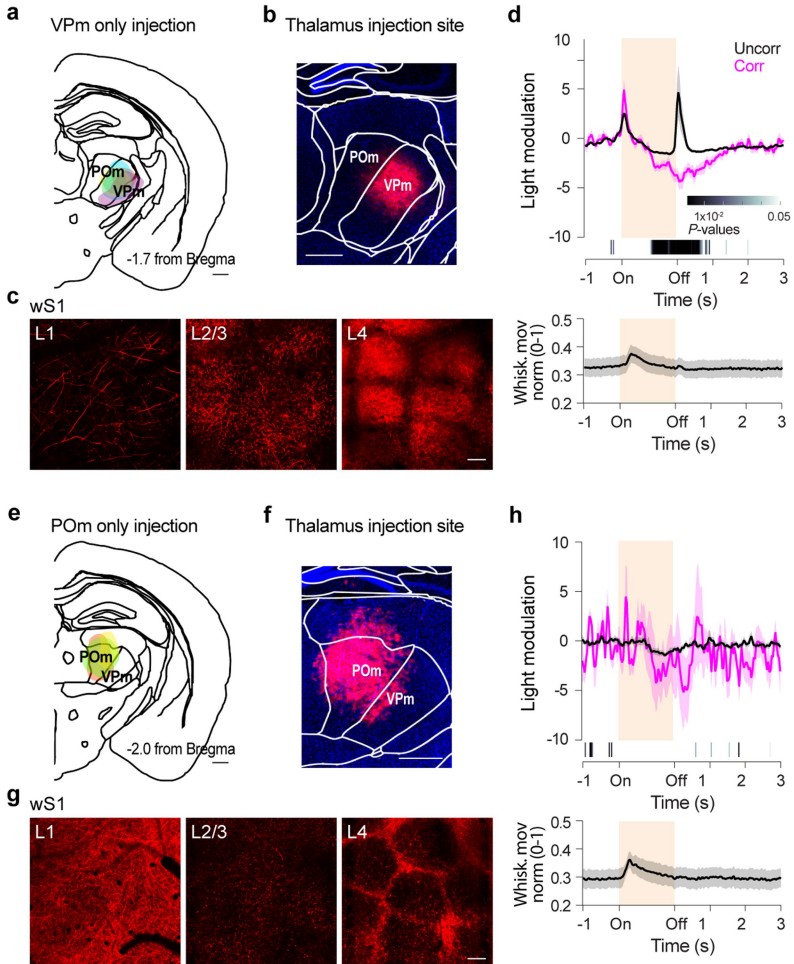

**Extended Data Fig. 17 | Optogenetic suppression of VPm and POm inputs in wS1 L2/3. a**, Expression of the opsin ArchT in VPm. ArchT⁺-tdTom⁺ areas are overlayed on the mouse atlas section corresponding to the center of the injection for VPm. Each shading color represents an individual mouse (*n* = 5). Scale bar, 0.5 mm. **b**, Example composite epifluorescence image of a coronal brain section showing ArchT⁺-tdTom⁺ cell bodies in VPm. Scale bar, 0.5 mm. **c**, Laminar distribution of VPm ArchT⁺-tdTom⁺ axons in wS1, as demonstrated by example 2-PT images. The characteristic patterns of innervation confirm the selectivity of injection during imaging, prior to histological analysis.

Scale bar, 0.1 mm. **d**, Baseline-subtracted mean activity (light modulation) of movement-uncorrelated (black) or movement-correlated (magenta) neurons significantly affected by light pulses (mean ± s.e.m.; *P* values, uncorr. vs. corr, two-sided Wilcoxon rank-sum test; *n* = 174 of 1108 uncorr. and *n* = 72 of 338 corr., 6 mice). Note that the movement-uncorrelated group includes 8 neurons (of 174) with a strong rebound activity; analysis with and without these neurons did not affect the results. **e-h**, As in a-d, but for the expression of the opsin ArchT in POm (neurons significantly changed by light stimulation, *n* = 32 of 585 uncorr. and *n* = 17 of 267 corr., 3 mice).

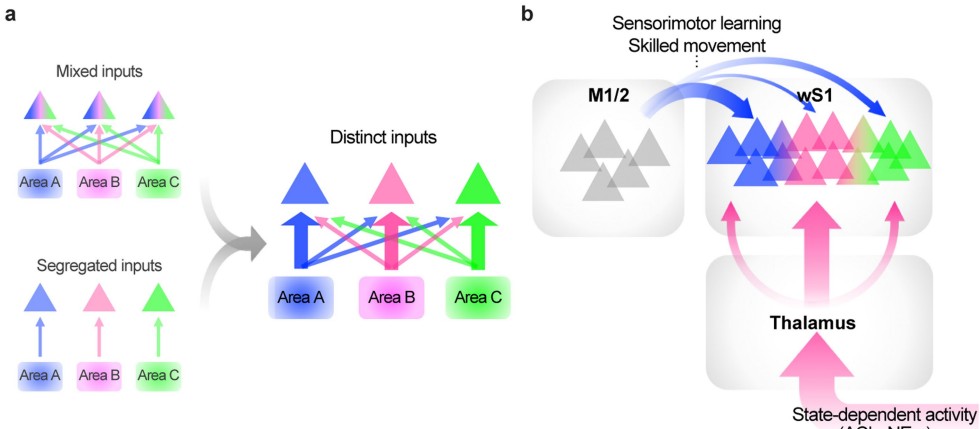

**Extended Data Fig. 18 | Summary of the anatomical presynaptic connectivity rule and its proposed functional implications. a**, Presynaptic connectivity rule underlying the functional heterogeneity of cortical PNs. Movement-uncorrelated and movement-correlated neurons did not receive separate sets of inputs nor completely random or mixed inputs from the distinct wS1-projecting areas. Instead, movement-uncorrelated and movement-correlated neurons had distinct presynaptic networks characterized by a selective fractional decrease in M1/2 inputs and increase in thalamic inputs. **b**, Hypothetical functional significance of distinct long-range inputs to wS1. While thalamic inputs may convey behavioral state-dependent activity, the M1/2 projection may instead be required for more complex changes in wS1 activity structure, such as establishment of new sensorimotor associations or acquisition of skilled movements.

# Reporting Summary

## Statistics

For all statistical analyses, confirm that the following items are present in the figure legend, table legend, main text, or Methods section.

| n/a | Confirmed | |
|---|---|---|
| ☐ | ☒ | The exact sample size (*n*) for each experimental group/condition, given as a discrete number and unit of measurement |
| ☐ | ☒ | A statement on whether measurements were taken from distinct samples or whether the same sample was measured repeatedly |
| ☐ | ☒ | The statistical test(s) used AND whether they are one- or two-sided<br>*Only common tests should be described solely by name; describe more complex techniques in the Methods section.* |
| ☐ | ☒ | A description of all covariates tested |
| ☐ | ☒ | A description of any assumptions or corrections, such as tests of normality and adjustment for multiple comparisons |
| ☐ | ☒ | A full description of the statistical parameters including central tendency (e.g. means) or other basic estimates (e.g. regression coefficient) AND variation (e.g. standard deviation) or associated estimates of uncertainty (e.g. confidence intervals) |
| ☐ | ☒ | For null hypothesis testing, the test statistic (e.g. *F*, *t*, *r*) with confidence intervals, effect sizes, degrees of freedom and *P* value noted<br>*Give P values as exact values whenever suitable.* |
| ☒ | ☐ | For Bayesian analysis, information on the choice of priors and Markov chain Monte Carlo settings |
| ☐ | ☒ | For hierarchical and complex designs, identification of the appropriate level for tests and full reporting of outcomes |
| ☐ | ☒ | Estimates of effect sizes (e.g. Cohen's *d*, Pearson's *r*), indicating how they were calculated |

*Our web collection on statistics for biologists contains articles on many of the points above.*

## Software and code

Policy information about availability of computer code

| | |
|---|---|
| Data collection | Two-photon calcium imaging data was acquired using a commercially available system, including software (PrarieView, 5.4 and 5.5), from Bruker.<br>Behavioral data was acquired using custom-written codes in LabVIEW (2017 and 2019, National Instruments) and the high speed digital video recording software StreamPix (7 and 9, NorPix).<br>Behavioral and neuronal data were synchronized through the Bruker system.<br>Histological images were acquired using Neurolucida (v2019-2021, MBF Bioscience), and software from Nikon (NIS-Elements 5.20.01) and Zeiss (ZEN 2.3 and 3.1), as specified throughout the Methods. |
| Data analysis | Two-photon calcium images were processed using Suite2p (0.10.1), in Python (Pachitariu et al., BioRxiv, 2017; reference provided in article).<br>Behavioral video recordings were preprocessed using custom MATLAB (R2020a and R2022a) routines.<br>Neural and behavioral signals were analyzed using custom-written MATLAB (R2020a and R2022a) or Python (3.7.6 and 3.11.5) routines, including the Scikit-learn library (1.3.0; Pedregosa et al., Journal of Machine Learning Research, 2011; reference provided in article).<br>Histological analyses were performed using NeuroInfo (2019-2023, MBF Bioscience), as well as ImageJ (1.52p, NIH).<br>Analysis of presynaptic networks was performed using Python (3.8.19), including the following library: POT Python Optimal Transport library (0.9.3, Villani, Topics in Optimal Transportation, 2021; reference provided in article).<br><br>The code for analysis is available at: https://github.com/NIMH-FNC/Brain-wide-presynaptic-networks. |

For manuscripts utilizing custom algorithms or software that are central to the research but not yet described in published literature, software must be made available to editors and reviewers. We strongly encourage code deposition in a community repository (e.g. GitHub). See the Nature Portfolio guidelines for submitting code & software for further information.

## Data

Policy information about availability of data

All manuscripts must include a data availability statement. This statement should provide the following information, where applicable:

- Accession codes, unique identifiers, or web links for publicly available datasets
- A description of any restrictions on data availability
- For clinical datasets or third party data, please ensure that the statement adheres to our policy

> The datasets are available from the corresponding authors upon reasonable request.

## Research involving human participants, their data, or biological material

Policy information about studies with human participants or human data. See also policy information about sex, gender (identity/presentation), and sexual orientation and race, ethnicity and racism.

| | |
|---|---|
| Reporting on sex and gender | N/A |
| Reporting on race, ethnicity, or other socially relevant groupings | N/A |
| Population characteristics | N/A |
| Recruitment | N/A |
| Ethics oversight | N/A |

Note that full information on the approval of the study protocol must also be provided in the manuscript.

# Field-specific reporting

Please select the one below that is the best fit for your research. If you are not sure, read the appropriate sections before making your selection.

☒ Life sciences ☐ Behavioural & social sciences ☐ Ecological, evolutionary & environmental sciences

For a reference copy of the document with all sections, see nature.com/documents/nr-reporting-summary-flat.pdf

# Life sciences study design

All studies must disclose on these points even when the disclosure is negative.

| | |
|---|---|
| Sample size | We did not use statistical methods to predetermine sample size. Our sample sizes were estimated based on previous publications using a similar methodology (Velez-Fort et al., Neuron, 2014; Wertz et al., Science, 2015; Rossi et al., Nature, 2020). |
| Data exclusions | Inclusion and exclusion criteria were established a priori. Experiments in which TTX did not silence neuronal activity over the entire FOV within 15-20 min, indicative of limited drug diffusion, were excluded (n = 2 mice). Brains containing less than 100 presynaptic cells were excluded from analysis (n = 1). |
| Replication | Results were reliably replicated across mice, as evidenced by the individual data points included in the Figures. Analytical routines and statistical tests were established using a subset of the data and then applied to entire datasets. |
| Randomization | Animals in test and control groups were littermates and randomly selected. In in vivo neuropharmacological experiments, receptor blocker application session sequences were randomly assigned across animals. In optogenetic experiments, light pulses (1-1.5 s) were randomly provided during the recording session. |
| Blinding | All data were acquired according to standard protocols, batch processed using the same codes, and independently processed and analyzed by analysts blinded to the experimental groups. In addition, manual and semi-automatic routines involving ROI curation in two-photon calcium imaging data, histological analysis, and classification of presynaptic neurons were done blindly, as specified in the Methods. |

# Reporting for specific materials, systems and methods

We require information from authors about some types of materials, experimental systems and methods used in many studies. Here, indicate whether each material, system or method listed is relevant to your study. If you are not sure if a list item applies to your research, read the appropriate section before selecting a response.

## Materials & experimental systems

| n/a | Involved in the study |
|---|---|
| ☐ | ☒ Antibodies |
| ☒ | ☐ Eukaryotic cell lines |
| ☒ | ☐ Palaeontology and archaeology |
| ☐ | ☒ Animals and other organisms |
| ☒ | ☐ Clinical data |
| ☒ | ☐ Dual use research of concern |
| ☒ | ☐ Plants |

## Methods

| n/a | Involved in the study |
|---|---|
| ☒ | ☐ ChIP-seq |
| ☒ | ☐ Flow cytometry |
| ☒ | ☐ MRI-based neuroimaging |

## Antibodies

| Antibodies used | Antibodies were used to detected red fluorescent proteins (RFP) and γ-Aminobutyric acid (GABA) in formaldehyde-fixed coronal brain sections.<br>Primary antibodies and respective dilutions:<br>- Chicken anti-RFP (600-901-379, Rockland), 1:500,<br>- Rabbit anti-GABA (A2052, Sigma), 1:500 .<br>Secondary antibodies and respective dilutions:<br>- Goat anti-chicken IgY-Alexa Fluor 555 (A21437, Thermo Fisher Scientific), 1:200,<br>- Goat anti-rabbit IgG-Alexa Fluor 647 (A21245, Thermo Fisher Scientific), 1:200 . |
|---|---|
| Validation | We used primary antibodies that have been validated by the manufacturers and are widely used in the field, as reflected by the large number of product citations including similar histological applications.<br><br>Anti-RFP manufacturer notes.<br>Applications. ELISA, SDS-PAGE, WB, FC, IF, IHC.<br>Purity/Specificity. RFP Antibody was prepared from egg yolks by a multi-step process which includes filtration, delipidation, salt fractionation and extensive dialysis against the buffer stated above. RFP Antibody was tested by western blot.<br><br>Anti-GABA manufacturer notes.<br>Technique(s). Dot blot: 1:10,000; immunohistochemistry (formalin-fixed, paraffin-embedded sections): 2.5 µg/mL using rat cerebellum.<br>General description. Anti-GABA is produced in rabbit using GABA-BSA as the immunogen. The antibody is isolated from antiserum by immunospecific methods of purification. Antigen specific affinity isolation removes essentially all rabbit serum proteins, including immunoglobulins which do not specifically bind to GABA.<br>Application. Expression of GABA in neucortical cells harvested from the brains of E19 day old rat embyros was detected by immunofluorescence using rabbit anti-GABA antibody. Triple IF staining was performed with the anti-GABA antibody and two anti-GAD antibodies. Expression of GABA was analyzed in cells isolated from the pallium of various animals including rats, mice, rabbits, guinea pigs, and lizards by immunohistochemistry. IHC was performed using rabbit anti-GABA antibody at 1:1000 diluted in a solution of 0.01M PBS ph 7.4 + 0.5% triton-x100.<br><br>Further details on the validation procedures and product citations can be found at:<br>- https://www.rockland.com/categories/primary-antibodies/rfp-antibody-600-901-379 (49 references),<br>- https://www.sigmaaldrich.com/US/en/product/sigma/a2052 (562 references). |

## Animals and other research organisms

Policy information about studies involving animals; ARRIVE guidelines recommended for reporting animal research, and Sex and Gender in Research

| Laboratory animals | We used the following transgenic mouse lines: Emx1-IRES-Cre (JAX 005628), tetO-GCaMP6s (JAX 024742), and CaMK2a-tTA (JAX 007004). We performed experiments on 10 Emx1-IRES-Cre, 71 tetO-GCaMP6s;CaMK2a-tTA, and 3 GCaMP6s+;CaMK2atTA- mice that were generated from the parental lines described above. Mice were 12-24 weeks old at the experimental endpoint. They were housed in groups, in individually ventilated and enriched laboratory cages, in climate-controlled rooms (T, 22 ºC; humidity, 45%), under a reverse 12 h light - 12 h dark cycle, and with ad libitum access to water and food. |
|---|---|
| Wild animals | No wild animal was used in the study. |
| Reporting on sex | We used both male and female mice (females, 45%). This study was not designed to identify sex-specific differences. Our sample sizes are not suitable to account for sex-dependent differences in the stability of the neuronal activity patterns, effect of ascending neuromodulatory systems or presynaptic connectivity motifs in the context of behavioral state. Nevertheless, overall, the percentage of Movup and Movdown neurons was similar between the male and female groups (30 ± 11% vs. 33 ± 11 %, n = 26 vs. 21 animals, respectively, P = 0.27, Wilcoxon rank-sum test). Similarly, the percentage of sensory-responsive neurons did not differ between males and females (13 ± 5.9% vs. 12 ± 6.1%, n = 19 vs. 14 animals, respectively; P = 0.57, Wilcoxon rank-sum test). |
| Field-collected samples | No field-collected samples were used in the study. |

Ethics oversight

All animal procedures were conducted in accordance with a protocol approved by the National Institutes of Health Institutional Animal Care and Use Committee (IACUC), Bethesda, MD, USA, and complied with the Public Health Service Policy on Humane Care and Use of Laboratory Animals and the Guide for the Care and Use of Laboratory Animals.

Note that full information on the approval of the study protocol must also be provided in the manuscript.

