## [Peer Review File · Nature]

Brain-wide presynaptic networks of functionally distinct cortical neurons

Corresponding Author: Dr Ana Inacio

Version 0:

Reviewer comments:

Referee #1

(Remarks to the Author)

Nature manuscript 2023-05-07457

Distinct brain-wide presynaptic networks underlie the functional identity of individual cortical neurons

In their study, Inácio and co-workers investigate in the primary somatosensory (S1) cortex the relationship between the selectivity of layer 2/3 (L2/3) pyramidal neurons (PN) to sensory stimuli and their presynaptic connectivity. The authors approach this question with an impressive number of techniques including two-photon calcium imaging, neuropharmacology, monosynaptic input tracing, and optogenetic activation and inactivation of brain regions. They authors observed spontaneous movements during their in vivo experiments: 1. short whisker movements without locomotion and longer/large amplitude whisker movement accompanied by locomotion (WL). The activity of these neurons is unregulated during locomotion while that of another fraction showed no change in activity. The authors aimed at studying the underlying principle of the relationship between whisker movement sponanteous locomotion in the L2/3 PNs of the S1 cortex. They found that neuronal activity in response to different behavioural states is stable over time and claim that it is dependent on neuromodulation by e.g. acetylcholine or noradrenaline. Rather, this activity patterns is solely dependent on glutamatergic inputs on a differential innervation by afferents from other cortical areas (such as the motor cortex) and the thalamus.

I have several concerns with this study which I will address point by point.

Specific points

Page 3, line 78-82

The authors use the terminology up- and down-modulated here although the later claim that 'neuromodulation' did not affect the relationship between locomotion and whisker movement. To avoid confusion, this should be amended.

Page 3, line 85-87

Was the number of active neurons dependent on stimulus strength? Did the authors check whether it was different in males or females or dependent on the day/night cycle? This is relevant for the claim that neuromodulation is not effective in changing the whisker movement/spont. locomotion relationship.

Page 5, line 137 to Page 6, line 138

There is ample evidence from both in vivo and in vitro studies that neurons in all layers of the neocortex including sensory cortices are subject to neuromodulation by acetylcholine, noradrenaline, dopamine and several neuropeptides, e.g. orexin although several of these are located in deeper cortical layers. The authors chose not to quote these studies. I think this should be amended

Page 5, line 139-152

The approach to test the relevance of neuromodulators for the activity patterns appears to be somewhat crude and does not take into account that the drug diffusion (i.e the ACh, NE and glutamate receptor antagonists) is likely to be highly non-uniform and affected primarily superficial but not deep layer PNs in whisker-related S1 cortex. Have the authors tested in the

spatial profile of drug diffusion? Using TTX to test for the effectiveness of drug diffusion is certainly not sufficient because even a partial block of some neurons is likely to cause the cessation of all activity.

Furthermore, how can they be certain that the concentration of the various antagonists is sufficiently high at the L2/3 PN synapses? Are the authors certain that all S1 PNs are unaffected by neuromodulators? At least the abstract of this study claims that this is the case. I think this point should be addressed in detail; otherwise the authors' claim is by far too strong. Furthermore, a proper assessment for the role of neuromodulators is not possible by applying antagonist via a port in the cranial window but rather by silencing neuromodulator-releasing brain regions such as the basal forebrain or the locus coeruleus. In case of the thalamus, the authors have shown that this approach would be feasible.

In addition, the application of NMDA and AMPA antagonists will in every case affect the neuronal activity as the block of ionotropic glutamate receptors will silence all excitatory activity. Here, it would have been far more interesting whether specific block of certain areas such as the motor cortex has differential effects on the activity patterns, not the block of the entire excitatory synaptic transmission. Please comment.

Finally, have the authors taken into account that the EC50 of different neuromodulator receptors? Please be more specific.

Figure 2 legend:

Please correct: DNQX is an AMPA receptor not a NMDA receptor antagonist. I assume the authors have used D-AP5.

Page 8, line 200-201

The authors state that 'presynaptic neurons were annotated according to anatomical area'. Please be more detailed here.

Page 8, line 206-208

The neuron selection (using only neuron not responsive to sensory stimulation) here appears to be subject to bias and needs more detailed justification. Please elaborate!!

Page 11, line 272-282

Most of the information provided here is not really novel as a number of both anatomical and in vitro and in vivo studies suggests. These should be quoted, even in the Results section to make this point clear.

Long-range ensembles

Page 11-14

As the authors state themselves, the finding that WL neurons receive less input from the motor areas of the neocortex than NM neurons is perplexing. This would warrant further investigation including L2/3 PNs that do not fall into the WL or NM group. Furthermore, did the different L2/3 PNs receive different input from other neocortical areas as well or was the sample size too low for that?

Page 12, line 304-309

Given the different innervation patterns of VPM and POrn thalamocortical axons in different cortical layers, it is surprising that presynaptic thalamic neurons from both nuclei were 'equally prevalent' in WL L2/3 PNs. This seems to contradict previous work (e.g. Wimmer et al., 2010; Oberlaender et al., 2012). It is not clear how systematically this point has been investigated here. Is there additional evidence to support these findings? Please elaborate!

Suppression of long-range thalamic inputs

Page 14-16, line 365-390

Why did the authors not investigate the effects of first-order (VPM) and higher-order (POrn) nuclei separately but rather inactivated 'the thalamus'. How selective was the expression of ArchT?

Page 16, line 394-398

Was the net change in activity observed in control mice significantly different from that in ArchT-expressing mice? Please be more specific here.

Discussion

The statement that neuromodulation plays only a limited role needs to be set into context and references should be provided. In addition, the discussion of the role of thalamic input should be more detailed, taking into account differences between input from first-order and higher-order nuclei.

Referee #2

(Remarks to the Author)

Using a combination of two-photon calcium imaging and retrograde monosynaptic tracing, the authors discovered that long-range connectivity of neurons in barrel cortex that reflected the animal's behavioral state differed from those of that did not. The neurons that tracked behavioral state received more thalamic inputs, and non-modulated neurons received more M1/M2 inputs. The experimental techniques are advanced and meticulous, and the scientific findings are novel, unexpected and of

general significance.

Major Comments

1. Are there any known tropisms to the rabies virus used in this paper? For example, do these viruses jump synapses better if they are soma-targeting compared to dendrite-targeting?
2. The authors showed a lack of a difference in proportion of GABAergic inputs, but perhaps there is a difference between different inhibitory interneuron types? I would expect more PV inputs on cells that receive more thalamocortical inputs (WL neurons), and more VIP inputs on cells that receive more motor cortex inputs (NM neurons).
3. In the Introduction, the authors mention: "Instead, functional specificity is flexible and may arise from differences in the intrinsic properties of individual neurons." However, the paper never rules out the possibility of intrinsic property differences. I think this would be very interesting to measure. That said, given that the paper is already filled with interesting experiments, perhaps it would make sense to delete this sentence from the Introduction and put in Discussion that intrinsic property differences were not ruled out.
4. In Fig. 2 experiments, the effectiveness of the dosage used for each pharmacological blocker should be verified. Perhaps this can be done in one additional mouse, by administering ACh/NE before and after the administration of the blockers?
5. In Fig. 2 experiments, was the order of pharmacological blocker experiments randomized? Randomization would be critical, but was not mentioned.
6. The BTX experiments were very interesting! More experiments along these lines would be highly informative in narrowing down the origin of these movement signals. Specifically, I suggest trimming all the whiskers, and correlating neural activity with locomotion. This would test whether the signal is an efference copy of whisking, or general locomotion signal.
7. I would appreciate further discussion of some key outstanding questions and future directions. Where do the authors think the locomotion signal originates? What were some of the existing hypotheses about the origin of these locomotion signals in barrel cortex, and do the results of this paper refute any of them? Also, can the authors speculate about the putative function of these signals? And predict whether the locomotion tracking vs non-tracking cells might have different output connectivity?

Medium-level Comments

1. The train and test set divide during the decoder analysis was not entirely clear to me. Were the inputs to the decoder frame-by-frame activity, or were the activity traces first divided into whisker movement/locomotion epochs and averaged across each epoch?
2. According to Fig. 1 legend, both Fig. 1g left and Fig. 1l show across-session decoder performance. If so, why do the distributions look different?
3. Fig. 1l would be more informative if it was represented with "days between recording sessions" as X-axis and "Prediction (R2)" as Y-axis.
4. What were the proportions of neurons showing significant up-modulation on W only, W and WL, and WL only?
5. In Fig. 2 experiments, was the pharmacological blocker applied during the two-photon imaging session? How long was the baseline recording period, the break during pharmacological application, and the post-administration recording period?
6. In Fig. 2 experiments, a sham control (e.g., ACSF injection) would have been informative.
7. In Fig. 2e-h, the number of neurons is much smaller for NMDA experiments than other experiments. Why is this?
8. In Fig. 2i, negative R2 values warrants an explanation (I assume it's because of the train vs test set divide). Also, I don't understand how 4/5 sessions showed a negative R2 value and yet the statistics was not significantly different from zero?
9. In Fig. 4e-f, which epochs was the 'Modulation' calculated over (e.g., just WL epochs, or both W and WL epochs)? Further, I'd like a clarification about how 'modulation' was calculated. Modulation is described as change in activity during movement relative to baseline. Was the activity during movement averaged across the entire movement epoch, regardless of how long the epoch was?
10. In Fig. 4k, a direct comparison with no-BTX control would be informative. Replotting of Fig. 1l would be acceptable, insofar as the number of days between the recording sessions were comparable between the two conditions.
11. Extended Data Fig. 11: It is interesting that there is no difference between VPM and POM. I suggest the authors add a similar comparison of M1 vs M2.

Minor Comments

1. There are too many abbreviations in this manuscript, and it's hard for the reader to keep up with them.
2. Typo in sentence: "A wheel speed encoder was used to measured locomotion speed."
3. Extended Data Fig. 8a: Both teal and magenta are marked 'WL'.

Version 1:

Reviewer comments:

Referee #1

(Remarks to the Author)

Revised Nature manuscript 2023-05-07457A

Distinct brain-wide presynaptic networks underlie the functional identity of individual cortical neurons

It is evident from the rebuttal letter and the new version of the manuscript that the authors have made substantial revisions.

The majority of my concerns raised with the initial manuscript have been adequately addressed, resulting in much improved version.

I acknowledge that the authors made major efforts to address my concerns regarding their statement regarding the role of neuromodulators and the effectiveness of blocking ACh and NE receptors. The revised version employs a rapid detection

approach for responses mediated by GPCRs, including muscarinic ACh receptors (mAChRs). However, ACh responses mediated by nicotinic ACh receptors (nAChRs) are significantly faster than those detected by GRABACH (GPCR-activation-based ACh) indicator. For this it would be helpful to mention the spatiotemporal resolution of the indicator responses and demonstrate its onset at a higher temporal resolution. That indicator response is certainly fast but as the paper by Jing et al. (2020) demonstrates but it appears to be on a time scale that would miss a major part of the nAChR response. The recordings presented in the new Extended Data Fig. 4 also suggest that only the late desensitized component can be recorded but not. It would be useful if the authors could provide more detail on the response kinetics of the ACh and NE indicators.

Furthermore, under the recording conditions in this *in vivo* study, the application speed of an antagonist is also likely to be relatively slow and may therefore not result in an effective block of the response.

I feel that a direct test of the effect of neuromodulatory input requires the block of the physiological sources of the neuromodulators (i.e. the nucleus basalis of Meynert and the locus coeruleus, respectively), e.g. by optogenetic inactivation. However, I am fully aware that this would be well beyond the scope of the present study.

Nevertheless, I suggest that the authors substantially tone down their assertion that neuromodulatory inputs do not affect or play a limited role in spontaneous movement activity. Based on the experimental results, the assertion made in the paper is in my view too strong.

Furthermore, the authors state on page 20 that the 'stable, sensory-independent activity of wS1 L2/3 neurons may constitute a reflection of internal models of the body and environment that allow for the integration of sensory information ...'. It would be helpful if they could mention whether this is supposed to be exclusively the function of L2/3 excitatory neurons and whether neurons in other cortical layers may fulfil other cortical functions.

Referee #2

(Remarks to the Author)

"Distinct brain-wide presynaptic networks underlie the functional identity of individual cortical neurons" by Inacio et al. is a beautiful paper with highly novel results that would be of wide interest, especially to cortical physiologists. I am very impressed by all the new experiments that the authors have conducted since the previous submission, particularly the distinction between VPM and POM, and the GRAB sensor experiments. In addition, I agree that investigating the differences in GABAergic neuron subtypes in the presynaptic network would be in exciting future direction – I will look forward to seeing it in the future! I only have a few minor comments and suggestions for the authors, but overall, I think this manuscript is more than ready for publication in Nature.

1. Individual neurons are classified into movement uncorrelated and Mov_up and Mov_down neurons. It would be more informative if the authors had an accompanying quantification of the correlation between neural activity and whisker movement/locomotion as a continuous variable (e.g., Pearson correlation or linear decoder weight). Such a quantification is done in Fig. 2, but there are several other places where it would be useful.

1-1) Such variable could be shown as a histogram rather than a pie chart in Fig. 1h

1-2) "Together, these results show that a subset of wS1 PNs reliably encodes spontaneous movements and that the coding of spontaneous movements and sensory stimuli is independent given that some neurons have both properties."

To make the claim that coding of spontaneous movements and sensory stimuli are truly independent, the authors would need to at least show the scatterplot across neurons of movement correlation vs sensory responsiveness. Alternatively, they could show that sensory-driven activity and spontaneous behavior-driven activity is largely orthogonal, as shown in V1 by Stringer et al (2019, DOI: 10.1126/science.aav789).

1-3) The correlation between spontaneous movement and Ach/NE concentration (measured with GRAB sensors) could be quantitatively compared with the distribution across neurons. Was the correlation higher or lower compared to Mov_up neurons?

1-4) How did the optogenetic terminal suppression experiments change the correlation with movement across neurons?

2. "Spontaneous movements, the proportion of spontaneous movement-dependent neuronal subsets, and cortical depth of the postsynaptic (target) neurons were similar between the two groups (Extended Data Fig. 9)."

This is worded in a confusing way. Please clarify that the former two (spontaneous movement and proportion of neuronal subset) refer to the two groups of mice, whereas the third point (cortical depth of postsynaptic neurons) refer to the two groups of electroporated neurons? Even though these ultimately mean the same thing, a clarification would be helpful.

3. Line 243: Section heading "Local (wS1) presynaptic networks" – It should be mentioned somewhere in the Results section that only layer 2/3 excitatory neurons were traced.

4. "Individual glutamatergic presynaptic networks were often characterized by a smaller fraction of L4 compared to L2/3 or L5 neurons." – Please clarify whether this was due to the fact that the authors only targeted non-sensory neurons, or whether the distribution is consistent with a typical wS1 L2/3 excitatory neuron.

5. "All long-range presynaptic neurons were glutamatergic." – Did the authors actually quantify this? If not, I would suggest rephrasing as "We expect all long-range presynaptic neurons to be glutamatergic." There have been some reports of long-range inhibitory connections, particularly from subcortical regions.

6. "We found that spontaneous movement-dependent activity is largely preserved even after combined bilateral whisker trimming and mystacial pad paralysis (Extended Data Fig. 15)." – I suggest clarifying that spontaneous movement refers to locomotion in this case.

7. "We observed that light pulses elicited a relatively brief (~0.5 s) whisker movement in both control and ArchT-expressing mice (Extended Data Fig. 16)." Given that this twitch is also observed in control mice, it is likely behaviorally induced by mice detecting the optogenetic light. Could the authors please expand on what measures they took to prevent the mice from seeing the light? Did they try masking light or light shielding? I understand if these measures were impossible to combine with simultaneous 2P imaging, I just would appreciate some mention of this either in Results or in Methods (and acknowledgment that mice were likely able to see the optogenetic light).

8. Pupil dilation has been observed to be highly correlated with locomotion. Some mention or discussion of pupil dilation would provide a nice link with a large volume of previous literature. Do the authors think the results would be the same if pupil dilation was investigated, particularly with regards to connectivity differences and the lack of dependence on Ach and NE?

Distinct brain-wide presynaptic networks underlie the functional identity of individual cortical neurons

We thank the reviewers for their careful assessment of the manuscript and constructive comments. Their feedback motivated us to conduct new sets of experiments and analyses. We also addressed the lack of clarity in our analyses, descriptions, and interpretations throughout the manuscript. We appreciate the reviewers' supportive, insightful, and thorough comments, which significantly improved our manuscript.

General response

We performed new lines of experiments to address the major common concerns:

1) The effectiveness of local application of neuromodulatory receptor blockers in wS1.

Both reviewers raised concerns regarding the effectiveness of our experimental approach to block neuromodulatory receptors locally in wS1 L2/3 neurons while simultaneously imaging their activity. To directly assess the effectiveness of these receptor blockers in our imaging experiments, we expressed an ACh sensor (GRAB_{ACh}, AAV9-hSyn-Ach4.3) or a NE sensor (GRAB_{NE}, AAV9-hSyn-NE2m) in wS1 L2/3 neurons^{1,2}. In the presence of an appropriate receptor blocker, ACh or NE binding to their respective sensor is prevented, thus abolishing fluorescence signal fluctuations^{1,2}.

We performed imaging and blocker application exactly as described previously for Fig. 2 (Pages 29-30, Lines 813-840). Consistent with previous studies, GRAB_{ACh} sensor activity significantly increased during spontaneous movements^{1,3-6}. However, application of ACh R blockers completely abolished the movement-dependent increase in GRAB_{ACh} sensor activity. Similarly, the transient increase in GRAB_{NE} sensor activity during spontaneous movements was abolished after application of NE R blockers^{2,7}. These results confirm that ACh and NE R blockers effectively reach all cells in the FOV, validating our experimental approach. These results have been updated in the revised manuscript (Extended Data Fig. 4; description in Page 6, Lines 153-158).

Extended Data Fig. 4: Effectiveness of local application of neuromodulatory receptor blockers through a cranial window port. **a**, Example imaging FOV denoting the expression of the ACh sensor GRAB_{ACh} in wS1 L2/3 neurons. Scale bar, 100 μ m. **b**, ΔF averaged over the full FOV and two example ROIs, aligned to the onset of WL events (vertical bars) before and after reapplication of Ringer's solution in an example mouse (sham session). **c**, Corresponding whisker movements. Results are presented as mean \pm s.e.m. (**b-c**). **d-f**, As in a-c, but instead exemplifying the expression of the ACh sensor GRAB_{ACh} and data recorded before and after application of ACh R blockers. **g-i**, As in a-c, but instead exemplifying the expression of the NE

sensor GRAB_{NE} and data recorded before and after application of NE R blockers. **j-l**, Baseline-subtracted mean ΔF during WL events (modulation during WL) for sham ($n = 2$, **j**), ACh-R blockade ($n = 2$, **k**), and NE R blockade ($n = 1$, **l**) sessions. WL-related increases in ACh and NE were abolished when the respective R blockers were applied, but not in sham sessions.

2) The contribution of VPM and POM inputs to spontaneous movement-dependent activity in wS1.

Reviewer 1 asked which sensory thalamic nuclei contribute to the movement-dependent activity of wS1 L2/3 PNs. To address this question, we selectively expressed ArchT either in VPM or in POM, instead of expressing ArchT in both thalamic nuclei. Suppression of VPM axon terminals innervating wS1 significantly reduced spontaneous movement-dependent activity in wS1 L2/3 PNs. However, the same manipulation of POM axon terminals showed a minimal effect on movement-dependent activity. These results suggest that, while movement-correlated neurons receive equally abundant thalamic inputs from VPM and POM, the VPM nucleus is primarily responsible for the movement-dependent activity in wS1 L2/3 PNs. This new data is now included in Extended Data Fig. 17 and described in Page 17, Lines 423-428.

Extended Data Fig. 17: Optogenetic suppression of VPM and POM inputs in wS1 L2/3. **a**, Expression of the opsin ArchT in VPM. ArchT-tdTom⁺ areas are overlaid on the mouse atlas section corresponding to the center of the injection for VPM.

Each shading color represents an individual mouse ($n = 5$). Scale bar, 0.5 mm. **b**, Example composite epifluorescence image of a coronal brain section showing ArchT-tdTom⁺ cell bodies in VPM. Scale bar, 0.5 mm. **c**, Laminar distribution of VPM ArchT-tdTom⁺ axons in wS1, as demonstrated by example 2-PT images. The characteristic patterns of innervation confirm the selectivity of injection during imaging, prior to histological analysis. Scale bar, 0.1 mm. **d**, Baseline-subtracted mean activity (light modulation) of movement-uncorrelated (black) or movement-correlated (magenta) neurons significantly affected by light pulses ($n = 174$ of 1108 uncorr., and $n = 72$ of 338 corr., 6 mice). P values, comparison of activity of movement-uncorrelated and movement-correlated neurons (Wilcoxon rank-sum test). Note that movement-uncorrelated group includes 8 neurons (of 174) with a strong rebound activity; running analysis with and without these neurons did not affect the results. **e-h**, As in a-d, but for the expression of the opsin ArchT in POM (neurons significantly changed by light stimulation, $n = 32$ of 585 uncorr., and $n = 17$ of 267 corr., 3 mice).

3) The role of motor cortical inputs in spontaneous movement-dependent activity in wS1.

In addition to dissecting the functional role of VPM and POM inputs to L2/3 PNs in wS1, we also investigated how motor cortical inputs affect movement-dependent activity in wS1. We performed a new calcium imaging experiment combined with optogenetic manipulation of axon terminals from motor cortex. Optogenetic suppression of inputs from motor cortex barely changed the movement-dependent activity in wS1. This result is consistent with the anatomically fewer monosynaptic inputs from the motor cortex to movement-correlated neurons in wS1. This result is now presented in Fig. 5 and described Page 17, Lines 428-434.

Part of Fig. 5: Optogenetic suppression of M1/2 inputs to spontaneous movement-related activity. **e**, Effect of light pulses on the activity of movement-correlated neurons. Example neuron from a M1/2 ArchT-expressing mouse. Top, responses to individual light pulses (F deconv.). Bottom, average PETH of baseline-subtracted activity. Orange shaded area indicates light pulse duration (time-normalized). **h**, Baseline-subtracted mean activity (light modulation) of movement-uncorrelated (black) and movement-correlated (magenta) neurons significantly affected by light pulses. P values, comparison of activity of movement-uncorrelated and movement-correlated neurons (Wilcoxon rank-sum test). M1/2 ArchT-expressing mice (affected neurons, $n = 39$ of 271 uncorr. and $n = 13$ of corr., 3 mice). **j**, Out-of-sample R^2 ratio ($R^2_{\text{across}} / R^2_{\text{within}}$, thalamus ArchT vs. control, $P = 0.012$; M1/2 ArchT vs. control, $P = 0.10$; Kruskal-Wallis ($P = 0.0005$) followed by Wilcoxon rank-sum tests).

4) GABAergic neuron subtypes in the presynaptic networks of movement-correlated neurons in wS1.

As reviewer 2 raised this question, we are also highly interested in the contribution of GABAergic neuron subtypes within the local presynaptic networks for movement-correlated vs. movement-uncorrelated cells. In the original manuscript, we quantified presynaptic GABAergic cells with GABA immunostaining, but did not differentiate further into GABAergic neuron subtypes. Since these brains ($n = 22$) had already been histologically processed, further investigation was not possible. We performed a new cohort of single-cell based input tracing experiments (total 7 brains, 4 movement-uncorrelated and 3 movement-correlated) and implemented a four primary antibody-based, five-color

confocal imaging protocol. This enabled us to identify three subtypes of GABAergic neurons (PV, SST, and VIP) within the local presynaptic network.

PASSAGE REDACTED. However, our sample size is limited. To obtain a more definitive answer to this question, a larger cohort of animals and a comprehensive strategy are needed not only to quantify these subtypes but also to probe each inhibitory motif based on postsynaptic activity. Given the time required to thoroughly investigate GABAergic presynaptic networks, we believe the scope of this investigation warrants a follow-up study. **PASSAGE REDACTED.**

FIGURE REDACTED.

5) The role of sensory feedback in spontaneous movement-dependent activity in wS1.

To further address whether sensory feedback is responsible for movement-dependent activity in wS1, we examined wS1 activity under bilateral whisker trimming and facial muscle paralysis conditions. As suggested by reviewer 2, we first bilaterally trimmed all the whiskers and recorded wS1 L2/3 activity. Despite this manipulation, the correlation between neural activity and locomotion speed remained

largely intact. Next, we combined bilateral whisker trimming with bilateral facial muscle paralysis induced by BOTOX (BTX) injection. Even under these combined manipulations, movement-dependent neuronal activity persisted. These results demonstrate that spontaneous movement activity in wS1 L2/3 is not primarily driven by sensory feedback. Results are summarized in Extended Data Fig.15, and described in the revised manuscript (Page 15, Lines 376-381).

Extended Data Fig. 15: Effect of bilateral whisker trimming and whisker trimming combined with mystacial pad paralysis on wS1 L2/3 PNs activity in relation to spontaneous movements. **a-c**, Top, Example images of videorecorded whisker movements (mean of the absolute difference between consecutive frames over ~14 s), before (intact, **a**) and after bilateral whisker trimming (Trimming_{only}, **b**), as well as combined bilateral whisker trimming and bilateral BTX injections (Trimming + BTX, **c**) in the same mouse. Bottom, Raster plots of neuronal activity during spontaneous movements and corresponding locomotion speed traces; individual neurons (690) tracked across the three imaging sections (intact, Trimming_{only}, and Trimming + BTX) are sorted from top to bottom by decreasing weight on the first principal component. **d**, Correlation (r) of the activity of individual neurons with locomotion speed before (intact) vs. after Trimming_{only} ($R^2 = 0.71$, $P < 0.0001$, regression, $n = 3$ FOVs of 2 mice). **e**, Correlation (r) of the activity of individual neurons with locomotion speed Trimming_{only} vs. Trimming + BTX ($R^2 = 0.70$, $P < 0.0001$, regression, $n = 3$ FOVs of 2 mice). **f**, Prediction of locomotion speed from population activity. Out-of-sample R^2 ratio ($R^2_{\text{across}} / R^2_{\text{within}}$) for chronic imaging (as in Fig. 11), for intact vs. Trimming_{only} (black circles), and for Trimming_{only} vs. Trimming + BTX (magenta circles) ($P > 0.05$, Wilcoxon rank-sum test).

In addition to these five new sets of experiments, we have addressed all the comments from the reviewers. We revised the manuscript, after integrating reviewers' comments and suggestions. Below are the detailed point-by-point responses to reviewers' comments (reviewer's comment is in blue, our response is in black).

Point-by-Point Response

Referees' comments:

Referee #1 (Remarks to the Author):

In their study, Inácio and co-workers investigate in the primary somatosensory (S1) cortex the relationship between the selectivity of layer 2/3 (L2/3) pyramidal neurons (PN) to sensory stimuli and their presynaptic connectivity. The authors approach this question with an impressive number of techniques including two-photon calcium imaging, neuropharmacology, monosynaptic input tracing, and optogenetic activation and inactivation of brain regions.

They authors observed spontaneous movements during their in vivo experiments: 1. short whisker movements without locomotion and longer/large amplitude whisker movement accompanied by locomotion (WL). The activity of these neurons is unregulated during locomotion while that of another fraction showed no change in activity. The authors aimed at studying the underlying principle of the relationship between whisker movement spontaneous locomotion in the L2/3 PNs of the S1 cortex. They found that neuronal activity in response to different behavioural states is stable over time and claim that it is dependent on neuromodulation by e.g. acetylcholine or noradrenaline. Rather, this activity patterns is solely dependent on glutamatergic inputs on a differential innervation by afferents from other cortical areas (such as the motor cortex) and the thalamus.

I have several concerns with this study which I will address point by point.

We thank the reviewer for their appreciation of this study. The reviewer raised several important questions, including regarding the effectiveness of ACh and NE R blockade in our experimental conditions, and the role of different excitatory sources in driving the spontaneous movement-dependent activity of L2/3 PNs in wS1. These questions motivated us to perform new sets of experiments (General response 1, 2, & 3). Additionally, we clarified some of the conclusions and interpretations of our study. We believe these revisions have strengthened our results, conclusions, and overall interpretation.

Specific points

Page 3, line 78-82

The authors use the terminology up- and down-modulated here although the later claim that 'neuromodulation' did not affect the relationship between locomotion and whisker movement. To avoid confusion, this should be amended.

1) We thank the reviewer for raising this point. We agree that the usage of 'modulation' is confusing. Therefore, we removed the terms 'up-modulated' and 'down-modulated'. Additionally, we simplified our terminology and reduced the number of non-standard abbreviations and acronyms throughout the manuscript. Terminology:

- Behavioral variables: whisker movements (**whisk. mov.**) and locomotion speed (**l. speed**).
- Behavioral events: whisker movements without locomotion (**W_{only}**) and whisker movements with locomotion (**WL**).
- Neurons classified based on activity changes during spontaneous movements: unchanged (**movement-uncorrelated, uncorr.**), increased (**Mov_{up}**), decreased (**Mov_{down}**).
- Most Mov_{up} neurons increased their activity during WL (for simplicity, these are now referred to as **movement-correlated, corr.**).

Page 3, line 85-87

Was the number of active neurons dependent on stimulus strength? Did the authors check whether it was different in males or females or dependent on the day/night cycle? This is relevant for the claim that neuromodulation is not effective in changing the whisker movement/spont. locomotion relationship.

2) We thank the reviewer for bringing up this point, which was not clear in the original manuscript. We used two whisker stimulation conditions, which differed in terms of stimulus duration: 50 ms and 250 ms. The percentage of sensory-responsive neurons did not differ when comparing these two stimuli (50 ms vs. 250 ms, $14 \pm 5.9\%$ vs. $11 \pm 6.0\%$, $n = 21$ vs. 12 animals, $P = 0.15$, Wilcoxon rank-sum test) (added to Page 28, Lines 787-790).

The percentage of sensory-responsive neurons did not differ between males and females ($13 \pm 5.9\%$ vs. $12 \pm 6.1\%$, $n = 19$ vs. 14 animals, respectively; $P = 0.57$, Wilcoxon rank-sum test). Similarly, the percentage of Mov_{up} and Mov_{down} neurons was comparable between the male and female groups ($30 \pm 11\%$ vs. $33 \pm 11\%$, $n = 26$ vs. 21 animals, respectively, $P = 0.27$, Wilcoxon rank-sum) (added to Page 26, Lines 707-711).

Mice were kept under a reverse 12-hour light-dark cycle (lights off at 9 a.m.). All recordings were performed during the dark cycle when animals are active. Recording times were consistent for each mouse; thus, we were unable to assess the sleep-wake cycle or circadian influences on recorded activity.

Page 5, line 137 to Page 6, line 138

There is ample evidence from both in vivo and in vitro studies that neurons in all layers of the neocortex including sensory cortices are subject to neuromodulation by acetylcholine, noradrenaline, dopamine and several neuropeptides, e.g. orexin although several of these are located in deeper cortical layers. The authors chose not to quote these studies. I think this should be amended

3) We agree with the reviewer on the importance of various neuromodulators for cortical activity. The current study focuses on two neuromodulators, ACh and NE, which have been highly implicated in behavioral state changes. Therefore, we primarily cited studies related to this aspect. In response to the reviewer's comment, we added a more general statement and references to highlight other neuromodulatory systems that influence cortical activity⁸⁻¹¹, and we also provided a more explicit context to our neuromodulatory receptor blocker experiments (Page 6, Lines 142-147).

Page 6, Lines 142-147: "*Neuromodulators, including neuropeptides, exert powerful effects over cortical function^{34,41-43}. Acetylcholine (ACh) and norepinephrine (NE) terminal activity in sensory cortices, for example, is highly correlated to spontaneous movements^{32,36,44-47}. These neuromodulators have been tightly linked to the locomotion-related gain modulation of sensory responses in PNs^{31,48,49}. However, a direct effect of ACh and NE on the activity of large populations of L2/3 PNs during spontaneous movements remains to be established.*"

Page 5, line 139-152

The approach to test the relevance of neuromodulators for the activity patterns appears to be somewhat crude and does not take into account that the drug diffusion (i.e the ACh, NE and glutamate receptor antagonists) is likely to be highly non-uniform and affected primarily superficial but not deep layer PNs in whisker-related S1 cortex. Have the authors tested in the spatial profile of drug diffusion? Using TTX to test for the effectiveness of drug diffusion is certainly not sufficient because even a partial block of some neurons is likely to cause the cessation of all activity.

Furthermore, how can they be certain that the concentration of the various antagonists is sufficiently high at the L2/3 PN synapses? Are the authors certain that all S1 PNs are unaffected by neuromodulators? At least the abstract of this study claims that this is the case. I think this point should be addressed in detail; otherwise the authors' claim is by far too strong.

4) We agree that this is an important point that needs to be addressed and thank the reviewer for encouraging us to consider additional experiments to test the effectiveness of our neuromodulatory receptor blocker application (General response 1).

First, we would like to clarify that our goal was to test if ACh and NE released from cholinergic and noradrenergic terminals directly innervating wS1 L2/3 drive the spontaneous movement-dependent activity of wS1 L2/3 PNs.

To test whether ACh and NE R blockers affect all neurons in the imaging FOV (typically 481 x 481 to 573 x 573 μm , at $\sim 300 \mu\text{m}$ of depth), we used a genetically engineered ACh (GRAB_{ACh}) or NE (GRAB_{NE}) sensor that reports ACh or NE binding through an increase in fluorescence signal^{1,2}. We reasoned that, in the presence of appropriate blockers, ACh or NE will no longer be able to bind to the respective sensor, and changes in fluorescence signals will be abolished^{1,2}. To express GRAB_{ACh} or GRAB_{NE}, we injected AAV9-hsyn-Ach4.3 (Ach3.0) or AAV9-hsyn-NE2m (NE3.1) (WZ Biosciences) in wS1 L2/3. We performed blocker delivery and imaging exactly as described previously for Fig. 2 (Pages 29-30, Lines 813-840).

As predicted from the literature, we observed a strong increase in GRAB_{ACh} fluorescence signals in wS1 L2/3 neurons during spontaneous movements^{1,3-6}. Application of ACh R blockers eliminated the spontaneous movement-related increase in GRAB_{ACh} fluorescence signals. Similarly, application of NE R blockers abolished the increased GRAB_{NE} fluorescence signal upon the onset of movement^{2,7}. Given that the application of ACh or NE R blockers nearly abolish the GRAB_{ACh} and GRAB_{NE} fluorescence signals under our experimental conditions, we believe the blocker concentrations used in our study are sufficiently high. In addition, we would like to note that the blocker concentrations used in the current study are similar to those used in previous *in vivo* studies¹²⁻¹⁶. These new results strongly support that ACh and NE R blockers affect all cells within the imaging FOV, validating our experimental approach. These new results are shown in Extended Data Fig. 4 and described in the revised manuscript (Page 6, Lines 153-158).

We concluded that ACh and NE are not the main drivers for this type of activity, as changes in the temporal profile of neuronal firing and population activity in relation to spontaneous movements were either subtle or none. However, we acknowledge a potential effect of these neuromodulators on activity levels. We report that these blockers have some effect on the magnitude of activity, particularly with respect to sensory responses (see Extended Data Fig 5). Therefore, while our data does not support the view that ACh and NE inputs are the main drivers or sources of spontaneous movement-dependent activity in wS1, our results are not at odds with previous studies showing that ACh influences the activity levels of some L2/3 cells and notably the gain of sensory responses¹⁵⁻¹⁷.

The abstract now reads: "*We show that behavioral state-dependent neuronal activity patterns are stable over time. These are minimally affected by neuromodulatory inputs and are instead driven by glutamatergic inputs.*".

Furthermore, a proper assessment for the role of neuromodulators is not possible by applying antagonist via a port in the cranial window but rather by silencing neuromodulator-releasing brain regions such as the basal forebrain or the locus coeruleus. In case of the thalamus, the authors have shown that this approach would be feasible.

5) We understand the reviewer's concern regarding the effectiveness of passive diffusion of the neuromodulatory receptor blockers under our experimental conditions. We believe that the imaging experiments using GRAB_{ACh} and GRAB_{NE} sensors to assess the blocker's application effectiveness address this concern.

Our approach, based on a custom cranial window with an access port, enables stable imaging of a large and same population of neurons over multiple days to test the effects of various pharmacological agents. We avoided direct injection of blockers using a relatively large pipette, as it could cause tissue disruption. We reasoned that a pharmacological approach would provide a more direct and robust suppression compared to optogenetic suppression of basal forebrain and locus coeruleus or of opsin-expressing ACh or NE terminals innervating wS1. Direct optogenetic manipulation of basal forebrain or locus coeruleus could affect cortical activity via multiple indirect pathways, making it difficult to establish a causal link between direct neuromodulatory inputs and the activity of cortical neurons. Regarding optogenetic manipulation of neuromodulatory axon terminals in wS1, we were concerned that short optogenetic light pulses over inhibitory opsin-expressing terminals might not effectively modulate the axon terminals. Neuromodulation through ACh and NE is thought to occur at different spatiotemporal dimensions compared to the characteristic of glutamatergic and GABAergic transmission. Extended periods of photo-stimulation lasting from seconds to minutes for optogenetic inhibition may be less effective¹⁸. Please note that, for the optogenetic inhibition of thalamic and motor cortical terminals, we employed light pulses lasting 1 to 1.5 s.

In addition, the application of NMDA and AMPA antagonists will in every case affect the neuronal activity as the block of ionotropic glutamate receptors will silence all excitatory activity. Here, it would have been far more interesting whether specific block of certain areas such as the motor cortex has differential effects on the activity patterns, not the block of the entire excitatory synaptic transmission. Please comment.

6) We agree with the reviewer's point that glutamatergic (NMDA and AMPA) receptor blockers have a powerful effect on all excitatory activity. However, we believe that the robust effect of the NMDA R blocker alone provides interesting insights regarding spontaneous movement-dependent cortical activity.

The reviewer's suggestion to test the effect of different excitatory input sources led us to examine the effect of motor cortical inputs on spontaneous movement-dependent activity in wS1. To specifically suppress excitatory inputs from the motor cortex, we conducted a new calcium imaging experiment combined with optogenetic inhibition of axon terminals from the motor cortex that innervate wS1 (General response 3). We chose terminal inhibition given that direct pharmacological inactivation of the motor cortex might affect wS1 activity not only through direct projections, but also indirectly by altering activity in other brain areas. Optogenetic suppression of axon terminals from the motor cortex showed minimal effects on the activity of movement-correlated neurons in wS1 L2/3. This result is now presented in Fig. 5 and described in Page 17, Lines 428-434.

Finally, have the authors taken into account that the EC₅₀ of different neuromodulator receptors? Please be more specific.

7) We understand the reviewer's concerns regarding the concentration of the neuromodulatory receptor blockers. We chose the concentration of each blocker based on previous studies in which cortical ACh

or NE R were blocked in experimental conditions similar to ours¹²⁻¹⁶. More importantly, the imaging experiments using GRAB sensors support the conclusion that the blocker concentrations used in this study are strong enough to effectively block ACh and NE R.

Figure 2 legend:

Please correct: DNQX is an AMPA receptor not a NMDA receptor antagonist. I assume the authors have used D-AP5.

8) Thank you for finding this error. We corrected the text accordingly (Page 8, Line 178; Page 30, Lines 827-828).

Page 8, line 200-201

The authors state that 'presynaptic neurons were annotated according to anatomical area'. Please be more detailed here.

9) We apologize if the analysis process was not clearly explained in the original manuscript. A full description of how neurons were annotated is included under Methods, Whole-brain reconstruction, annotation, and registration (Page 36, Lines 1019-1035). To highlight this section in the Results section, we have now added "see Methods, Whole-brain reconstruction, annotation, and registration" at the end of the respective paragraph.

Page 8, line 206-208

The neuron selection (using only neuron not responsive to sensory stimulation) here appears to be subject to bias and needs more detailed justification. Please elaborate!!

10) We apologize for the unclear description regarding the selection criteria for target neurons in the single-cell based monosynaptic retrograde tracing experiments. Analysis was performed online, during the experiment, and confirmed post-hoc. Neurons were classified statistically as movement-uncorrelated or movement-correlated (see Methods, Modulation of neuronal activity, Page 35, Lines 979-991). For movement-correlated neurons, we targeted neurons exhibiting the highest modulation values (i.e., change in activity during movement relative to baseline, prior to movement onset) within the FOV. Neurons were additionally classified statistically as responsive or not responsive to whisker stimulation (see Methods, Modulation of neuronal activity, Page 35, Line 997-1003), and we targeted only neurons that did not respond to whisker stimulation. It was imperative to use two groups of neurons which activities varied strictly along the spontaneous-movements dimension. Using movement-correlated neurons and movement-uncorrelated neurons irrespectively of their responses to sensory stimulation would preclude separating input characteristic related to spontaneous movements vs. sensory responses. To clarify this point, we rephrased that sentence (Page 9, Lines 228-230).

Page 11, line 272-282

Most of the information provided here is not really novel as a number of both anatomical and in vitro and in vivo studies suggests. These should be quoted, even in the Results section to make this point clear.

11) As suggested by the reviewer, we have included references that report synaptic connections from L4 to L2/3 using ex vivo and in vivo electrophysiological approaches^{19,20}.

We also added the following concluding sentence to Page 10, Lines 270 to 272: “*In summary, the glutamatergic and GABAergic presynaptic networks of single L2/3 PNs in wS1 exhibit anatomical features found in population studies⁵⁵⁻⁵⁸.*”.

However, we respectfully disagree that these results lack novelty. While GABAergic and glutamatergic synaptic properties and connectivity within L2/3 and across different layers have been extensively studied under ex vivo and in vivo conditions, the unbiased spatial distribution of monosynaptically connected presynaptic networks within a local cortical area has not been well characterized at the *single-cell level*. Recent detailed connectome studies using electron microscopy have revealed spatial information; however, most of these studies do not provide the characterization of functional properties. We argue that the unbiased anatomical analysis of presynaptic networks in *functionally identified single neurons* in the cortex provides crucial insights of the anatomical and functional architecture of the cortex.

Longe-range ensembles

Page 11-14

As the authors state themselves, the finding that WL neurons receive less input from the motor areas of the neocortex than NM neurons is perplexing. This would warrant further investigation including L2/3 PNs that do not fall into the WL or NM group. Furthermore, did the different L2/3 PNs receive different input from other neocortical areas as well or was the sample size too low for that?

12) As noted by the reviewer, the decrease in presynaptic inputs from the motor cortex to movement-correlated neurons in wS1 is puzzling. Building on the reviewer’s insights and previous feedback, we investigated the impact of motor cortical inputs on movement-correlated neurons in wS1 using an optogenetic approach (General response 3). Our findings indicate that motor cortical inputs minimally affect movement-correlated neurons, whereas optogenetic inhibition of thalamic terminals significantly modulates their activity. These results support the long-range anatomical presynaptic networks outlined in our original manuscript. We now included these results regarding the optogenetic inhibition of motor cortical axons in wS1 in Fig. 5.

We agree with the reviewer that characterizing the brain-wide presynaptic structure from a wide range of functionally heterogeneous cortical PNs would be ideal. However, we think that scaling up the current study by mapping the brain-wide presynaptic networks of other functionally distinct PNs may not be feasible within the scope of the current work.

Regarding other neocortical areas, presynaptic neurons were identified at the secondary somatosensory cortex (S2), other sensory cortical areas (SenCtx), contralateral wS1 (cwS1), and perirhinal cortex (PrhCtx), besides the motor cortex (M1/2). The presynaptic inputs from these cortical areas were comparable between the movement-uncorrelated and movement-correlated groups (now more clearly stated in legend to Fig. 4a-b). Please note that in Fig. 4a-b, areas are sorted by predominance of presynaptic neurons across the two groups, and only statistically significant results (movement-uncorrelated vs. movement-correlated) are indicated and further detailed in Fig. 4c-d. To aid in the qualitative assessment of the cortical presynaptic networks, we generated cortical flat maps with geometry preservation (Extended Data Fig. 13).

Page 12, line 304-309

Given the different innervation patterns of VPM and POm thalamocortical axons in different cortical

layers, it is surprising that presynaptic thalamic neurons from both nuclei were 'equally prevalent' in WL L2/3 PNs. This seems to contradict previous work (e.g. Wimmer et al., 2010; Oberlaender et al., 2012). It is not clear how systematically this point has been investigated here. Is there additional evidence to support these findings? Please elaborate!

13) We thank the reviewer for bringing up this issue. As the reviewer pointed out, the well-accepted view on thalamocortical circuits is that VPm neurons mostly innervate L4, and POm neurons mostly project to L5a and L1. Many studies described the biased innervations using histological analysis and electrophysiological recordings. While we fully accept this model of thalamocortical circuits, we also acknowledge that this view is highly simplified. Multiple studies reported direct projections from VPm to L2/3 PNs, in addition to the POm inputs. For example, a circuit mapping study by Petreanu et al. (2009) demonstrated that L2/3 pyramidal neurons receive direct excitatory monosynaptic inputs from VPm and POm (Figure 2b in reference 23)²¹. Reconstruction of individual VPm neurons demonstrate VPm neuronal axons innervating L2/3 (Figures 3E, 6A, B in reference 24)²², and quantifications of VPm and POm boutons in wS1 show a strong innervation of both structures in L2/3²³. In addition, connectomic reconstruction of a cortical column using electron microscopy reveals that VPm axons make direct synaptic connections to PNs in L2/3²⁴.

Suppression of long-range thalamic inputs

Page 14-16, line 365-390

Why did the authors not investigate the effects of first-order (VPm) and higher-order (POm) nuclei separately but rather inactivated 'the thalamus'. How selective was the expression of ArchT?

14) We thank the reviewer for raising these important points. Initially, the reason for inactivating both VPm and POm axon terminals in wS1 was because the presynaptic input fractions in VPm and POm for movement-correlated neurons were similar. However, inspired by the reviewer's suggestion, we added a new experiment through which we independently tested the impact of VPm inputs and POm inputs on the activity of movement-correlated neurons in wS1 (General response 2). We expressed an inhibitory opsin (ArchT) either only in VPm or only in POm, and we suppressed the opsin-expressing axonal terminals in wS1 while imaging the activity of L2/3 PNs also in wS1. We found that the optogenetic inhibition of VPm axon terminals significantly affected the activity of movement-correlated neurons, whereas the suppression of POm terminals showed no significant effect. Thus, while the monosynaptically connected presynaptic neurons were equally abundant in POm and VPm for the movement-correlated neurons in wS1, VPm synaptic inputs may provide stronger excitatory inputs to the movement-correlated neurons. These results are presented in Extended Data Fig. 17 and described in the main manuscript (Page 17, Lines 423-428).

To address the reviewer's question regarding the selective injection of VPm and POm, we added a new analysis on the ArchT expression in the VPm and POm. In addition to evaluating the ArchT-tdTomato expression in VPm and POm histologically, we also checked the expression of tdTomato in the cortex in vivo, using 2-PT structural imaging, to examine where the thalamic terminals mostly resided. In VPm targeted animals, we confirmed that the tdTomato expression discretely filled barrel columns at L4. In contrast, we confirmed that most tdTomato expressing axons innervate L4 septa in POm injected animals. Importantly, we would like to note that optogenetic stimulation was applied to thalamic axon terminals in wS1, not to the thalamic nucleus itself. The validation of thalamic injection sites and their axonal innervation to wS1 is presented in Fig. 5b and Extended Data Fig. 17.

Was the net change in activity observed in control mice significantly different from that in ArchT-expressing mice? Please be more specific here.

15) To answer the reviewer's question, we compared the optogenetic inhibition-induced net changes in neuronal activity from movement-correlated cells among control, thalamic ArchT-injected, and motor cortex ArchT-injected groups. Only the thalamic ArchT-injected group showed a significant reduction compared to control group (Supp. Figure 1).

Supp. Fig. 1: Related to Fig. 5f-h. a-c, Baseline-subtracted mean activity (light modulation) of movement-correlated neurons significantly affected by light pulses in control vs. thalamus ArchT-expressing mice (a), control vs. M1/2 ArchT-expressing mice (b), and M1/2 vs. thalamus ArchT-expressing mice (c). *P* values, activity comparisons (Wilcoxon rank-sum tests).

Discussion

The statement that neuromodulation plays only a limited role needs to be set into context and references should be provided.

In addition, the discussion of the role of thalamic input should be more detailed, taking into account differences between input from first-order and higher-order nuclei.

16) We have now more clearly introduced cortical neuromodulation in the Results, where we clarify the context (Page 6, Lines 142-147): “*Neuromodulators, including neuropeptides, exert powerful effects over cortical function*^{34,41-43}. *Acetylcholine (ACh) and norepinephrine (NE) terminal activity in sensory cortices, for example, is highly correlated to spontaneous movements*^{32,36,44-47}. *These neuromodulators have been tightly linked to the locomotion-related gain modulation of sensory responses in PNs*^{31,48,49}. *However, a direct effect of ACh and NE on the activity of large populations of L2/3 PNs during spontaneous movements remains to be established.*”

In addition, we added the following text to the Discussion (Page 19, Lines 485-492): “*The release of ACh and NE in cortex is closely linked to spontaneous movements, as also confirmed in our GRAB sensor experiments*^{32,36,44-47}. *Our results show that these inputs may influence the activity levels, especially with respect to sensory responses, consistent with the gain modulation of sensory responses during movement by neuromodulation*^{31,48,49}. *However, our neuromodulatory blockade results suggest that the direct neuromodulatory inputs to the cortex are not the main drivers for the neuronal activity in*

relation to spontaneous movements. Instead, the subset of neurons that reliably tracked the behavioral state appeared to be driven by long-range glutamatergic inputs, particularly from the thalamus.”

Regarding VPm and POm, we included references on their innervation patterns in wS1 and further discussed our single-cell tracing data and new optogenetic experiments (Discussion, Page 19, Lines 472-484): *“While movement-correlated PNs receive almost equally abundant inputs from VPm and POm, our optogenetic study demonstrates that the VPm nucleus is primarily responsible for driving the movement-dependent activity in wS1 L2/3. Eliminating whisker movements through muscle paralysis and sensory responses through whisker trimming did not disrupt spontaneous movement-dependent activity in wS1, strongly supporting the notion that this activity is unlikely to be a direct consequence of sensory feedback^{60,65,67}. This raises an interesting issue: what do VPm neurons relay to the primary somatosensory cortex beyond simple sensory transmission, and how is the transmission of sensory and non-sensory information achieved? Sensory thalamic neurons, as cortical neurons, are also active in the absence of sensory stimuli and feedback⁶⁰, suggesting that they can encode non-sensory information. Anatomical analysis supports the heterogeneous innervation patterns of individual VPm neurons across different cortical layers^{68,69}. Future studies will be needed to address the potential functional heterogeneity in VPm neurons⁶⁴.”*

Referee #2 (Remarks to the Author):

Using a combination of two-photon calcium imaging and retrograde monosynaptic tracing, the authors discovered that long-range connectivity of neurons in barrel cortex that reflected the animal's behavioral state differed from those of that did not. The neurons that tracked behavioral state received more thalamic inputs, and non-modulated neurons received more M1/M2 inputs. The experimental techniques are advanced and meticulous, and the scientific findings are novel, unexpected and of general significance.

We thank reviewer for the positive review and constructive comments.

Major Comments

1. Are there any known tropisms to the rabies virus used in this paper? For example, do these viruses jump synapses better if they are soma-targeting compared to dendrite-targeting?

Monosynaptic retrograde tracing using modified rabies virus is a powerful way to investigate the brain-wide anatomical inputs to populations of neurons or single neurons, which at the present remains difficult using electrophysiology or electron microscopy. Yet, it is limited due the fact that not all inputs are labelled, lack of knowledge on the strength of labelled connections, and potential bias associated with viral tropism²⁵. To our knowledge, there is no evidence for a differential spread through soma-targeting vs. dendritic targeting synapses. While we are not aware of such biases, to overcome this potential issue, we directly compared the brain-wide presynaptic networks of two functionally distinct groups using the same method. If a bias existed, it is reasonable to assume that it would be similar in both groups (Fig. 3f and Extended Data Fig. 10a). Moreover, for long-range inputs, we performed functional optogenetic experiments that support a limited role of M1/2 and active participation of thalamic inputs in spontaneous movement-dependent activity. Finally, we used CVS-N2cDG, which is known to be more suitable for long-range labeling²⁶. We added this aspect to the Discussion (Page 20, Lines 512-518).

2. The authors showed a lack of a difference in proportion of GABAergic inputs, but perhaps there is a difference between different inhibitory interneuron types? I would expect more PV inputs on cells that receive more thalamocortical inputs (WL neurons), and more VIP inputs on cells that receive more motor cortex inputs (NM neurons).

We thank the reviewer for raising this important point. This question on the organization of GABAergic neuron subtypes in presynaptic networks of functionally defined single neurons is undoubtedly of high interest to us. The single-cell based input tracing brains included in the first version of our manuscript (n = 22) had already been histologically processed, and no further histological investigation was possible. Therefore, we placed substantial effort into performing a new cohort of single-cell based input tracing experiments and development of a four primary antibody-based (PV, SST, VIP and RFP for presynaptic cells), five-color confocal imaging protocol (General response 4). From this experimental cohort, we analyzed 7 brains (4 movement-uncorrelated and 3 movement-correlated). Presynaptic cells (RFP⁺) were manually annotated as PV⁺, SST⁺ or VIP⁺ by two experimenters blinded to group identity.

PASSAGE REDACTED. A larger cohort of animals along with a comprehensive strategy to not only quantify, but also probe each inhibitory motif will be required to make conclusive claims about specific inhibitory input structure based on postsynaptic activity. The time frame needed to substantiate such an investigation is not consistent with the time frame of a revision, and it is our belief that the magnitude of such an investigation render it as a follow-up study.

3. In the Introduction, the authors mention: “Instead, functional specificity is flexible and may arise from differences in the intrinsic properties of individual neurons.” However, the paper never rules out the possibility of intrinsic property differences. I think this would be very interesting to measure. That said, given that the paper is already filled with interesting experiments, perhaps it would make sense to delete this sentence from the Introduction and put in Discussion that intrinsic property differences were not ruled out.

We agree with the reviewer’s suggestion. We removed that sentence from the Introduction and added that aspect to the Discussion (Page 20, Lines 523-524).

4. In Fig. 2 experiments, the effectiveness of the dosage used for each pharmacological blocker should be verified. Perhaps this can be done in one additional mouse, by administering ACh/NE before and after the administration of the blockers?

We thank the reviewer for raising these important concerns. These issues were also raised by reviewer 1 (General response 1). We agreed that a decisive experiment to test the effectiveness of the application and concentration of the neuromodulatory receptor blockers is necessary, in addition to the TTX experiment. To directly address these concerns, we conducted a new set of experiments by imaging the fluorescence signals from ACh and NE sensors before and after application of the respective receptor blockers.

We expressed the genetically encoded ACh sensor (GRAB_{ACh}, AAV9-hsyn-Ach4.3 (Ach3.0)) or NE sensor (GRAB_{NE}, AAV9-hsyn-NE2m (NE3.1)) in wS1 L2/3 neurons^{1,2}. In the presence of appropriate receptor blockers, ACh or NE will no longer be able to bind to the respective sensor, and the changes

in fluorescence signals will be abolished. We performed imaging and blocker application exactly as described previously for Fig. 2 (Pages 29-30, Lines 813-840). We first investigated the correlation between spontaneous movements and the fluorescence signals of each GRAB sensor. Consistent with previous studies, GRAB_{ACh} activity significantly increased during spontaneous movements^{1,3-6}. When we applied ACh R blockers, the spontaneous movement-related increases in fluorescence signals from the GRAB_{ACh} sensor were eliminated. In addition, fluctuations in fluorescence signals were abolished throughout the entire FOV within 20 min after blocker application. Similarly, increased GRAB_{NE} sensor activity during spontaneous movements was abolished after application of NE R blockers^{2,7}. We believe these results validate the effectiveness of the receptor blocker application approach and concentrations used in this study. We presented these results in a new figure (Extended Data Fig. 4) and described them in the main text (Page 6, Lines 153-158).

5. In Fig. 2 experiments, was the order of pharmacological blocker experiments randomized? Randomization would be critical, but was not mentioned.

We thank the reviewer for bringing up this point. We applied the different receptor blockers according to two imaging session sequences, ACh R – NE R – Glut R (n = 3) or NE R – Glut R – ACh R (n = 2), so that either ACh or NE R blockers were applied in the first session. Imaging sessions were 1-2 days apart. In two mice, only ACh (n = 1) or NE (n = 1) R blockade sessions were performed. Sequences were randomly assigned per animal. We compared the effect of given type of neuromodulatory receptor blockers on the correlation of individual neurons with whisker movements when applied at the first session vs. the later session. We did not find a significant effect of the ACh or NE R blockers sequence position on neuronal correlations with whisker movements ($P > 0.05$, linear mixed model controlling for days as confounding factor for position). These results are included in Supp. Fig. 3 and Page 30, Lines 830-834.

Supp. Fig. 3: Related to Fig. 2. a-b, Scatter plots of the correlation (r) of individual neurons with whisker movements during ACh (a) or NE (b) R blockade vs. baseline.

6. The BTX experiments were very interesting! More experiments along these lines would be highly informative in narrowing down the origin of these movement signals. Specifically, I suggest trimming all the whiskers, and correlating neural activity with locomotion. This would test whether the signal is an efference copy of whisking, or general locomotion signal.

We thank the reviewer for the helpful suggestion. The reviewer's suggestion led us to further investigate whether sensory feedback is responsible for spontaneous movement-dependent activity (General response 5). We conducted a new set of experiments. First, as suggested by the reviewer, we bilaterally trimmed all whiskers and correlated neuronal activity with locomotion speed. The neuronal correlation with locomotion under the bilateral whisker trimming condition did not differ from that of the control condition where all whiskers were intact. Second, to eliminate neuronal activity induced by facial muscle movement, we bilaterally injected BTX into the facial muscles of bilaterally whisker-trimmed animals. Even with these combined manipulations, the movement-dependent neuronal activity persisted. A linear decoder reliably predicted locomotion speed from population activity across different conditions where both sides of whiskers were all trimmed, or whisker trimming was combined with BTX injection (R^2 , 0.47 ± 0.26 values). The R^2 values were similar to that of a linear decoder used to assess the stability of population activity over days in relation to locomotion speed injection (0.30 ± 0.15). These results strongly support that movement-dependent activity in wS1 is not caused by sensory feedback. These results are now summarized in Extended Data Fig. 15 and described in Page 15, Lines 376-381.

7. I would appreciate further discussion of some key outstanding questions and future directions. Where do the authors think the locomotion signal originates? What were some of the existing hypotheses about the origin of these locomotion signals in barrel cortex, and do the results of this paper refute any of them? Also, can the authors speculate about the putative function of these signals? And predict whether the locomotion tracking vs non-tracking cells might have different output connectivity?

Our experiments were designed primarily to evaluate if PNs with distinct functional properties have distinct presynaptic compositions. We were particularly interested in functional properties not directly related with sensory stimulation, as the cortical architecture underlying putative internally generated activity patterns is far less known but critical for understanding of cortical function. Our findings on the presynaptic compositing of movement-correlated cells, together with additional neuropharmacology and optogenetics experiments, do not support the following two main hypotheses on the source of spontaneous movement-dependent activity in wS1 L2/3.

As more clearly stated in our revised manuscript, one hypothesis is that spontaneous movement-related signals in the sensory cortices are prompted by direct ACh and NE inputs. This hypothesis is based mainly on two lines of evidence. First, the activity of ACh and NE terminals in the sensory cortices is remarkably correlated with spontaneous movements, as also confirmed through our GRAB sensor control experiments. Second, the release of ACh in the sensory cortices mediates the locomotion-related gain of PNs sensory responses. We have now further discussed our neuromodulatory results in the context of previous studies (Page 19, Lines 485-492).

The other hypothesis is that this type of activity is generated in M1/2 and transmitted to wS1 via M1/M2 projecting neurons. However, our anatomical and functional data strongly indicates that this is not the case. As stated in the discussion, along with providing other references, we propose that M1/2 pathway may be more relevant for sensorimotor learning in wS1, instead of determining this innate behavioral state-dependent representations (Pages 19-20, Lines 496-511).

Our results rather favor that thalamic inputs may support movement-dependent activity in the cortex. This could occur via thalamic neuromodulation. As stated in the discussion (Pages 19, Lines 492-495), neuromodulators may utilize strong thalamic connections to influence PNs that track behavioral states, rather than directly driving these neurons (Extended Data Fig. 18).

Thalamic results are now discussed in further detail (Page 19, Lines 472-484).

Finally, we speculate that the thalamic origin of spontaneous movement activity in S1, particularly

via VPm, may reflect a developmental trace. At early stages of life, most cortical activity, even in the motor cortex, is driven by inputs arising from the sensory periphery^{e.g.27-35}. Through development and consistent with the emergence of active behaviors, cortical networks gradually switch from an early “passive somatosensory mode” to an adult “active mode”, capable of producing activity disengaged from the sensory periphery.

Regarding the function of movement-related activity, we expanded our Discussion. The text now reads (Pages 20-21, Lines 525-533): *“We found highly convergent yet characteristic presynaptic connectivity patterns depending on the functional property of each neuron. This raises questions about the functions and projection patterns of the movement-correlated neurons within the local network, and their connections to other brain areas. These neurons may report moment-to-moment behavioral states within the sensory cortex and provide a self-referenced framework for integrating information across other brain areas. The stable, sensory-independent activity of wS1 L2/3 neurons may constitute a reflection of internal models of the body and environment that allow for the integration of sensory information with external and internal contexts, prediction of sensory consequences, and guidance of actions⁷⁷.”*

As for the output connectivity, our proposal is now more clearly stated in the Discussion (Page 21, Lines 533-536): *“Parallel to their input connectivity, their output connectivity may follow similar rules, whereby functionally distinct neurons project to a wide range of other cortical areas, yet with a selective amplification or reduction of specific projections^{78,79}. This information will be critical to build a full model of the anatomical and functional landscape for heterogenous cortical neurons.”*

Medium-level Comments

1. The train and test set divide during the decoder analysis was not entirely clear to me. Were the inputs to the decoder frame-by-frame activity, or were the activity traces first divided into whisker movement/locomotion epochs and averaged across each epoch?

We apologize for the lack of clarity in the description of the decoder analysis. The decoders decode the behavior variable from neuronal activity frame by frame. The samples (behavior and corresponding neuronal activity per frame) were uniform-randomly split into training (75%) and test (25%) sets. The decoding weights were estimated from the training set, and then applied to the test set for performance evaluation (R^2). The whisker movements and locomotion speed were recorded simultaneously and thus decoded in parallel. We updated the text accordingly (Page 34, Lines 953-965).

2. According to Fig. 1 legend, both Fig. 1g left and Fig. 1l show across-session decoder performance. If so, why do the distributions look different?

We apologize for the confusion regarding our results in Figure 1 g and l. While in Fig. 1f we show the “within session” decoder performance for an example session, in Fig. 1g, we show the “within session” decoder performance for multiple, independent recording sessions.

To better describe the results, we rephrased Fig. 1g legend (Page 5, Lines 112-114) as such: *“g, Prediction of whisker movements and locomotion speed from neuronal activity. Variance explained within session (whisk. mov., $P = 0.00011$; l. speed, $P = 0.0010$; paired sample t-test $R^2 \leq 0$ vs. $R^2 > 0$; $n = 6$ FOVs, 6 sessions, 5 mice).”*

Our decoding analysis nomenclature is described in Page 34, Lines 970-972: “Using the weight vector estimated from the training set, we decoded the behavioral variables on the test (held out) set within the same condition or session (referred as within) and the trials of other conditions or sessions (referred as across).”.

3. Fig. 1I would be more informative if it was represented with “days between recording sessions” as X-axis and “Prediction (R²)” as Y-axis.

We thank the reviewer for their suggestion. We have replotted Fig. 1I accordingly.

4. What where the proportion of neurons showing significant up-modulation on W only, W and WL, and WL only?

The neurons showing significantly increased activity during spontaneous movements (Mov_{up}) can be categorized as follows, based on types of behavioral events: 25 ± 19% for W_{only}, 41 ± 26 % for WL, and 34 ± 19 % for W_{only} + WL. These data were clarified in the Extended Data Fig. 1 legend. Please note that the W_{only} cell cluster still shows an increase in activity during WL compared to uncorrelated cells (Extended Data Fig. 1), albeit less robust and typically short-lived after WL onset. We used all behavioral events, including W_{only} which have a relatively small duration, as well as event onsets and offsets, to increase classification accuracy.

5. In Fig. 2 experiments, was the pharmacological blocker applied during the two-photon imaging session? How long was the baseline recording period, the break during pharmacological application, and the post-administration recording period?

We apologize for the lack of clarity in the description of the in vivo neuropharmacology experimental steps. Pharmacological blockers were applied during the 2-PT calcium imaging sessions, each lasting a median of 2 h 15 min. The effective time for recording spontaneous activity was 1 h 8 min. The sessions were structured as follows: a baseline period, 42 min (effective spontaneous activity recording time, 30 min); drug application or Ringer’s replacement, 3 min; a waiting period following drug application or Ringer’s solution replacement, 21 min; and a receptor blockade period (or Ringer’s solution), 45 min (effective spontaneous activity recording time, 30 min). There was no significant difference in recording times between baseline and sham/R blocker conditions (P > 0.05, Wilcoxon signed-rank test) (added to Pages 29-30, Lines 820 to 825).

Please note that NMDA and NMDA + AMPA (Glut) R blocker(s) were sequentially applied in the same session. To keep the duration of each session and the effective spontaneous activity recording time reasonable, we reduced the duration of the receptor blockade period by approximately 13 min.

TTX application abolished neuronal activity across the entire FOV (481.4 x 481.4 to 572.9 x 572.9 μm²) within 10-20 minutes. Similarly, our new imaging experiments using GRAB sensors also showed complete suppression of fluctuations in fluorescence signals and therefore suppression of ACh and NE R binding across the entire FOV (481.4 x 481.4 to 572.9 x 572.9 μm²) within 10-20 minutes.

6. In Fig. 2 experiments, a sham control (e.g., ACSF injection) would have been informative.

We thank the reviewer for this important suggestion. We performed sham experiments in which,

following recordings of baseline neural activity and behavior, the Ringer's solution was replaced by Ringer's solution without any receptor blocker. All experimental steps were performed exactly as for experiments in which receptor blockers were used (Pages 29-30, Lines 813 to 840).

We evaluated the effect of the different conditions (i.e., sham – no blocker, as well as ACh, NE, NMDA, and Glut R blockers) on the correlation of single neurons with whisker movements. Specifically, we used linear fixed effects models with interaction between drug type and neuronal correlation to compare regression line slopes (Fig. 2e-h). Consistent with our previous results, NMDA R blockade alone had a robust effect on the correlation of single neurons with whisker movements, and Glut R blockade further decreased this correlation. We also detected a statistically significant effect of ACh R blockade, although the effect size was minor (sham slope = 0.91; ACh R slope = 0.85) (Fig 2e). We computed the out-of-sample R^2 ratio ($R^2_{\text{across}} / R^2_{\text{within}}$) allowing us to compare the relative drop in variance explained from the within to the across condition. There was no significant difference between ACh nor NE R blockade compared to sham. However, the R^2 ratio was significantly lower for NMDA and Glut R blockade compared sham (Fig. 2i).

Part of Fig. 2: Effect of Ringer's solution application. e-h, Scatter plots of the correlation (r) of individual neurons with whisker movements during ACh ($n = 1300$, 6 sessions, 6 mice, e), NE ($n = 1483$, 6 sessions, 6 mice, f), NMDA ($n = 720$, 5 sessions, 4 mice, g), or Glut ($n = 956$, 6 sessions, 6 mice, h) R blockade vs. baseline, respectively; sham sessions (baseline

vs. Ringer's only) are included in e-h (cyan, $n = 1598$, 5 sessions, 3 mice) (null hypothesis, equal slopes: ACh R vs. sham, $P = 0.033$; NE R vs. sham, $P = 0.13$; NMDA R vs. sham, $P < 0.0001$; Glut R vs. sham, $P < 0.0001$; multiple comparisons). i, Prediction of whisker movements from population activity. Out-of-sample R^2 ratio ($R_{\text{across}} / R_{\text{within}}$) (ACh R vs. sham, $P = 0.33$; NE R vs. sham, $P > 0.05$; NMDA R vs. sham, $P = 0.032$; Glut R vs. sham, $P = 0.017$; NMDA R vs. Glut R, $P < 0.0001$; Kruskal-Wallis test ($P < 0.001$) followed by Wilcoxon rank-sum tests).

7. In Fig. 2e-h, the number of neurons is much smaller for NMDA experiments than other experiments. Why is this?

The number of neurons in the NMDA and Glut plots is different (lower for NMDA) because in one animal, besides ACh and NE R blockade, we only performed Glut R (NMDA + AMPA) blockade and did not perform NMDA R only blockade ($n_{\text{NMDA}} = 5$ mice; $n_{\text{GLUT}} = 6$ mice). In addition, for 3 mice, we only performed either ACh, NE or NMDA/GLUT R blockade sessions. We conducted additional neuromodulatory experiments, including a new NMDA/GLUT R blockade session (updated Fig. 2e-h).

8. In Fig. 2i, negative R^2 values warrants an explanation (I assume it's because of the train vs test set divide). Also, I don't understand how 4/5 sessions showed a negative R^2 value and yet the statistics was not significantly different from zero?

We thank the reviewer for bringing up this point. All R^2 values were calculated using test sets, also known as out-of-sample R^2 s. A negative value indicates a predictability poorer than the sample mean and implies that the behavioral readout was inconsistent for the NMDA/Glut R blockers. We updated the text to clarify this point (Page 34, Lines 966-976).

The negative R^2 is not significant because we did one-sided test for the R^2 , $H_0: R^2 \leq 0$ vs $H_1: R^2 > 0$. It was stated in the figure captions. We chose one-sided test because the point was about the consistency of decoding across conditions. A negative R^2 , significant or not, does not change the conclusion.

9. In Fig. 4e-f, which epochs was the 'Modulation' calculated over (e.g., just WL epochs, or both W and WL epochs)? Further, I'd like a clarification about how 'modulation' was calculated. Modulation is described as change in activity during movement relative to baseline. Was the activity during movement averaged across the entire movement epoch, regardless of how long the epoch was?

Modulation was calculated over both W_{only} and WL epochs, as specified in the figure legend (Pages 14-15, Lines 364-365): "**e-f**, Motor cortical (**e**) and thalamic (**f**) input fraction as function of the average modulation of the postsynaptic neurons across spontaneous movements ($W_{\text{only}} + \text{WL}$) ($M1/2$, $P = 0.0024$; thalamus, $P = 0.0045$; regression).".

Modulation was calculated as specified in the Methods section, under Modulation of neuronal activity (Page 35, Lines 979-991). We added the expression "irrespective of movement duration" to address the reviewer's comment: "*The activity (F deconvolved) of each neuron was aligned to the onset and offset of spontaneous movements, W_{only} and WL. Baseline and post-offset activities refer to a 0.5 s window preceding movement onset and a 0.5 s window after movement offset, respectively. A neuron was considered as Mov_{up} if its average activity during W_{only} and/or WL events was significantly higher than its average activity during baseline and/or significantly higher than its average activity post-event ($P < 0.01$, paired-sample t -test). Conversely, a neuron was considered Mov_{down} if its average activity during W_{only} and/or WL events was significantly lower than its baseline and/or significantly lower than*

its post-event average activity ($P < 0.01$, paired-sample t -test). Neurons exhibiting opposite changes in activity for W_{only} and WL were rare ($0.7 \pm 0.2\%$ of all cells) and were not considered for further analysis. Otherwise, neurons were considered as movement-uncorrelated. Modulation refers to the mean activity during movement, irrespective of movement duration, minus the mean activity during baseline, averaged across spontaneous movements (W_{only} , WL, or $W_{only} + WL$).

We also added “(see Methods, Modulation of neuronal activity)” in the Results (Page 13, Lines 328-329).

In Supp. Fig. 4, we present the motor cortical and thalamic input fraction as function of the average modulation of the postsynaptic neuron during WL for comparison with data included in Fig. 4.

Supp. Fig. 4, Related to Fig. 4c-d: Motor cortical and thalamic input fraction as function of the average modulation of the postsynaptic neuron during $W_{only} + WL$ (a, same as included in Fig 4c-d), and during WL (b, M1/2, $P = 0.0048$, and thalamus, $P = 0.022$; regression).

10. In Fig. 4k, a direct comparison with no-BTX control would be informative. Replotting of Fig. 1l would be acceptable, insofar as the number of days between the recording sessions were comparable between the two conditions.

We thank the reviewer for the insightful suggestion. We replotted Fig. 4k accordingly. The across R^2 and R^2 ratio ($R^2_{across} / R^2_{within}$) based on chronic imaging data and based on BTX experiments are comparable (Suppl. Fig. 5 and Fig. 4k).

Supp. Fig. 5: Related to Fig. 4g-k. Prediction of whisker movements from population activity. **a**, Variance explained across conditions for chronic experiments (day 1 vs. day 3 or day 4) and BTX experiments (before BTX vs. 1-2 days after BTX) ($P < 0.0001$ for chronic and BTX; paired-sample t -test across $R^2 \leq 0$ vs. $R^2 > 0$). **b**, Out-of-sample R^2 ratio as a function of de facto days between sessions for both chronic and BTX experiments.

11. Extended Data Fig. 11: It is interesting that there is no difference between VPM and POM. I suggest the authors add a similar comparison of M1 vs M2.

We thank the reviewer for this suggestion. Previously, we compared the 3D spatial distributions of M1/2 cells and found no difference between the movement-uncorrelated and movement-correlated groups, suggesting that there is no group-based anatomical bias in cell distribution (Extended Data Fig. 13 and Results, Page 13, Lines 331-333). However, we had not specifically evaluated the proportions of M1 and M2 cells across brains. We performed a new analysis to compare the fraction of presynaptic cells in M1 vs. M2. It has been reported that neurons in the motor cortex projecting to wS1 are mostly located near the border between M1 and M2³⁶. Therefore, a binary classification of M1/2 cells into either M1 or M2 could result in uncertain estimates depending on the proximity of each cell to the border, potentially leading to spurious differences. To unbiasedly evaluate the distribution of presynaptic neurons in M1/2, we additionally analyzed cell distributions based on the shortest 3D Euclidean distance of each M1/2 cell from the border between M1 (positive distance) and M2 (negative distance). These distributions were similar between the movement-uncorrelated and movement-correlated groups. We incorporated this results in Extended Data Fig. 12 and describe them in the main text (Page 13, Lines 324-326).

Part of Extended Data Fig. 12: Motor cortical (M1 and M2) presynaptic networks. **a**, Three-dimensional distribution of all presynaptic neurons within motor cortex (M1 and M2) for the movement-uncorrelated and movement-correlated postsynaptic neuron groups (data aligned to the Allen Mouse Common Coordinate Framework; uncorr. vs. corr., $P > 0.05$, randomization test on 2-Wasserstein distance, $n = 11$ brains per group). **b**, M1 and M2 presynaptic neurons as fraction of long-range presynaptic neurons (uncorr. vs. corr.; M1, $P < 0.05$; M2, $P = 0.057$; Wilcoxon rank-sum test). Thin lines, individual data points. Thick lines, mean. **c**, Relative proportion of M1 vs. M2 neurons ($P > 0.05$, Wilcoxon rank-sum test). **d**, Weighted distribution of motor cortical presynaptic neurons as function of distance from the M1 and M2 border (uncorr. vs. corr., $P > 0.05$,

randomization test).

Minor Comments

1. There are too many abbreviations in this manuscript, and it's hard for the reader to keep up with them.

We thank the reviewer for raising this point, which is in line with point one of reviewer 1. To add clarity to the text, we limited the use of non-standard abbreviations and acronyms, and we attempted at simplifying our terminology as such:

- Behavioral variables: whisker movements (**whisk. mov.**) and locomotion speed (**l. speed**).
- Behavioral events: whisker movements without locomotion (**W_{only}**) and whisker movements with locomotion (**WL**).
- Neurons classified based on activity changes during spontaneous movements: unchanged (**movement-uncorrelated, uncorr.**), increased (**Mov_{up}**), decreased (**Mov_{down}**).
- Most Mov_{up} neurons increased their activity during WL (for simplicity, these are now referred to as **movement-correlated, corr.**). Note that these neurons are the most robustly changed during spontaneous movements (Extended Data Fig. 1) and were targeted for single-cell based input tracing.

Apart from **W_{only}** and **WL**, as well as **Mov_{up}** and **Mov_{down}**, all other abbreviations and acronyms are referred once in the text but used only in Figures.

2. Typo in sentence: "A wheel speed encoder was used to measured locomotion speed."

Thank you, we corrected it.

3. Extended Data Fig. 8a: Both teal and magenta are marked 'WL'.

Thank you. We corrected it.

References

- 1 Jing, M. *et al.* An optimized acetylcholine sensor for monitoring in vivo cholinergic activity. *Nat Methods* **17**, 1139-1146 (2020).
- 2 Feng, J. *et al.* Monitoring norepinephrine release in vivo using next-generation GRAB(NE) sensors. *Neuron* **112**, 1930-1942.e1936 (2024).
- 3 Eggermann, E., Kremer, Y., Crochet, S. & Petersen, C. C. H. Cholinergic signals in mouse barrel cortex during active whisker sensing. *Cell Rep* **9**, 1654-1660 (2014).
- 4 Lohani, S. *et al.* Spatiotemporally heterogeneous coordination of cholinergic and neocortical activity. *Nat Neurosci* **25**, 1706-1713 (2022).
- 5 Collins, L., Francis, J., Emanuel, B. & McCormick, D. A. Cholinergic and noradrenergic axonal activity contains a behavioral-state signal that is coordinated across the dorsal cortex. *Elife* **12**, 81826 (2023).
- 6 Neyhart, E. *et al.* Cortical acetylcholine dynamics are predicted by cholinergic axon activity and behavior state. *bioRxiv*, 2023.2011.2014.567116 (2024).
- 7 Reimer, J. *et al.* Pupil fluctuations track rapid changes in adrenergic and cholinergic activity in cortex. *Nat Commun* **7**, 13289 (2016).
- 8 Hasselmo, M. E. Neuromodulation and cortical function: modeling the physiological basis of behavior. *Behav Brain Res* **67**, 1-27 (1995).
- 9 Berridge, C. W. & Waterhouse, B. D. The locus coeruleus-noradrenergic system: modulation of behavioral state and state-dependent cognitive processes. *Brain Res Brain Res Rev* **42**, 33-84 (2003).
- 10 Muñoz, W. & Rudy, B. Spatiotemporal specificity in cholinergic control of neocortical function. *Curr Opin Neurobiol* **26**, 149-160 (2014).
- 11 McCormick, D. A., Nestvogel, D. B. & He, B. J. Neuromodulation of Brain State and Behavior. *Annu Rev Neurosci* **43**, 391-415 (2020).
- 12 Constantinople, C. M. & Bruno, R. M. Effects and mechanisms of wakefulness on local cortical networks. *Neuron* **69**, 1061-1068 (2011).
- 13 Alitto, H. J. & Dan, Y. Cell-type-specific modulation of neocortical activity by basal forebrain input. *Front Syst Neurosci* **6**, 79 (2012).
- 14 Pinto, L. *et al.* Fast modulation of visual perception by basal forebrain cholinergic neurons. *Nat Neurosci* **16**, 1857-1863 (2013).
- 15 Fu, Y. *et al.* A cortical circuit for gain control by behavioral state. *Cell* **156**, 1139-1152 (2014).
- 16 Gasselino, C., Hohl, B., Vernet, A., Crochet, S. & Petersen, C. C. H. Cell-type-specific nicotinic input disinhibits mouse barrel cortex during active sensing. *Neuron* **109**, 778-787.e773 (2021).
- 17 Polack, P. O., Friedman, J. & Golshani, P. Cellular mechanisms of brain state-dependent gain modulation in visual cortex. *Nat Neurosci* **16**, 1331-1339 (2013).
- 18 Mahn, M., Prigge, M., Ron, S., Levy, R. & Yizhar, O. Biophysical constraints of optogenetic inhibition at presynaptic terminals. *Nat Neurosci* **19**, 554-556 (2016).
- 19 Lübke, J., Egger, V., Sakmann, B. & Feldmeyer, D. Columnar organization of dendrites and axons of single and synaptically coupled excitatory spiny neurons in layer 4 of the rat barrel cortex. *J Neurosci* **20**, 5300-5311 (2000).

- 20 Feldmeyer, D. Excitatory neuronal connectivity in the barrel cortex. *Front Neuroanat* **6**, 24 (2012).
- 21 Petreanu, L., Mao, T., Sternson, S. M. & Svoboda, K. The subcellular organization of neocortical excitatory connections. *Nature* **457**, 1142-1145 (2009).
- 22 Oberlaender, M. *et al.* Cell type-specific three-dimensional structure of thalamocortical circuits in a column of rat vibrissal cortex. *Cereb Cortex* **22**, 2375-2391 (2012).
- 23 Meyer, H. S. *et al.* Cell type-specific thalamic innervation in a column of rat vibrissal cortex. *Cereb Cortex* **20**, 2287-2303, doi:10.1093/cercor/bhq069 (2010).
- 24 Sievers, M. *et al.* Connectomic reconstruction of a cortical column. *bioRxiv*, 2024.2003.2022.586254 (2024).
- 25 Rogers, A. & Beier, K. T. Can transsynaptic viral strategies be used to reveal functional aspects of neural circuitry? *J Neurosci Methods* **348**, 109005 (2021).
- 26 Reardon, T. R. *et al.* Rabies Virus CVS-N2c(Δ G) Strain Enhances Retrograde Synaptic Transfer and Neuronal Viability. *Neuron* **89**, 711-724 (2016).
- 27 Khazipov, R. *et al.* Early motor activity drives spindle bursts in the developing somatosensory cortex. *Nature* **432**, 758-761 (2004).
- 28 Milh, M. *et al.* Rapid cortical oscillations and early motor activity in premature human neonate. *Cereb Cortex* **17**, 1582-1594 (2007).
- 29 Colonnese, M. T. & Khazipov, R. "Slow activity transients" in infant rat visual cortex: a spreading synchronous oscillation patterned by retinal waves. *J Neurosci* **30**, 4325-4337 (2010).
- 30 Ackman, J. B., Burbridge, T. J. & Crair, M. C. Retinal waves coordinate patterned activity throughout the developing visual system. *Nature* **490**, 219-225 (2012).
- 31 Yang, J. W. *et al.* Thalamic network oscillations synchronize ontogenetic columns in the newborn rat barrel cortex. *Cereb Cortex* **23**, 1299-1316 (2013).
- 32 An, S., Kilb, W. & Luhmann, H. J. Sensory-evoked and spontaneous gamma and spindle bursts in neonatal rat motor cortex. *J Neurosci* **34**, 10870-10883 (2014).
- 33 Dooley, J. C. & Blumberg, M. S. Developmental 'awakening' of primary motor cortex to the sensory consequences of movement. *Elife* **7**, 41841 (2018).
- 34 Valeeva, G. *et al.* Emergence of Coordinated Activity in the Developing Entorhinal-Hippocampal Network. *Cereb Cortex* **29**, 906-920 (2019).
- 35 Inácio, A. R., Nasretdinov, A., Lebedeva, J. & Khazipov, R. Sensory feedback synchronizes motor and sensory neuronal networks in the neonatal rat spinal cord. *Nat Commun* **7**, 13060 (2016).
- 36 Aronoff, R. *et al.* Long-range connectivity of mouse primary somatosensory barrel cortex. *Eur J Neurosci* **31**, 2221-2233 (2010).

Revised manuscript #: 2023-05-07457A

Distinct brain-wide presynaptic networks underlie the functional identity of individual cortical neurons

Point-by-Point Response

Referees' comments:

Referee #1 (Remarks to the Author):

It is evident from the rebuttal letter and the new version of the manuscript that the authors have made substantial revisions. The majority of my concerns raised with the initial manuscript have been adequately addressed, resulting in much improved version.

We thank the reviewer for the kind comments. We are pleased to hear most of the reviewer's concerns have been adequately addressed.

I acknowledge that the authors made major efforts to address my concerns regarding their statement regarding the role of neuromodulators and the effectiveness of blocking ACh and NE receptors. The revised version employs a rapid detection approach for responses mediated by GPCRs, including muscarinic ACh receptors (mAChRs). However, ACh responses mediated by nicotinic ACh receptors (nAChRs) are significantly faster than those detected by GRABACH (GPCR-activation-based ACh) indicator. For this it would be helpful to mention the spatiotemporal resolution of the indicator responses and demonstrate its onset at a higher temporal resolution. That indicator response is certainly fast but as the paper by Jing et al. (2020) demonstrates but I appears to be on a time scale that would miss a major part of the nAChR response. The recordings presented in the new Extended Data Fig. 4 also suggest that only the late desensitised component can be recorded but not. It would be useful if the authors could provide more detail on the response kinetics of the ACh and NE indicators.

The spatiotemporal resolution of the sensors was reported in the original papers in which the kinetics of these sensors were described (Jing et al., 2020; Feng et al., 2024) and these references are cited in the current manuscript.

Furthermore, under the recording conditions in this in vivo study, the application speed of an antagonist is also likely to be relatively slow and may therefore not result in an effective block of the response. I feel that a direct test of the effect of neuromodulatory input requires the block of the physiological sources of the neuromodulators (i.e. the nucleus basalis of Meynert and the locus coeruleus, respectively), e.g. by optogenetic inactivation. However, I am fully aware that this would be well beyond the scope of the present study.

We appreciate the reviewer's suggestions regarding the direct manipulation of neuromodulatory sources. However, this approach would not test the impact of direct neuromodulatory inputs to

the cortex. Additionally, we were concerned that optogenetic inhibition over extended periods using currently available inhibitory opsins would not be effective. Given our intention to inactivate these direct inputs and latter concerns, we believe that applying neuromodulatory R blockers is a more appropriate approach.

Nevertheless, I suggest that the authors substantially tone down their assertion that neuromodulatory inputs do not affect or play a limited role in spontaneous movement activity. Based on the experimental results, the assertion made in the paper is in my view too strong.

The reviewer raised a similar concern in the previous review, and we adequately revised the manuscript to acknowledge some effect of direct neuromodulatory inputs on movement-dependent individual neuronal activity.

Abstract (Page 1, Lines 25-26): “These are minimally affected by direct neuromodulatory inputs but primarily driven by glutamatergic inputs.”.

Discussion (Page 13, Lines 376-381): “The release of ACh and NE in cortex is closely linked to spontaneous movements, as also confirmed in our GRAB sensor experiments^{25,32-35}. Our results shows that these inputs may influence the activity levels, especially with respect to sensory responses, consistent with the gain modulation of sensory responses during movement by neuromodulation^{24,36,37}. However, our neuromodulatory blockade results suggest that the direct neuromodulatory inputs to the cortex are may not be the main drivers for the neuronal activity in relation to spontaneous movements.”.

Furthermore, the authors state on page 20 that the ‘stable, sensory-independent activity of wS1 L2/3 neurons may constitute a reflection of internal models of the body and environment that allow for the integration of sensory information ...’. It would be helpful if they could mention whether this is supposed to be exclusively the function of L2/3 excitatory neurons and whether neurons in other cortical layers may fulfil other cortical functions.

Regarding the statement on the internal models, we do not know whether it is somewhat a unique function of L2/3 cells or not.

Referee #2 (Remarks to the Author):

“Distinct brain-wide presynaptic networks underlie the functional identity of individual cortical neurons” by Inacio et al. is a beautiful paper with highly novel results that would be of wide interest, especially to cortical physiologists. I am very impressed by all the new experiments that the authors have conducted since the previous submission, particularly the distinction between VPM and POM, and the GRAB sensor experiments. In addition, I agree that investigating the differences in GABAergic neuron subtypes in the presynaptic network would be in exciting future direction – I will look forward to seeing it in the future! I only have a few minor comments and

suggestions for the authors, but overall, I think this manuscript is more than ready for publication in Nature.

We appreciate the reviewer's enthusiasm and are glad to know that most of the concerns have been effectively addressed.

1. Individual neurons are classified into movement uncorrelated and Mov_up and Mov_down neurons. It would be more informative if the authors had an accompanying quantification of the correlation between neural activity and whisker movement/locomotion as a continuous variable (e.g., Pearson correlation or linear decoder weight). Such a quantification is done in Fig. 2, but there are several other places where it would be useful.

1-1) Such variable could be shown as a histogram rather than a pie chart in Fig. 1h

The detailed analysis regarding the correlation between neuronal activity and whisker movements/locomotion is presented in Extended Data Fig. 1.

1-2) "Together, these results show that a subset of wS1 PNs reliably encodes spontaneous movements and that the coding of spontaneous movements and sensory stimuli is independent given that some neurons have both properties."

To make the claim that coding of spontaneous movements and sensory stimuli are truly independent, the authors would need to at least show the scatterplot across neurons of movement correlation vs sensory responsiveness. Alternatively, they could show that sensory-driven activity and spontaneous behavior-driven activity is largely orthogonal, as shown in V1 by Stringer et al (2019, DOI: 10.1126/science.aav789).

We agree that we cannot exclude an interaction between both signals, which may be particularly hard to discern in the wS1 solely based on statistical methods. We modified our statement to address the reviewer's concern.

Results (Page 4, Lines 94-98): "Together, these results show that a subset of wS1 PNs reliably encodes spontaneous movements and suggest that the coding of spontaneous movements and sensory stimuli may be independent given that some neurons have either one of these or both properties."

1-3) The correlation between spontaneous movement and Ach/NE concentration (measured with GRAB sensors) could be quantitatively compared with the distribution across neurons. Was the correlation higher or lower compared to Movup neurons?

While addressing the raised question would be informative, we expressed GRAB sensors from all neuronal types. Additionally, this approach does not provide single-cell resolution to detect individual neuronal sensitivity to ACh and NE.

1-4) How did the optogenetic terminal suppression experiments change the correlation with movement across neurons?

To address this question, we built a linear decoder using LED-off periods (normal thalamic input) and tested the model on data acquired during LED-on periods lasting 1 to 1.5 seconds (optogenetic thalamic terminal suppression in ArchT-expressing mice). We found a consistently larger decrease in R^2 values for ArchT-expressing mice (thalamic terminal suppression condition) compared to mice that did not express ArchT (control condition).

2. “Spontaneous movements, the proportion of spontaneous movement-dependent neuronal subsets, and cortical depth of the postsynaptic (target) neurons were similar between the two groups (Extended Data Fig. 9).”

This is worded in a confusing way. Please clarify that the former two (spontaneous movement and proportion of neuronal subset) refer to the two groups of mice, whereas the third point (cortical depth of postsynaptic neurons) refer to the two groups of electroporated neurons? Even though these ultimately mean the same thing, a clarification would be helpful.

We modified the text to address the reviewer’s comment.

Results (Pages 6-7, Lines 186-188): “The spontaneous movements and proportion of spontaneous movement-dependent neuronal subsets were comparable between both groups. In addition, the cortical depth of the movement-uncorrelated and movement-correlated postsynaptic neurons was similar (Extended Data Fig. 9).”

3. Line 243: Section heading “Local (wS1) presynaptic networks” – It should be mentioned somewhere in the Results section that only layer 2/3 excitatory neurons were traced.

That information was previously included in the Results (Lines 166-168), but we now added further clarification (Results, Lines 182-184).

Results (Pages 5-6, Lines 166-168): We first imaged wS1 L2/3 PNs and selected a target neuron based on its activity profile during spontaneous movements (example movement-correlated neuron, Fig. 3a, b).

Results (Page 6, Lines 182-184): We performed single-cell based retrograde tracing of two functionally distinct subsets of L2/3 PNs neurons: movement-uncorrelated ($n = 11$, 11 mice) and movement-correlated ($n = 11$, 11 mice) neurons (Fig. 3b, d).

The analysis of local (wS1) presynaptic networks includes all cortical layers.

4. “Individual glutamatergic presynaptic networks were often characterized by a smaller fraction of L4 compared to L2/3 or L5 neurons.” – Please clarify whether this was due to the fact that the authors only targeted non-sensory neurons, or whether the distribution is consistent with a typical wS1 L2/3 excitatory neuron.

This statement in the manuscript is intended to describe the distribution is consistent with a typical wS1 L2/3 excitatory neuron, in line with previous literature.

5. “All long-range presynaptic neurons were glutamatergic.” – Did the authors actually quantify this? If not, I would suggest rephrasing as “We expect all long-range presynaptic neurons to be glutamatergic.” There have been some reports of long-range inhibitory connections, particularly from subcortical regions.

We performed GABA immunostaining on all the brain tissues but did not detect any GABA⁺ presynaptic neurons in other brain regions. Thus, we stated “all long-range presynaptic neurons were glutamatergic.”

6. “We found that spontaneous movement-dependent activity is largely preserved even after combined bilateral whisker trimming and mystacial pad paralysis (Extended Data Fig. 15).” – I suggest clarifying that spontaneous movement refers to locomotion in this case.

We revised ‘spontaneous movement’ to ‘locomotion’ as suggested by the reviewer.

7. “We observed that light pulses elicited a relatively brief (~0.5 s) whisker movement in both control and ArchT-expressing mice (Extended Data Fig. 16).” Given that this twitch is also observed in control mice, it is likely behaviorally induced by mice detecting the optogenetic light. Could the authors please expand on what measures they took to prevent the mice from seeing the light? Did they try masking light or light shielding? I understand if these measures were impossible to combine with simultaneous 2P imaging, I just would appreciate some mention of this either in Results or in Methods (and acknowledgment that mice were likely able to see the optogenetic light).

We described our efforts to mask optogenetic light and added a statement acknowledging that the brief whisker movements are likely due to the mice perceiving the optogenetic light in the Methods (Pages 21-22, Lines 650-653): “To minimize the effect of light stimulation on the spontaneous movements of mice, we shielded the objective lens. However, this shield did not completely block the light stimulation. We observed a brief (< 0.5 s) whisker movement in both control and ArchT-expressing mice at the onset of light stimulation (Extended Data Fig. 16).”

8. Pupil dilation has been observed to be highly correlated with locomotion. Some mention or discussion of pupil dilation would provide a nice link with a large volume of previous literature. Do the authors think the results would be the same if pupil dilation was investigated, particularly with regards to connectivity differences and the lack of dependence on Ach and NE?

We predict pupil dilation during locomotion, as previously described, and have added this to the manuscript (Page2, Lines 55-56). In our working model (Extended Data Figure 18), we propose that the activity of neuromodulatory neurons increases during spontaneous movement, which strongly affects thalamic activity. While the direct neuromodulatory inputs to the cortex may not

significantly influence movement-dependent cortical neurons, those neurons that receive abundant thalamic inputs become sensitive to movement.